# Robust Inter-Series Dependency Modeling for Time Series Forecasting via Information-Theoretic Alignment

Wuqing Yu [1]  Weichen Guo [1]  Jian Zhou [1]  Shuyu Luo [2]  Jiacai Zhang [1]

## Abstract

While iTransformer pioneered general inter-variate dependency (IVD) modeling in Transformers for multivariate time series forecasting (MTSF), subsequent research on such universal paradigms has been surprisingly scarce. Through comprehensive analysis, we identify a critical structural inconsistency in Variate Transformers: typically capturing inter-variate dependencies via shallow self-attention layers while neglecting the critical requirement for deep-layer IVD modeling, which causes spurious correlations modeling and difficulties in model optimization. To address these limitations, we propose `CGTFra`, as a general framework for consistent IVD modeling. Specifically, we reconsider existing timestamp-based modeling and introduce a frequency-domain masking and resampling method for periodicity preservation, which serves as a general strategy for input feature enhancement. Additionally, `CGTFra` promotes consistent IVD modeling from two perspectives. Initially, a dynamic graph learning framework is integrated into Transformers to explicitly model IVD in deep network layer. Furthermore, grounded in the Information Bottleneck principle, we further propose a consistency-constrained alignment to learn more robust IVD and temporal feature representations. These three core design philosophies of `CGTFra` can be integrated into any existing Variate Transformer-based framework, and `CGTFra` achieves superior predictive performance across 13 long- and short-term datasets with high computational efficiency and desirable interpretability. Code is available at https://github.com/05Pikachu24/Consistent-CGTFra.

[1]School of Artificial Intelligence, Beijing Normal University, Beijing, China [2]School of Future Technology, South China University of Technology, Guangzhou, China. Correspondence to: Jiacai Zhang <jiacai.zhang@bnu.edu.cn>.

*Proceedings of the $43^{rd}$ International Conference on Machine Learning*, Seoul, South Korea. PMLR 306, 2026. Copyright 2026 by the author(s).

## 1. Introduction

Multivariate time series, such as traffic flow, are critical for forecasting the future dynamics of real-world systems. Multivariate time series forecasting (MTSF) is challenged mainly by two factors: the intricate temporal patterns of individual variables (intra-series dependency) and the dynamic dependencies among these variables (i.e., inter-series or inter-variate dependency), where one variable's fluctuation can affect the others (Zhang & Yan, 2023).

Recently, to achieve more accurate MTSF, Transformer-based networks have gained prominence due to their inherent strength in capturing long-range dependencies (Vaswani et al., 2017; Wang et al., 2025). However, after a comprehensive analysis of existing Transformer methods, we argue that they still face the following two significant limitations.

(1) **Over-reliance on Timestamps for Input Representation**. Existing methods typically employ learnable encodings derived from timestamps to capture temporal periodicity information, as seen in models including Informer (Zhou et al., 2021), Autoformer (Wu et al., 2021), iTransformer (Liu et al., 2024), VCformer (Yang et al., 2024)

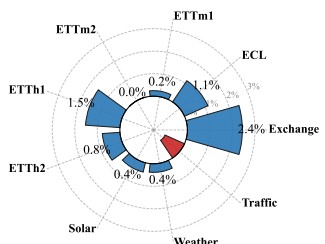

*Figure 1.* Impact of timestamps on iTransformer. The radar chart presents an improved (blue) or decreased (red) percentage.

and others. However, *timestamp formats can vary across datasets, and issues such as missing or erroneous timestamps all cannot be effectively handled. Its actual effectiveness, moreover, is yet to be fully established.*

To investigate the actual efficacy of such temporal information, we conducted an ablation study on iTransformer where we removed the timestamp embedding and instead up-sampled the input signal using a single linear layer. However, as shown in Figure 1, this substitution leads to performance improvements on eight datasets (Full results and more analysis are provided in Appendix B.1).

Therefore, we propose a novel and universal Frequency-

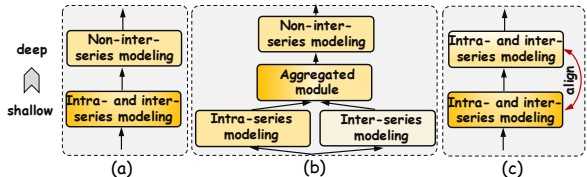

*Figure 2.* A comparison of paradigms for modeling temporal and inter-variable dependencies. (a) and (b) illustrate existing Transformer-based approaches for modeling IVD. (c) depicts the consistent dependency modeling paradigm proposed in this work.

domain Masking and Resampling (FMR) method, which performs learnable feature enhancement and periodicity capture directly on the frequency components of the signal by applying a learnable mask and a subsequent linear resampling. Detailed motivation is provided in Appendix B.

(2) **Inconsistency in Modeling Inter-variate Dependencies**. Transformer consists of two key stages: the multi-head self-attention (MHSA) layer and the feed-forward network (FFN). iTransformer (Liu et al., 2024) introduced the "Variate Transformer" paradigm, which explicitly models IVD by encoding each variable as an individual token. Nevertheless, we argue that a potential limitation exists here: an inconsistency in how temporal and inter-variate dependencies are modeled, that is, IVD are modeled exclusively within the shallow self-attention layers. In contrast, the deeper FFNs focus solely on capturing the temporal dynamics within each individual variable (see Figure 2(a)). We acknowledge that numerous Transformer variants have been proposed to better model IVD, including methods based on metric learning and graph transformers, such as DUET (Qiu et al., 2025), STGAGRTN (Wu et al., 2023a) and GL-STGTN (Li et al., 2024). However, a typical trait in these methods is that they integrate the learned variable dependencies into the self-attention layer, typically as an attention mask or a bias. We categorize this parallel fusion as the method depicted in Figure 2(b). We argue that ❶ **these methods do not address the inconsistency in modeling temporal and inter-variate dependencies, and such inconsistency results in capturing spurious correlations, severely compromising generalization capabilities under distribution shifts, and posing challenges for model optimization** (see Figure 5). ❷ **Crucially, it is difficult to transfer the IVD modeling of these improved methods to other architectures.**

The challenge then lies in how to implement **general** IVD modeling within the Transformer deep layers. Motivated by the theoretical parallels between self-attention in Transformers and information aggregation in Graph Neural Networks (GNNs), we propose a Dynamic Graph Learning (DGL) framework that dynamically optimizes the graph structure by integrating learnable static node priors with input-specific features. Concurrently, it strategically employs two linear layers to aggregate multi-hop neighbor information and ex-

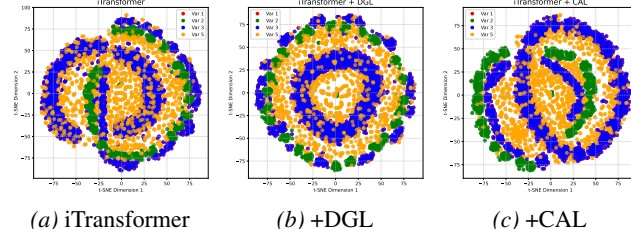

*(a)* iTransformer  *(b)* +DGL  *(c)* +CAL

*Figure 3.* T-SNE visualization for latent representations of variable 1, 2, 3, and 5 on the 1500 samples of ETTh1 test set. Based on the DTW and PCC metrics (as detailed in Figure 10), the dependency strengths of these variables are ranked in descending order: 1&3 > 3&5 > 2&5. Given that the learned latent embeddings of variable 1 and 3 nearly overlap, this suggests that a higher degree of dependency or similarity corresponds to a smaller spatial distance. More analysis is provided in Appendix K.1.

tract deep temporal features, which allows us to replace the FFNs in the Transformer with DGL. As depicted in Figure 2(c), we consistently model both temporal and inter-variate dependencies, underscoring the importance of modeling IVD throughout the network, not only in shallow MHSA. We elucidate the rationale for employing GNNs in deep layers to model IVD in Appendix D and E. By integrating DGL into iTransformer (replacing the FFNs), as shown in Figure 3(b), in the outermost layer, more blue elements are distributed within the intervals between the green ones, indicating that the deep IVD modeling provided by DGL effectively mitigates the anomaly observed in Figure 3(a), where the distance between variables 2 and 5 was incorrectly smaller than that between variables 3 and 5.

However, merely explicitly modeling IVD in deep layers may not guarantee effective constraints on robust IVD, given the black-box nature (stochastic optimization landscape) of neural networks. It remains questionable whether this architecture truly achieves such consistency. This raises a critical question: **do the dependency structures modeled at these two stages actually exhibit the expected similarity?**

Our analysis reveals that despite a high degree of similarity, discrepancies exist (as detailed in the Figure 11), which offer unique advantages: Self-attention mechanisms typically capture **global and dense dependencies**; however, these dense connections often include spurious correlations that do not reflect true underlying relationships. In contrast, GNNs excel at capturing differentiated **indirect and sparse dependencies**, we aim to find a balance between these two types of dependency modeling. Therefore, based on Information Bottleneck (IB) principle, we further propose a consistency alignment loss (CAL) constraint by introducing Kullback-Leibler (KL) divergence to quantify the distance between these two dependency distributions with solid information theoretical guarantees. As illustrated in Figure 3(c), the latent representations learned with CAL appropriately reflect the inter-variable distances across the majority of samples.

Synthesizing the foregoing analysis, we propose `CGTFra`, a compact framework that considers consistency in modeling IVD. Our primary contributions are as follows:

- We propose a novel, position-agnostic approach based on learnable frequency-domain masking and linear resampling, which serves not only as an effective supplement but also as a potential replacement for existing timestamps encoding or up-sampling methods.

- Driven by the key observation that consistent modeling of both intra- and inter-series dependencies across shallow and deep network layers is essential, we propose a dynamic graph learning framework that can be integrated into existing Variate Transformers as a universal method for deep IVD modeling.

- To the best of our knowledge, we are the first to investigate the relationship between IVD modeled at shallow and deep network layers. Building upon a tight theoretical upper bound, we reveal that the proposed consistency dependency constraint acts as a potential universal mechanism.

- `CGTFra` sets a new state-of-the-art in both long- and short-term time series forecasting on 13 datasets with superior computational efficiency (see Appendix O).

## 2. Related Work

**Application of Timestamp Encoding in Time Series Forecasting.** Inspired by the effectiveness of positional encoding in NLP, numerous Transformer-based studies in MTSF have adopted this technique. The fusion of timestamp positional and data encodings is primarily achieved through two strategies: **direct summation**, as seen in models like Informer, TimesNet (Wu et al., 2023b), Autoformer (Wu et al., 2021), and Fedformer (Zhou et al., 2022), or **concatenation**, employed by iTransformer and VCformer. Notably, a distinct approach is presented in GLAFF (Wang et al., 2024). This work proposes the independent learning of timestamp information—encompassing both historical and future timestamps—and the data features. These two streams of information are then fused using an adaptive weighting mechanism, leading to superior forecasting performance. However, an unbalanced reliance on calendar time can induce rigid periodic biases within the model rather than capturing the underlying dynamic patterns of the series. And, *timestamp formats can be different across diverse datasets.* Methods like GLAFF are ill-equipped to handle these situations.

**Modeling Inter-Variate Dependencies with Transformers.** Conventional temporal Transformers for MTSF typically encode information from different variables at the same timestamp into a single token. This approach, however, leads to a loss of IVD information, as seen in temporal

Transformer-based studies (Chen et al., 2024; Luo & Wang, 2024; Nie et al., 2023). Crossformer (Zhang & Yan, 2023) employs a tailored two-stage attention layer to explicitly model both intra- and inter-series dependencies. iTransformer encodes each individual time series as a single token, offering greater universality in modeling IVD compared to Crossformer. TokenGT (Kim et al., 2022) treats nodes and edges as independent, learnable tokens, which are then fed into the Transformer alongside the input tokens. DUET captures IVD in the frequency domain using metric learning. The resulting dependency is then integrated into the self-attention mechanism as a mask for the attention scores.

**Modeling Inter-Variate Dependencies with Graph Transformers.** SageFormer (Zhang et al., 2024) first employs a GNN to capture IVD from the input MTS. The resulting global, graph-enhanced embeddings are then fused with the original series to serve as the input for a vanilla Transformer. STGAGRTN (Wu et al., 2023a) utilizes a gating mechanism to fuse the IVD learned separately by a GAT and a spatial Transformer. GL-STGTN (Li et al., 2024) learns the graph structure from both global and local perspectives, and then the learned IVD are encoded into a spatial attention mechanism. In summary, existing researches can be broadly categorized into two main strategies: (1) methods like DUET, STGAGRTN, and GL-STGTN, which integrate learned IVD into the self-attention mechanism as a mask or bias; and (2) approaches such as SageFormer and TokenGT, which embed graph-structural information directly into the input embeddings. *However, a common limitation of all these methods is their failure to consider the consistency and correlation of IVD modeling between the shallow and deep network layers.* For the detailed implementation specifics of these methods, please see Appendix G.

## 3. Methodology

For MTSF tasks, let $\mathbf{X}=[\mathbf{X}_1^{1:T}, \mathbf{X}_2^{1:T}, \ldots, \mathbf{X}_N^{1:T}] \in \mathbb{R}^{T \times N}$ be the historical input, where $T$ is the input length and $N$ is the number of variates, and each $\mathbf{X}_N^{1:T} \in \mathbb{R}^T$ is the $N$-th variate, and thereby, we use `CGTFra` to forecast $\mathbf{Y}=[\mathbf{X}_1^{T+1:T+F}, \mathbf{X}_2^{T+1:T+F}, ..., \mathbf{X}_N^{T+1:T+F}] \in \mathbb{R}^{F \times N}$ during future $F$ time steps.

As illustrated in Figure 4, we propose `CGTFra`, a graph transformer framework designed for consistent IVD modeling. `CGTFra` inherits the Transformer's proficiency in capturing long-range dependencies while simultaneously demonstrating exceptional capabilities in modeling IVD. Technically, `CGTFra` is built upon three core design principles that is *plug-and-play* for existing Transformers: (1) A universal, adaptive Frequency-domain Masking and Resampling (`FMR`) (Upsampling or downsampling). (2) A Dynamic Graph Learning (`DGL`) framework that explicitly models IVD in the deep network layer. (3) A Consistency

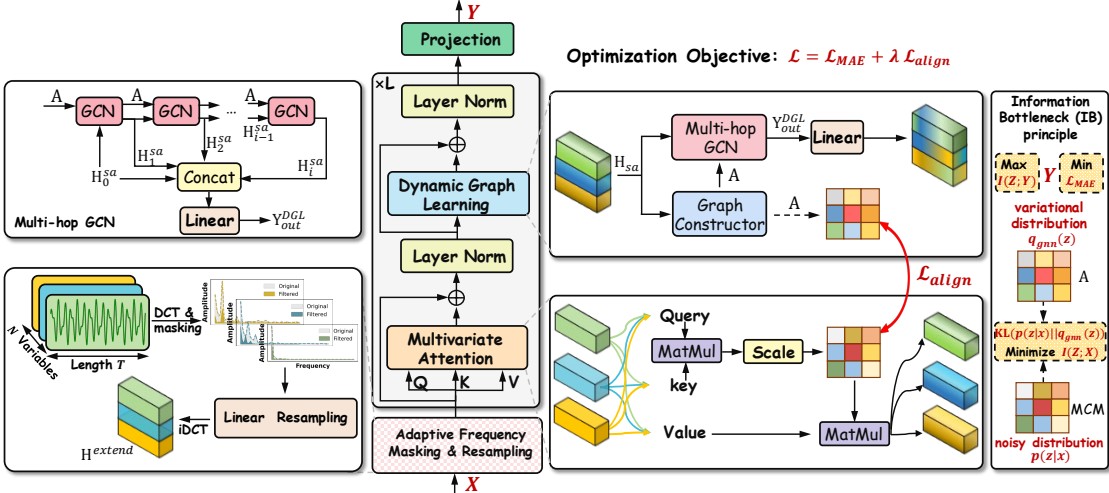

*Figure 4.* Illustration of proposed `CGTFra`. The output of the $i$-hop GCN is denoted by $\mathbf{H}_i^{sa}$. The deep `CGTFra` can be constructed by stacking multiple layers (×L) of self-attention and `DGL` modules. At this stage, each layer is subjected to alignment constraints (`CAL`). The right panel illustrates the Variational Information Bottleneck principle within `CGTFra`. The alignment loss $\mathcal{L}_{align}$ minimizes the KL divergence between the noisy encoder distribution $p(z|x)$ (MCM) and the compressed variational distribution $q_{gnn}(z)$ (A), effectively filtering noisy IVD while preserving predictive information maximized. See Appendix F for theoretical derivations.

Alignment Loss (`CAL`) that further promotes consistent IVD modeling between the shallow and deep network layers.

### 3.1. Adaptive Frequency Masking and Resampling

Compared to resampling directly in time domain with a linear layer, resampling in the frequency domain introduces a powerful inductive bias of a global receptive field. This paper utilizes DCT for frequency domain analysis (the motivation is provided in Appendix B). Furthermore, given that each variable possesses its own intrinsic dynamics, we learn an independent frequency mask for each variable. This allows the model to adaptively highlight critical frequencies and attenuating irrelevant or detrimental ones. Given an MTS $\mathbf{X}=\{\mathbf{X}_1, \mathbf{X}_2, ..., \mathbf{X}_N\} \in \mathbb{R}^{T \times N}$ where $\mathbf{X}_n=[x_0, x_1, ..., x_{T-1}]^\top$ denotes the sequence values for the $n$-th variable (For simplicity, we explicitly denote the variable dimension only when computing the DCT and iDCT), this process is formulated as:

$$\mathbf{F}_n(k) = c(k)\sqrt{\frac{2}{T}} \sum_{t=0}^{T-1} \mathbf{X}_n(t) \cos[\frac{\pi k(2t+1)}{2T}], \quad (1)$$

$$c(k) = \begin{cases} \sqrt{\frac{1}{2}}, & k = 0 \\ 1, & k > 0 \end{cases} \quad (2)$$

$$\tilde{\mathbf{F}} = \mathbf{F}(k) \odot \text{softplus}(\mathcal{M}), \quad (3)$$

where $\mathbf{F}(k), \tilde{\mathbf{F}} \in \mathbb{R}^{T \times N}$ represent the DCT coefficients and the masked frequency coefficients. $\mathcal{M} \in \mathbb{R}^{T \times N}$ denotes the variable-specific learnable mask. In Equation (1), $k \in \{0, 1, \ldots, T-1\}$ is the DCT index. Subsequently, a learnable linear layer is employed to perform

resampling on the masked frequency components, yielding the expanded frequency representation $\hat{\mathbf{F}} \in \mathbb{R}^{D \times N}$, $D$ is the hyperparameter of resampling size. Subsequently, the inverse Discrete Cosine Transform (iDCT) is applied to convert the frequency components $\hat{\mathbf{F}}$ back into a temporal signal $\mathbf{H}^{extend} \in \mathbb{R}^{D \times N}$. This process is formulated as:

$$\hat{\mathbf{F}} = \text{Resampling}(\tilde{\mathbf{F}}), \quad (4)$$

$$\mathbf{H}_n^{extend} = \sqrt{\frac{2}{D}} \sum_{k=0}^{D-1} c(k)\hat{\mathbf{F}}_n(k) \cos[\frac{\pi k(2t+1)}{2D}], \quad (5)$$

where Resampling($\cdot$) is implemented by the learnable linear interpolation. By performing masking and resampling within the frequency domain, the signal's periodicity is robustly preserved and even enhanced (see Appendix B.2). Therefore, the importance of timestamps is diminished.

### 3.2. Dynamic Graph Learning

Grounded in the theoretical analysis (detailed in Appendix D and E) comparing the similarities and distinctions between GNNs and self-attention mechanisms in modeling IVD, we strategically employ GNNs within the deep layers of the network. Our proposed compact, single-branch architecture distinguishes itself from existing Graph Transformer approaches—in terms of both motivation and methodology (see Appendix G)—which typically treat GNNs merely as a supplementary module for IVD modeling (often via an additional GNN branch). Driven by the necessity of deep-layer IVD modeling, we utilize the representations, containing global and dense IVD captured by self-attention, as the input for Dynamic Graph Learning (`DGL`). This design facilitates

the `DGL` to uncover sparser and more authentic correlations. Furthermore, this design principle is generalizable (*plug-and-play*) to existing Variate Transformers (see Section 4.4), demonstrating robust effectiveness.

Unlike Sageformer and MSGNet (Cai et al., 2024), which rely solely on self-learned node embeddings to construct graph structure—a process prone to learning spurious correlations (Fan et al., 2024), we inject the input (i.e., the output $\mathbf{H}_{sa} \in \mathbb{R}^{N \times D}$ of the self-attention layer) into the node embedding generation process. This allows us to define the graph topology from a global perspective based on the input tokens, aligning the global modeling by self-attention. Specifically, we first use a linear transformation to derive an adaptive gating weight $\mathbf{W}_{gating}$ for each node from the static node embedding and global input. This weight is then multiplied with a linearly transformed representation of the node features, obtaining a dynamic node embedding that is continuously updated throughout the network.

$$\mathbf{W}_{gating} = \text{ReLU}(\text{Tanh}(\text{Linear}(\text{Concat}(\mathbf{H}_{sa}^l, \Theta_*^l)))), \quad (6)$$

$$\Theta_*^l = \mathbf{W}_{gating} \odot \text{Linear}(\mathbf{H}_{sa}^l) + \Theta_*^l, * \in \{1, 2\}, \quad (7)$$

where $\Theta_*^l$ includes $\Theta_1^l, \Theta_2^l \in \mathbb{R}^{N \times nd}$, which are trainable parameters (with random initialization) of $l$-th layer, $nd$ is a hyperparameter, denoting the dimension of node. $\Theta_1^l$ and $\Theta_2^l$ employ the same update strategy as in Equation (7), but without parameter sharing. $\odot$ is the Hadamard Product. Then, the adjacent matrix $\mathbf{A}^l \in \mathbb{R}^{N \times N}$ of $l$-th layer can be represented as: $\mathbf{A}^l = \text{Softmax}(\text{ReLU}(\Theta_1^l \cdot (\Theta_2^l)^\top))$. Therefore, the graph structure at the $l$-th layer can be denoted as $\mathcal{G}^l = (\mathbf{A}^l, \mathbf{H}_{sa}^l)$. To reconcile the discrepancy between the local neighborhood aggregation of GNNs and the global modeling of Transformers' self-attention, we employ a multi-hop GCN (Hamilton et al., 2017) to capture IVD at the deep feature level. The information from different hop neighborhoods is then combined using a linear layer by $\mathbf{Y}_{out}^{DGL} = \text{MLP}(\text{GCN}(\mathbf{H}_{sa}, \mathbf{A}))$. By aggregating information from its $i$-hop neighborhood, an $i$-hop GCN enlarges each node's receptive field, enabling the capture of higher-order graph structures.

`DGL` strategically mirrors the two-layer MLP design of a FFN. Specifically, the first MLP layer is adapted to aggregate multi-hop neighborhood information, while the second MLP layer extracts temporal features from the deep representations that have already been enriched with IVD.

### 3.3. Consistency Alignment Loss Function

The self-attention mechanism in a Transformer is essentially a GNN operating on a fully-connected graph, which implies that they can describe the same underlying correlation structure. However, according our previous analysis of their diversity (see Figure 11), self-attention mechanisms often struggle to capture authentic IVD, whereas GNNs excel at

modeling indirect dependencies. Consequently, grounded in the Information Bottleneck (IB) principle, we further propose a consistency constraint that establishes a rigorous theoretical optimization upper bound (see Appendix F for detailed derivation). Following iTransformer, each variable $\mathbf{H}^{extend}[n, :] \in \mathbb{R}^{1 \times D}$, $n = 1, 2, \ldots, N$, is regarded as an independent token and the self-attention layer then is applied to model multivariate correlations:

$$head_i = \text{Softmax}\left(\frac{(\mathbf{H}^{extend}\mathbf{W}_i^Q) \cdot (\mathbf{H}^{extend}\mathbf{W}_i^K)^\top}{\sqrt{d_K}}\right), \quad (8)$$

where $\mathbf{W}_i^Q, \mathbf{W}_i^K \in \mathbb{R}^{D \times \frac{D}{h}}$ are the projection metrices of $i$-th head, and $h$ is the number of attention heads. We use $\text{MCM} \in \mathbb{R}^{h \times N \times N}$ ($\text{MCM} = \text{Concat}(head_1, ..., head_h)$) to represent the multivariate correlation map (a.k.a., attention score). Therefore, the total alignment loss of $L$ layer CGTFra for consistent IVD modeling can be formalized as follows by Kullback-Leibler (KL) Divergence:

$$\mathcal{L}_{align} = \sum_{l=1}^{L} \text{KL}(P_l \| Q_l) = \sum_{l=1}^{L} \sum_{k=1}^{N^2} e^{p_{l,k}}(p_{l,k} - q_{l,k}), \quad (9)$$

where $p_l = \log P_l = \log(\text{softmax}(\text{Vec}(\text{Avg}(\text{MCM}^l))))$, and $q_l = \log Q_l = \log(\text{softmax}(\text{Vec}(\mathbf{A}^l)))$. In our implementation, we directly compute the log-probabilities to avoid numerical instability such as $\log(0)$ errors. $\text{Avg}(\cdot)$ denotes averaging the attention score along $h$ attention heads, and $\text{Vec}(\cdot)$ denotes vectorizing the correlation matrix into a one-dimensional vector. Therefore, the total loss function for optimizing CGTFra is formulated as:

$$\mathcal{L}_{total} = \mathcal{L}_{MAE} + \lambda \mathcal{L}_{align}, \quad (10)$$

where $\mathcal{L}_{MAE} = \frac{1}{F} \sum_{i=1}^{F} |y_i - \hat{y}_i|$ represents the Mean Absolute Error (MAE) for evaluating prediction accuracy with the forecasting length $F$. $y_i$ and $\hat{y}_i$ are the ground truth and predicted value at time $i$, and $\lambda$ is a hyperparameter, controlling the contribution of alignment loss. Here, for simplicity, we omit the batch dimension and illustrate the loss calculation for a single variable.

We provide the theory guarantee for the design of Equation (10) based on the Variational Information Bottleneck principle. Specifically, minimizing $\mathcal{L}_{MAE}$ encourages the model to maximize the mutual information between the representation and the ground truth, while minimizing $\mathcal{L}_{align}$ compresses the representation to filter out noisy IVD. Detailed theoretical derivations are provided in Appendix F.

## 4. Experiments

### 4.1. Datasets

We select 13 real-world datasets to comprehensively verify our `CGTFra` following iTransformer, including ETT

(4 subsets), Weather, Exchange, Electricity (ECL), Solar-Energy, Traffic, PEMS03, PEMS04, PEMS07 and PEMS08. All datasets are preprocessed following iTransformer. And more details of these datasets are provided in Appendix H.

### 4.2. Baselines and Experimental Settings

We choose 13 sota forecasting methods as our benchmarks, including (1) inter-series dependency models: DUET (Qiu et al., 2025), Soatten (Wu, 2025), Vcformer (Yang et al., 2024), iTransformer (Liu et al., 2024), Crossformer (Zhang & Yan, 2023), and MSGNet (Cai et al., 2024); (2) intra-series dependency approaches: PatchTST (Nie et al., 2023), FilterNet (Yi et al., 2024), RLinear (Li et al., 2023), TiDE (Das et al., 2023), TimesNet (Wu et al., 2023b), DLinear (Zeng et al., 2023) and TimePro (Ma et al., 2025). Following established practice, we evaluate `CGTFra` using Mean Absolute Error (MAE) and Mean Squared Error (MSE). Additional implementation details are in the Appendix I.

### 4.3. Main Results

The long-term and short-term forecasting results are presented in Table 1 and Table 8. Overall, `CGTFra` demonstrates superior performance in both forecasting tasks. This superiority is particularly pronounced on datasets with a large number of variables, such as ECL and Traffic, where modeling IVD poses a significant challenge for existing methods. Specifically, compared to DUET, `CGTFra` reduces MSE (MAE) by 5.1% (4.5%) on the Traffic. Additionally, in most scenarios, `CGTFra` exhibits enhanced performance when applied to datasets with inherent low predictability (see Table 6), including ETT and Solar, showing the effectiveness of `CGTFra` to modeling long-term intra- and inter-variate dependencies.

### 4.4. Framework Generality

To evaluate the effectiveness and scalability of the three core designs in `CGTFra`: Frequency Masking and Resampling (`FMR`), Dynamic Graph Learning (`DGL`) framework, and Consistency Alignment Loss (`CAL`), we conducted a series of integration and replacement experiments within existing SOTA models, including DUET, iTransformer, VCformer, FilterNet and CASA (Lee et al., 2025). For fair comparison, we use their originally published hyperparameter settings. "+ FMR", "+ DGL": substituting their input up-sampling methods with our FMR and their FFNs with our `DGL`. "+ CAL": on top of the `DGL` substitution, we introduce the `CAL`. The averaged comparison results are presented in Table 2. `FMR` and `DGL` demonstrated consistent performance improvements in almost all datasets, and **the substantial performance gains brought by `DGL` underscore the importance of deep IVD modeling, which has been entirely overlooked in their studies**. In addition, we observe that

by introducing `CAL`, compared to their original performance, iTransformer (VCformer) reduces the MSE by **4.9% (7.4%) and 7.4% (10.2%) on Weather and ECL datasets, respectively, approaching or even surpassing the latest sota methods DUET and TimePro**. Furthermore, to specifically verify the efficacy of the `DGL` within the broader family of Variate Transformers, we integrated it into four architectures mentioned in the iTransformer: iFlashformer (Dao et al., 2022), iFlowformer (Wu et al., 2022), iInformer (Zhou et al., 2021), and iReformer (Kitaev et al., 2020), and the full results are in Table 12. Since MAE and MSE primarily focus on point-wise rather than sequence-level predictions, therefore, we provide additional evaluation metrics in Section K to validate the effectiveness of `DGL` and `CAL`.

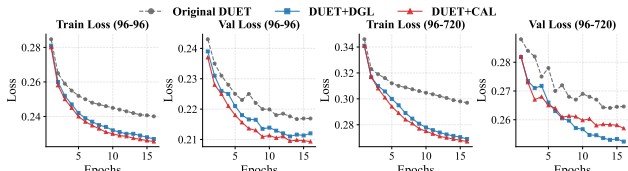

*Figure 5.* Visualization of training and validation loss curves for DUET on two prediction scenarios on ECL. Loss curves for additional datasets and baselines are provided in Appendix K.2.

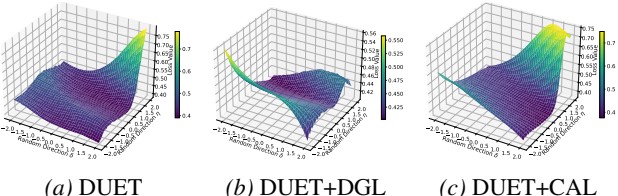

| *(a)* DUET | *(b)* DUET+DGL | *(c)* DUET+CAL |

*Figure 6.* Loss landscape visualizations around the converged model parameters on the same batch of ECL test set (96-96).

As shown in Figure 5, we plot the training and validation loss curves of DUET after incorporating `DGL` and `CAL`. **The trajectories indicate that deep IVD modeling accelerates parameter adjustment towards lower loss**. And in Figure 6, the corresponding loss landscape comparison shows that the introduction of `CAL` results in a smoother optimization landscape with wider valleys, indicating that the model possesses superior generalization performance. More analysis is provided in Appendix K.2.

### 4.5. Ablation Study

The comparison between aba1, and `CGTFra` vs aba3, demonstrates that the `FMR`, by effectively purifying and enhancing input features, substantially enhances the robustness of deep-layer IVD modeling, particularly on the ECL and Traffic. Furthermore, by introducing `DGL` and `CAL` (see aba1 vs aba2 and `CGTFra` vs aba2), consistent performance improvement indicates that constraining the consistency between shallow- and deep-layer IVD modeling enables the

*Table 1.* Long-term forecasting results with fixed input length $T$=96 and forecasting horizons $F \in \{96, 192, 336, 720\}$. The results are averaged from four forecasting horizons. Full results, short-term forecasting results, and the additional comparison scenarios (with different input lengths) are all provided in Appendix J. **Bold**: best results, underline: second best one.

| Models | CGTFra (*Ours*) | | DUET (KDD'25) | | TimePro (ICML'25) | | Soatten (AAAI'25) | | VCformer (IJCAI'24) | | FilterNet (NeurIPS'24) | | iTransformer (ICLR'24) | | MSGNet (AAAI'24) | | PatchTST (ICLR'23) | |
|---|---|---|---|---|---|---|---|---|---|---|---|---|---|---|---|---|---|---|
| Metrics | MSE | MAE | MSE | MAE | MSE | MAE | MSE | MAE | MSE | MAE | MSE | MAE | MSE | MAE | MSE | MAE | MSE | MAE |
| ETTm1 | 0.388 | **0.386** | 0.390 | 0.393 | 0.391 | 0.400 | 0.394 | 0.402 | 0.387 | 0.397 | **0.384** | 0.398 | 0.407 | 0.410 | 0.398 | 0.411 | 0.387 | 0.400 |
| ETTm2 | 0.277 | **0.316** | 0.280 | 0.324 | 0.281 | 0.326 | 0.287 | 0.331 | 0.285 | 0.330 | **0.276** | 0.322 | 0.288 | 0.332 | 0.288 | 0.330 | 0.281 | 0.326 |
| ETTh1 | **0.434** | **0.427** | 0.443 | 0.436 | 0.438 | 0.438 | 0.447 | 0.440 | 0.439 | 0.437 | 0.440 | 0.432 | 0.454 | 0.447 | 0.452 | 0.452 | 0.469 | 0.454 |
| ETTh2 | **0.369** | **0.394** | 0.372 | 0.397 | 0.377 | 0.403 | 0.379 | 0.405 | 0.377 | 0.403 | 0.378 | 0.404 | 0.383 | 0.407 | 0.396 | 0.417 | 0.387 | 0.407 |
| Exchange | **0.312** | **0.382** | 0.318 | 0.384 | 0.352 | 0.399 | 0.359 | 0.404 | 0.355 | 0.402 | 0.356 | 0.395 | 0.360 | 0.403 | 0.399 | 0.430 | 0.367 | 0.404 |
| Weather | **0.238** | **0.260** | 0.251 | 0.273 | 0.251 | 0.276 | 0.245 | 0.273 | 0.258 | 0.282 | 0.245 | 0.272 | 0.258 | 0.278 | 0.249 | 0.278 | 0.259 | 0.281 |
| ECL | **0.165** | **0.253** | 0.172 | 0.258 | 0.169 | 0.262 | 0.166 | 0.259 | 0.180 | 0.267 | 0.173 | 0.268 | 0.178 | 0.270 | 0.194 | 0.300 | 0.205 | 0.290 |
| Solar | **0.224** | **0.228** | 0.237 | 0.233 | 0.232 | 0.266 | 0.229 | 0.261 | - | - | - | - | 0.233 | 0.262 | - | - | 0.270 | 0.307 |
| Traffic | **0.427** | **0.257** | 0.451 | 0.269 | - | - | 0.437 | 0.286 | 0.483 | 0.325 | 0.463 | 0.310 | 0.428 | 0.282 | - | - | 0.555 | 0.362 |

*Table 2.* Verification of Framework Generality. Results are averaged from four forecasting horizons. Full results, additional valuation metrics and further analysis are in Appendix K. For a fair comparison, the results in Table 1 are taken from their officially released reports, whereas the results below are reproduced under our experimental environment (with identical hyperparameters as publicly released by the authors), and consequently, some discrepancies exist. "–" denotes that the original method was not evaluated on certain datasets, or that we encountered out-of-memory issues. "iTrans" and "Filter" denote iTransformer and FilterNet, respectively.

| Datasets | | ETTm1 | | ETTm2 | | ETTh1 | | ETTh2 | | Exchange | | Weather | | ECL | | Solar | | Traffic | |
|---|---|---|---|---|---|---|---|---|---|---|---|---|---|---|---|---|---|---|---|---|
| Metrics | | MSE | MAE | MSE | MAE | MSE | MAE | MSE | MAE | MSE | MAE | MSE | MAE | MSE | MAE | MSE | MAE | MSE | MAE |
| DUET | original | 0.391 | 0.394 | 0.279 | 0.322 | 0.449 | 0.440 | 0.372 | 0.398 | 0.309 | 0.380 | 0.247 | 0.270 | 0.172 | 0.258 | **0.241** | **0.246** | 0.451 | 0.269 |
| | + DGL | **0.389** | **0.391** | **0.277** | **0.320** | 0.444 | 0.436 | **0.368** | **0.395** | **0.296** | **0.373** | 0.247 | 0.272 | 0.166 | 0.255 | 0.243 | 0.258 | **0.448** | **0.268** |
| | + CAL | 0.390 | 0.392 | 0.279 | 0.322 | **0.438** | **0.432** | 0.369 | **0.395** | 0.305 | 0.376 | **0.237** | **0.263** | **0.164** | **0.253** | 0.242 | 0.253 | 0.452 | 0.269 |
| iTrans | original | 0.408 | 0.412 | 0.293 | 0.337 | 0.457 | 0.449 | 0.384 | 0.407 | 0.369 | 0.409 | 0.262 | 0.283 | 0.176 | 0.268 | 0.235 | 0.261 | **0.422** | 0.282 |
| | + FMR | 0.403 | 0.406 | **0.291** | **0.333** | 0.448 | **0.440** | **0.381** | **0.406** | 0.358 | 0.404 | 0.259 | 0.282 | 0.175 | 0.266 | **0.229** | **0.260** | 0.423 | 0.281 |
| | + DGL | 0.400 | 0.406 | 0.293 | 0.335 | 0.449 | 0.442 | 0.390 | 0.412 | 0.368 | 0.409 | 0.252 | 0.278 | 0.169 | 0.263 | 0.234 | 0.263 | 0.434 | 0.288 |
| | + CAL | **0.397** | **0.403** | **0.291** | **0.333** | **0.444** | **0.440** | 0.385 | 0.408 | **0.354** | **0.403** | **0.249** | **0.276** | **0.163** | **0.257** | 0.233 | 0.262 | 0.436 | 0.286 |
| VCformer | original | 0.404 | 0.406 | 0.292 | 0.334 | 0.488 | 0.460 | **0.384** | **0.405** | 0.358 | **0.403** | 0.269 | 0.286 | 0.186 | 0.278 | - | - | - | - |
| | + FMR | 0.398 | 0.402 | 0.291 | 0.333 | 0.457 | **0.441** | 0.385 | 0.406 | 0.367 | 0.409 | 0.265 | 0.285 | 0.182 | 0.275 | - | - | - | - |
| | + DGL | 0.398 | **0.401** | 0.289 | 0.333 | 0.456 | 0.447 | 0.389 | 0.410 | 0.363 | 0.404 | **0.249** | **0.275** | 0.174 | 0.266 | - | - | - | - |
| | + CAL | 0.398 | 0.402 | **0.287** | **0.331** | 0.451 | 0.444 | **0.384** | 0.406 | 0.361 | 0.406 | **0.249** | **0.275** | **0.167** | **0.261** | - | - | - | - |
| CASA | original | **0.391** | **0.400** | 0.279 | 0.323 | **0.442** | **0.440** | 0.383 | 0.406 | - | - | 0.249 | 0.276 | 0.172 | 0.265 | 0.226 | 0.261 | **0.427** | **0.278** |
| | + FMR | 0.392 | 0.401 | **0.277** | **0.322** | **0.442** | **0.440** | **0.378** | **0.404** | - | - | **0.245** | **0.273** | **0.169** | **0.263** | **0.223** | **0.259** | 0.444 | 0.279 |
| Filter | original | 0.384 | **0.398** | 0.277 | **0.322** | 0.451 | **0.437** | **0.379** | **0.405** | - | - | 0.253 | 0.280 | 0.179 | 0.272 | - | - | 0.460 | 0.304 |
| | + FMR | **0.383** | **0.398** | **0.276** | **0.322** | **0.450** | **0.437** | **0.379** | **0.405** | - | - | **0.248** | **0.276** | **0.177** | **0.271** | - | - | **0.455** | **0.300** |

*Table 3.* Ablation studies on five diverse datasets. The results are averaged from four forecasting horizons. Full results are provided in Table 15 of Appendix. F/D/C: FMR/DGL/CAL.

| Part | F/D/C | ETTm1 MSE | ETTh1 MSE | Weather MSE | ECL MSE | Traffic MSE |
|---|---|---|---|---|---|---|
| **CGTFra** | ✓/✓/✓ | **0.388** | **0.434** | **0.238** | **0.165** | **0.427** |
| aba1 | ✓/✗/✗ | 0.397 | 0.442 | 0.245 | 0.170 | 0.431 |
| aba2 | ✓/✓/✗ | 0.389 | 0.437 | 0.242 | 0.168 | 0.430 |
| aba3 | ✗/✓/✓ | 0.392 | 0.437 | 0.243 | 0.173 | 0.444 |

model to achieve a more robust balance of dependencies. To further validate the necessity of modeling inter-variable dependencies at deeper layers, we present experiments on variants of CGTFra in Appendix L.

### 4.6. Analysis of Inter-Series Dependency Modeling

To further analyze CGTFra's effectiveness in modeling IVD and extracting complex temporal dynamics, we select a sample from the Weather dataset's test set (all variable dynamics are in Figure 12 (b)). Within this sample (with 21 variables), four highly correlated variables (variables 3, 7, 8, and 13) are chosen for visualization and analysis.

As depicted in Figure 7, we visualize the prediction curves of four variables ([3,7,8,13]) predicted by CGTFra, alongside the PCC and DTW among the ground truth, and CGT-Fra predicted results. We observed that although the predicted sequences do not greatly match with the true sequences, the overall trends are correctly captured. Furthermore, the close proximity of the predicted PCC and DTW values to their true counterparts indicates the model's commendable ability to capture inter-variate dependencies. Figure 27 presents a comparison of prediction curves for these four variables. CGTFra demonstrates superior trend forecasting performance compared to iTransformer and DUET, both of which are also capable of modeling inter-variable dependencies.

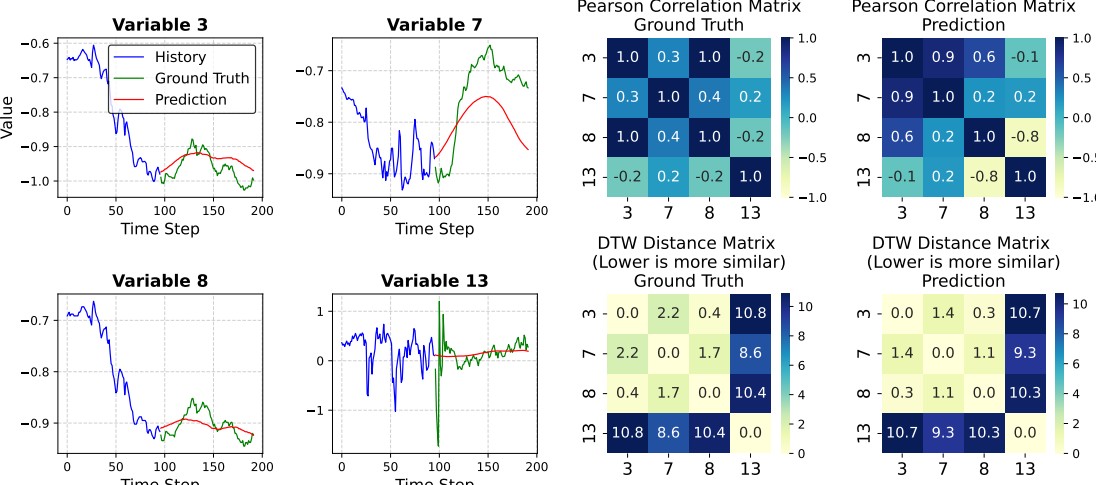

Figure 7. Prediction curves for CGTFra (Input 96-Predict 96) and the DTW and PCC comparison between ground truth and predicted sequences among variables [3, 7, 8, 13]. According to DTW and PCC, variable 3 exhibits a strong association with variable 8, while variable 7 also shows substantial correlations with both variables 3 and 8, as indicated by small DTW distances and high PCC.

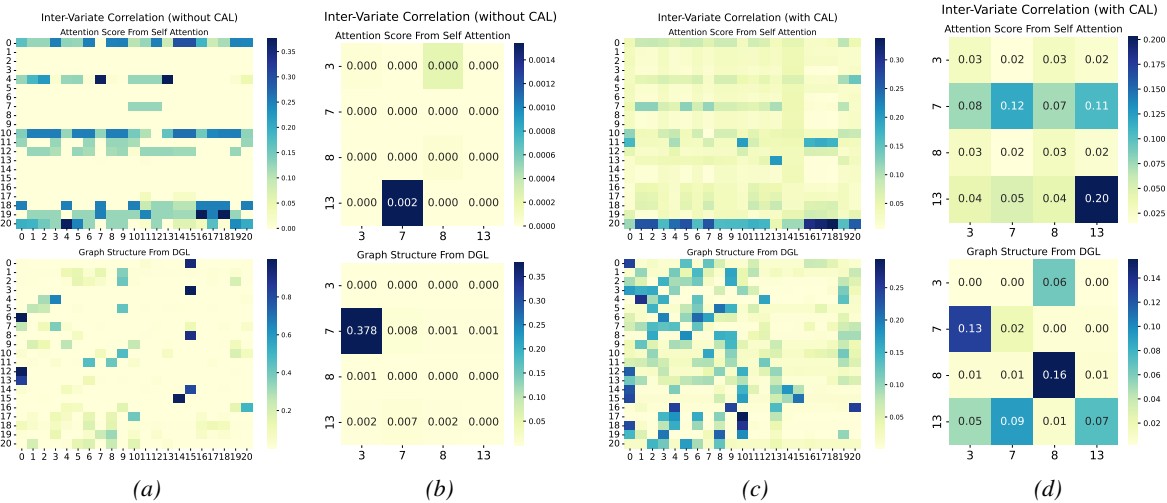

Figure 8. Inter-variate correlation learned by CGTFra on the test sample (Input96-Predict96). (a) and (b): Dependencies from **CGTFra without CAL**; (c) and (d): Dependencies from the **complete CGTFra with CAL**. (a) and (c): Correlation matrices from the shallow self-attention layer and deep DGL; (b) and (d): Zoomed-in visualization of dependencies for variables 3, 7, 8, and 13.

Moving to Figure 8, we present two inter-variate correlation matrices learned by CGTFra from the selected test sample: one from the self-attention layer and the other from the DGL. Observing Figure 8(b), we are surprised to find that, **without the CAL constraint, neither the self-attention layer nor the DGL success to capture critical dependencies.** This phenomenon is not attributed to a performance degradation caused by introducing DGL, but rather likely represents an inherent modeling challenge for the network (CGTFra's performance without CAL in the Weather test set is MSE: 0.157 and MAE: 0.195, both outperforming existing methods, as shown in Table 15 and 7). Nevertheless, DGL still successfully captured the correlation between variables 3 and 7 (see coordinates (3, 7)), **which is consistent with our**

**analysis in Figure 11 on ETTh1, where DGL is shown to capture indirect dependencies (between variable 4 and 5). This finding indicates that, compared with the global self-attention mechanism, GNNs possess an advantage in capturing indirect (or potential) dependencies by aggregating information from adjacent nodes**—for example, in Weather dataset, the relationship between variable 3 and variable 8 is apparent (direct), that between variable 3 (or 8) and variable 7 constitutes an indirect dependency (they also have smaller DTW distances and higher PCC).

Upon the introduction of CAL, both the self-attention layer and DGL effectively model prominent dependency correlations, as illustrated in Figures 8(c) and 8(d). Let us first

examine two strongly correlated variables: variable 3 and 8 (see (8, 3)). The self-attention layer capture a weight of 0.03, whereas `DGL` captures a weight of 0.06. Subsequently, we observe variable 3 and 7 (see (3, 7)), where the self-attention layer learns a weight of 0.08, while `DGL` captures 0.13. Furthermore, for variable 8 and 7 (see (8, 7)), they show 0.07 and 0.0, respectively. These observations suggest that the self-attention mechanism and `DGL`, possess distinct focuses and advantages. Crucially, the introduction of `CAL` promotes both mechanisms to achieve a more balanced and robust dependency correlations.

Figure 9 visualizes the t-SNE (Maaten & Hinton, 2008) embeddings learned from 1,500 test samples of the Weather dataset. Consistent with prior analysis, the embeddings for variables 3 and 8 learned by all three models (`CGTFra`, iTransformer, and DUET) are observed to be nearly overlapping (owing to their strong dependency). Building upon this, `CGTFra` demonstrates a shorter intra-variable distance, indicating that its representations for the same variable across different samples are more compact. Furthermore, in the embedding space of `CGTFra`, variable 13 is positioned more distantly from the others, and its sample representations are more tightly clustered. These observations suggest that `CGTFra` possesses a superior representation capability for learning individual variable features while more accurately capturing their inter-dependencies.

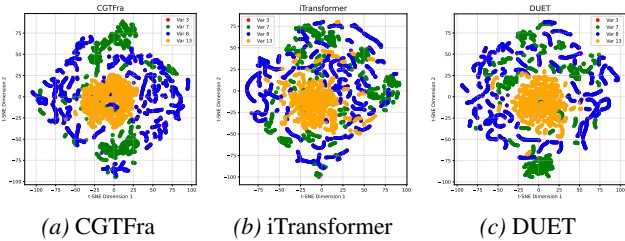

*(a)* CGTFra     *(b)* iTransformer     *(c)* DUET

*Figure 9.* T-SNE visualization for variable 3, 7, 8, and 13 on the 1500 samples of Weather test set.

## 5. Conclusion

By conducting a theoretical investigation into the distinctions and connections between how Variate Transformers and GNNs model IVD, this paper proposes `CGTFra`. This framework addresses the limitation of existing Variate Transformers that neglect deep-layer IVD modeling, **in a generic manner**. Furthermore, based on Information Bottleneck principle, we introduce, for the first time, a consistency constraint applied to IVD learned by both self-attention and deep graph learning frameworks, enabling the model to capture more consistent and robust IVD. This novel learning paradigm has been validated across multiple existing Variate Transformers. We believe that exploring further mutual guidance principles between graph structures and Transformer-based inter-variate dependency modeling repre-

sents a promising future research direction. We will further investigate whether consistent inter-variable dependency modeling remains necessary for non-variate Transformer frameworks. Additional limitations about `CGTFra` are provided in Appendix P.

## Acknowledgements

We thank the Area Chairs and Reviewers for their detailed feedback and constructive comments. This work is funded by the Special Fund for Research on National Major Research Instruments of the Nature Science Foundation of China (NO. 62227801).

## Impact Statement

Modeling inter-variate dependencies is pivotal in multivariate time series analysis. Since iTransformer, however, subsequent research on universal dependency modeling has been scarce. Identifying that the lack of deep-layer dependency modeling leads to optimization difficulties, we propose a consistent inter-variate dependency modeling framework grounded in the Information Bottleneck principle. This approach addresses existing limitations with high interpretability and generality. This paper presents work whose goal is to advance the field of machine learning. There are many potential societal consequences of our work, none of which we feel must be specifically highlighted here.

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

## A. Detailed Analysis of IVD Similarity Modeled by Self-Attention and DGL

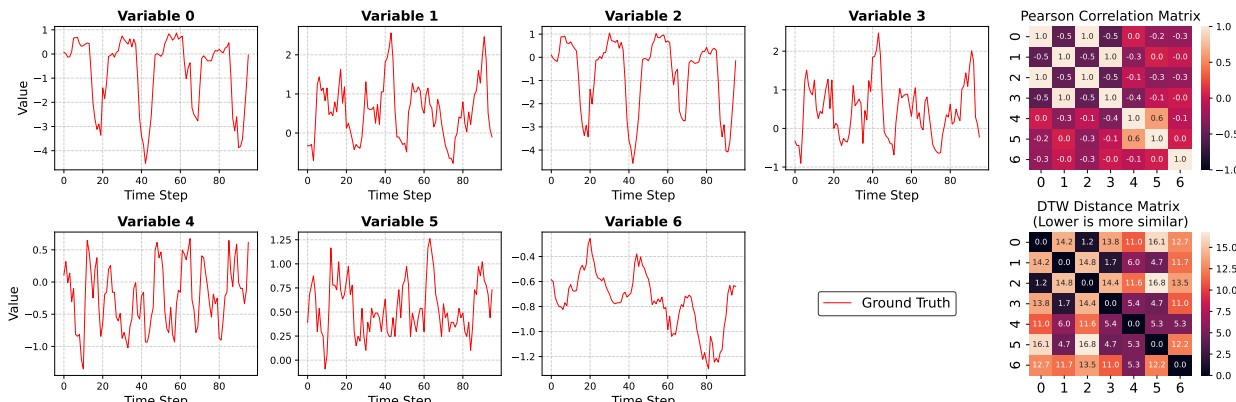

*Figure 10.* Intra- and inter-series dynamics on ETTh1 dataset. PCM and DTW are used to reveal inter-variable similarities and dependencies (See Appendix C for more details). We observe two highly similar pairs of variables: variables 0 with 2, and variables 1 with 3, and these pairs exhibit high PCM coefficients and low DTW distances, as indicated at coordinates (2,0) and (3,1), where (x-axis, y-axis) correspond to variable indices. Furthermore, their dependency patterns with other variables are also analogous (see row 0 vs. row 2, row 1 vs. row 3) in both the PCM and DTW matrices. Additionally, the strong correlation between variables 4 and 5 (see coordinate (5,4) in PCM and DTW), is noteworthy and will be further discussed in the context of Figure 11.

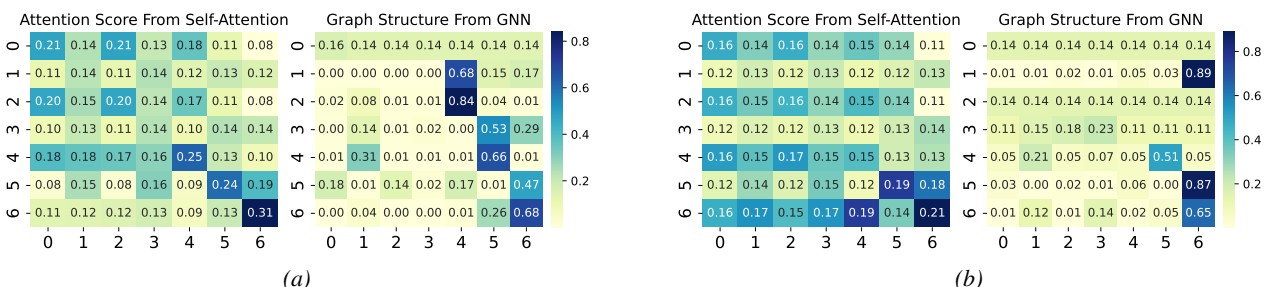

*Figure 11.* Comparative analysis of dependency matrices from self-attention and GNN (derived from the 7 variables in Figure 10). (a) before alignment; (b) after applying the alignment constraint. Consistent with the PCM and DTW matrices in Figure 10, the self-attention mechanism successfully captures dependencies between highly similar variables, such as the pairs (2, 0) and (3, 1), and show similar dependency correlations with other variables (see rows (0 vs. 2) and (1 vs. 3) in Attention Score Map). **However, we observe that the dependency matrices in self-attention are relatively dense, and the weight disparity between strongly and weakly correlated variables is minimal. And self-attention fails to capture less direct correlations, such as the one between variables 4 and 5 (See Figure 10, their PCC and DTW are 0.6, 5.3, respectively), which is successfully identified by the GNN (see coordinate (5,4)).** This result effectively demonstrates the efficacy of using DGL to model IVD in the deeper layers of our network. However, we also observed that the dependencies modeled by DGL can be exaggerated in some cases (e.g., at coordinate (4, 2)). To address this, we further introduced CAL based on information bottleneck principle (see Appendix F) to impose constraints on the IVD modeling. As shown by the graph structure in Figure (b), this inconsistency is significantly mitigated: compared to Figure (a), the KL divergence between the attention score and graph structure reduces form 0.0260 to 0.0249.

## B. Motivation of Frequency Masking and Resampling

Benefiting from the global receptive field of the frequency domain space, analyzing time series in the frequency space has become a prevailing trend, as seen in methods such as Fedformer (Zhou et al., 2022), TSLANet (Eldele et al., 2024), FilterNet (Yi et al., 2024), and DUET (Qiu et al., 2025). However, these approaches rely on the Discrete Fourier Transform (DFT) for frequency-domain analysis. Since DFT involves both real and imaginary components, it is computationally more complex than the Discrete Cosine Transform (DCT). Moreover, methods such as TSLANet and FilterNet primarily perform filtering on frequency components—similar to the masking mechanism proposed in this work—before transforming the filtered components back into the time domain for subsequent abstract feature learning. **This procedure introduces a potential risk: if critical frequency information is inadvertently filtered out (similar to Gibbs Phenomenon (Hewitt & Hewitt, 1979)), the subsequent feature extractor may struggle to capture informative representations. Consequently, such methods require both carefully designed frequency-domain filters and well-structured downstream feature extractors to achieve competitive performance.** Therefore, this paper proposes leveraging DCT to directly conduct frequency-domain

analysis in the real-valued space and applying linear resampling to the masked frequency components, transcending the independence assumption of individual frequency components, enabling explicit cross-frequency interaction and providing a more expressive latent space for reconstructing complex temporal dynamics, which makes the implementation of a **general** feature enhancement module feasible. To this end, we provide a theoretical analysis of the relationship between DFT and DCT as well as their computational complexity.

Like Section 3, let $\{f(l)\}, l = 0, 1, \ldots, L - 1$ be a input sequence. And let an extended sequence $\{e_l\}$ be symmetric about the $(2L - 1)/2$ point, that is, $e_l$ can be constructed by:

$$e_l = \begin{cases} f(l), & l = 0, 1, ..., L - 1 \\ f(2L - l - 1), & l = L, L + 1, ..., 2L - 1 \end{cases} \tag{11}$$

Here, suppose $L$=4, then the $\{f(l)\}$ and $\{e_l\}$ are:

$$\{f(l)\} = \{f(0), f(1), f(2), f(3)\}$$

$$\{e_l\} = \{f(0), f(1), f(2), f(3), f(3), f(2), f(1), f(0)\}$$

Let $W_{2L}$ denote $\exp(-j2\pi/2L)$, therefore the Discrete Fourier Transform (DFT) of $e_l$ can be given by:

$$E_\mu = \sum_{l=0}^{2L-1} e_l W_{2L}^{l\mu} \tag{12}$$

it can be easily reduced to

$$
\begin{aligned}
E_\mu &= \sum_{l=0}^{L-1} f(l) W_{2L}^{l\mu} + \sum_{l=L}^{2L-1} f(2L - l - 1) W_{2L}^{l\mu} \\
&= \sum_{l=0}^{L-1} f(l) W_{2L}^{l\mu} + \sum_{l=0}^{L-1} f(l) W_{2L}^{(2L-l-1)\mu} \\
&= \sum_{l=0}^{L-1} f(l) [W_{2L}^{l\mu} + W_{2L}^{-(l+1)\mu}], \ \mu = 0, 1, ..., 2L - 1.
\end{aligned}
\tag{13}
$$

If we use a factor of $\frac{1}{2} W_{2L}^{\mu/2}$ to multiply both sides of Equation 13, resulting in

$$\frac{1}{2} W_{2L}^{\mu/2} E_\mu = \sum_{l=0}^{L-1} f(l) \cos[\frac{\pi\mu(2l + 1)}{2L}] \tag{14}$$

We can see that Equation 14 can be approximately Equation 1 of the $L$-point sequence $f(t)$, differing only by the scaling factors. In Equation 12, $E_\mu$ is the $2L$-point DFT of $\{e_l\}$ and Equation 14 indicates that for $\mu = 0, 1, ..., L - 1$, after properly scaled, the transformed sequence $\{E_\mu\}$ can become the Type II DCT of $\{f(l)\}$.

When $\{f(l)\}$ is real and $e_l$ is symmetric, $\{E_\mu\}$ can be computed via two $N$-point FFTs instead of via a single $2N$-point FFT. Given that the computational complexity of an $N$-point FFT algorithm scales as $\mathcal{O}(N log_2 N)$ complex operations, this optimization reduces the $N log_2 N$ FFT operation count by $2N$ complex operations.

## B.1. Actual Efficacy of Timestamps Information

In the introduction, to investigate the actual contribution of timestamp information to iTransformer, we replace its original timestamp-embedded input upsampling module with a single linear layer without timestamp embedding. The performance comparison in Table 4 shows that timestamp information improves prediction performance only on the Traffic dataset, while leading to degradation on all other datasets, suggesting that its effectiveness deserves reconsideration. To explore this, we visualize partial time segments of the top five variables from the 862 variables in the Traffic dataset (see Figure 12 (a)). The results reveal fixed fluctuation patterns in traffic flow at nearly the same periods each day, and importantly, other variables exhibit highly similar variations. This observation may explain why timestamp information benefits iTransformer on Traffic.

However, such characteristics are rare in real-world systems like weather or stock volatility, where variables tend to have more complex dependencies (see Figure 12 (b)).

The frequency-domain representations of the signals inherently provides a global perspective, and the periodic and seasonal characteristics of the signals are effectively represented in its frequency domain components (Zhou et al., 2022). Based on this insight, we propose a frequency-domain masking and resampling method (FMR) that preserves and enhances signal periodicity, thereby mitigating the over-reliance of existing methods on timestamp information for providing additional periodic insights. As shown in Table 4 (also in Table 2 or Table 11), FMR consistently improves performance across almost all datasets, further diminishing the importance of timestamp information.

*Table 4.* Verification of timestamps with four prediction length $F \in \{96, 192, 336, 720\}$ and fixed input $T$=96. All results were reproduced using their released code and identical hyperparameters. "iTrans" is iTransformer, and "R Linear" represents that we replace the input upsampling method within iTransformer with a sigle linear layer without timestamp embedding. **Best**: best results, underline: second results.

| Models | | ETTm1 | | ETTm2 | | ETTh1 | | ETTh2 | | Exchange | | Weather | | ECL | | Solar | | Traffic | |
|---|---|---|---|---|---|---|---|---|---|---|---|---|---|---|---|---|---|---|---|
| Metrics | | MSE | MAE | MSE | MAE | MSE | MAE | MSE | MAE | MSE | MAE | MSE | MAE | MSE | MAE | MSE | MAE | MSE | MAE |
| iTrans original | 96 | 0.342 | 0.377 | 0.186 | 0.272 | 0.387 | 0.405 | 0.301 | **0.350** | 0.086 | 0.206 | 0.181 | **0.221** | 0.148 | 0.239 | 0.201 | 0.234 | **0.392** | **0.268** |
| | 192 | 0.383 | 0.396 | 0.254 | 0.314 | 0.441 | 0.436 | 0.381 | 0.399 | 0.181 | 0.303 | 0.226 | 0.259 | 0.167 | 0.258 | 0.239 | 0.263 | **0.413** | 0.277 |
| | 336 | 0.418 | 0.418 | 0.317 | 0.353 | 0.491 | 0.462 | 0.423 | 0.432 | 0.338 | 0.422 | 0.283 | 0.300 | 0.181 | 0.275 | 0.248 | 0.272 | **0.425** | 0.283 |
| | 720 | 0.487 | 0.456 | 0.416 | 0.408 | 0.509 | 0.494 | 0.430 | 0.446 | 0.869 | 0.704 | 0.359 | 0.351 | **0.209** | **0.299** | 0.250 | 0.275 | 0.459 | 0.300 |
| iTrans R Linear | 96 | 0.347 | 0.377 | 0.184 | 0.267 | 0.383 | 0.401 | 0.303 | 0.352 | 0.085 | 0.205 | 0.183 | 0.223 | 0.147 | 0.239 | 0.201 | **0.233** | 0.396 | 0.270 |
| | 192 | 0.384 | 0.393 | 0.253 | 0.312 | **0.434** | 0.430 | 0.378 | 0.397 | 0.178 | 0.301 | 0.226 | 0.259 | 0.162 | 0.253 | 0.239 | 0.263 | 0.416 | 0.277 |
| | 336 | 0.416 | 0.414 | 0.319 | 0.354 | 0.487 | 0.457 | **0.417** | **0.429** | **0.336** | **0.420** | 0.281 | **0.299** | 0.175 | 0.267 | 0.248 | 0.273 | 0.431 | 0.285 |
| | 720 | 0.483 | 0.451 | **0.414** | **0.406** | 0.496 | 0.483 | **0.424** | **0.444** | 0.842 | 0.692 | **0.356** | **0.347** | 0.211 | 0.301 | 0.249 | 0.275 | 0.465 | 0.302 |
| iTrans + FMR | 96 | **0.340** | **0.373** | **0.183** | **0.265** | **0.382** | **0.398** | **0.299** | **0.350** | **0.084** | **0.204** | **0.180** | 0.222 | **0.141** | **0.235** | **0.199** | 0.237 | 0.393 | **0.268** |
| | 192 | **0.377** | **0.389** | **0.249** | **0.309** | **0.434** | **0.429** | **0.379** | **0.399** | **0.176** | **0.299** | **0.222** | **0.258** | **0.157** | **0.250** | **0.233** | **0.259** | **0.413** | **0.276** |
| | 336 | **0.412** | **0.411** | **0.314** | **0.350** | **0.483** | **0.454** | 0.419 | **0.430** | 0.339 | 0.423 | **0.279** | 0.300 | **0.171** | **0.264** | **0.242** | **0.269** | 0.428 | **0.282** |
| | 720 | **0.481** | **0.450** | 0.418 | 0.409 | **0.492** | **0.480** | 0.426 | 0.445 | **0.834** | **0.690** | **0.356** | 0.350 | 0.233 | 0.316 | **0.244** | **0.273** | **0.458** | **0.299** |

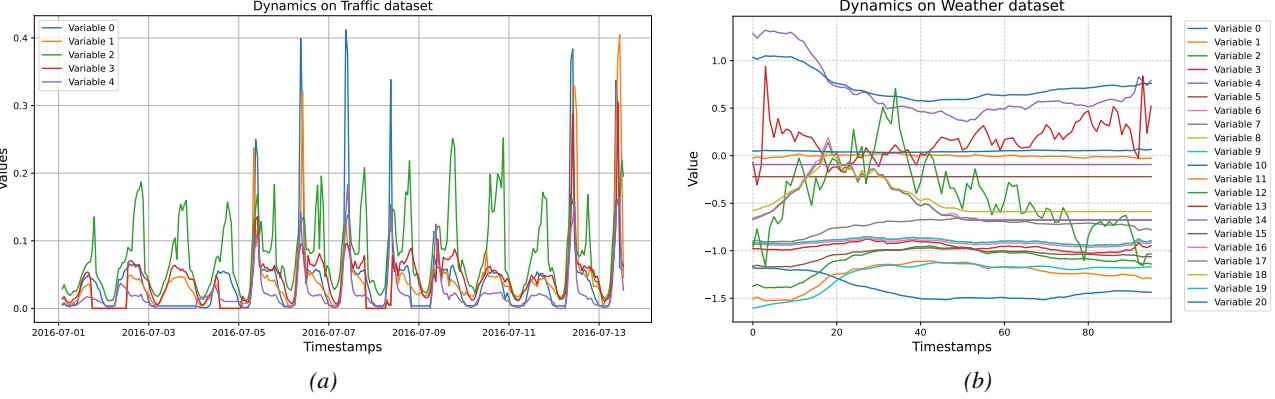

*Figure 12.* Time series trends of different variables in the Traffic (a) and Weather (b) datasets.

## B.2. Visualization of Spectrum

To demonstrate that the proposed FMR preserves signal periodicity and enhances the input signal, we performed a Fourier Transform on a real signal from ETTh1. We then plotted the spectrum of the original signal, the spectrum of the embedding obtained by direct single-linear-layer upsampling of the signal, and the spectrum after processing with the proposed FMR, as shown in Figure 13. Compared to direct linear embedding in the time domain (the commonly adopted approaches in existing methods include embedding techniques that incorporate timestamps), FMR retains more low-frequency information (where the signal's primary information is preserved, as seen in the second subplot) by learning variable-independent masks and performing linear interpolation in the frequency domain. Simultaneously, FMR exhibits a mid-to-high frequency energy distribution closer to that of the real signal, demonstrating better periodicity information retention capabilities than linear embedding directly in time domain.

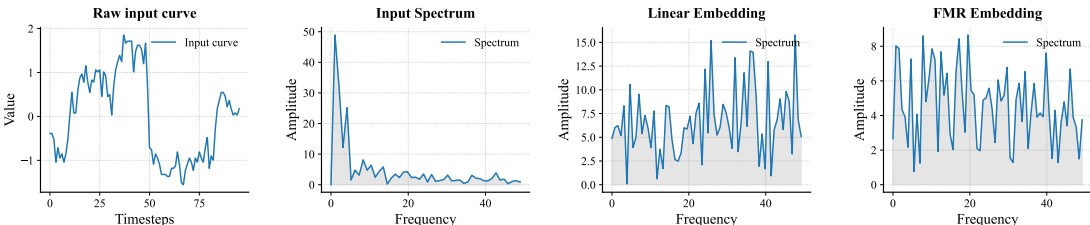

*Figure 13.* Visualization of spetrum as for raw signal and different embedding methods.

As depicted in Figure 14, we visualize the learned masks for variables 0, 2, 3, and 6 within the ETTh1 dataset. A high degree of similarity is observed between the masks for variables 0 and 2, which is consistent with their strong interdependency (PCC = 1.0 in Figure 10). Conversely, the masks for variables 3 and 6, being learned independently, exhibit notable distinctions. Specifically, compared to other variables, the mask for variable 3 suppresses more high-frequency components, which may be because variable 3 exhibits greater volatility and noise. Importantly, we also observe that the masks for all variables predominantly preserve low-frequency components, which contain the signal's periodic and trend information (Zhou et al., 2022). This highlights the ability of our FMR to learn adaptive, variable-specific masks that align with the unique properties of each series.

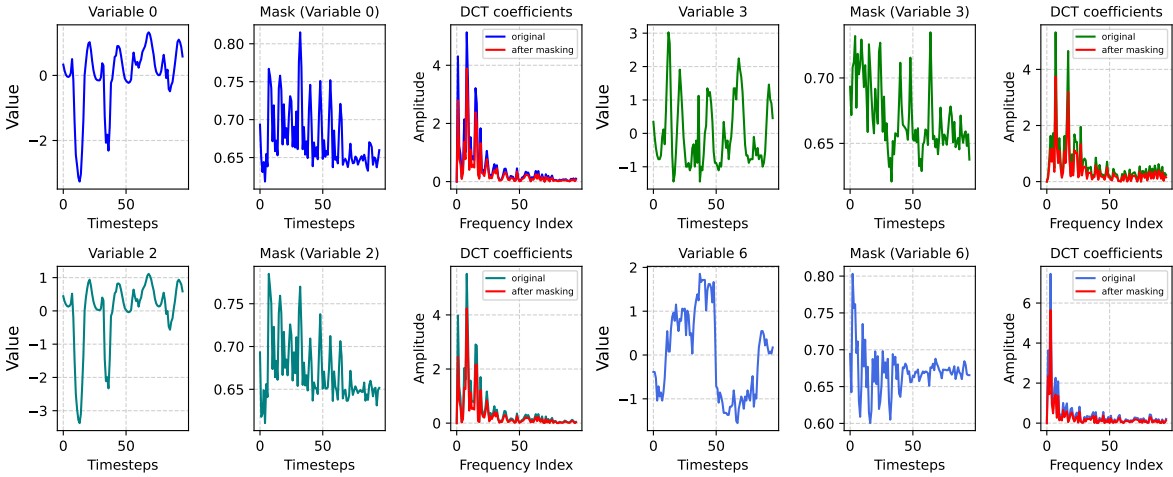

*Figure 14.* Visualization of learned masks on variable 0, 2, 3, and 6 of ETTh1.

## C. Additional Evaluation Metrics

To evaluate the correlations and similarities among variables in multivariate time series, we introduce Dynamic Time Warping (DTW) (Müller, 2007) and Pearson Correlation Coefficient (PCC) (Benesty et al., 2009).

**Dynamic Time Warping.** Dynamic Time Warping (DTW) calculates the similarity between two time series by finding the optimal matching path between them. DTW effectively handles irregularities such as temporal shifts and varying speeds within sequences, demonstrating strong performance in practical problems like speech and gesture recognition. Given two time series $Y = \{y_0, y_1, ..., y_{T-1}\} \in \mathbb{R}^T$ and $\hat{Y} = \{\hat{y}_0, \hat{y}_0, ..., \hat{y}_{T-1}\} \in \mathbb{R}^T$, the DTW distance can be formulated as:

$$\text{DTW}(Y, \hat{Y}) = \min_{\mathbf{A} \in \mathcal{A}(Y, \hat{Y})} \sum_{(i,j) \in \mathbf{A}} d(y_i, \hat{y}_j) = \sum_{(i,j) \in \mathbf{A}^*} d(y_i, \hat{y}_j), \tag{15}$$

Here, $d(\cdot, \cdot)$ represents a distance metric, commonly the squared Euclidean distance. A warping path, denoted by $\mathbf{A}$, comprises $K$ index pairs $\{(i_0, j_0), (i_1, j_1), \ldots, (i_{K-1}, j_{K-1})\}$, with indices $i_k, j_k$ ranging from 0 to $T-1$. The collection of all valid warping paths is given by $\mathcal{A}(Y, \hat{Y})$. The optimal path, $\mathbf{A}^* \in \mathcal{A}(Y, \hat{Y})$, is the one that minimizes the cumulative distance across aligned time steps. A warping path $\mathbf{A}$ is deemed valid if it fulfills the subsequent conditions:

- **Boundary Constraint:** $(i_0, j_0) = (0, 0)$ and $(i_{K-1}, j_{K-1}) = (T-1, T-1)$.

- **Monotonicity Constraint:** The indices must be non-decreasing along the path, specifically $i_{k+1} \geq i_k$ and $j_{k+1} \geq j_k$ for all $k \in [0, K-2]$.

- **Step Size Constraint:** Each step from $(i_k, j_k)$ to $(i_{k+1}, j_{k+1})$ must advance by one unit horizontally, vertically, or diagonally. Formally, $(i_{k+1} - i_k, j_{k+1} - j_k) \in \{(1,0), (0,1), (1,1)\}$, for all $k \in [0, K-2]$.

**Pearson Correlation Coefficient.** Pearson Correlation Coefficient (PCC) evaluates how strongly two variables are linearly related. Given two tokens $Y = \{y_0, y_1, ..., y_{T-1}\} \in \mathbb{R}^T$ and $\hat{Y} = \{\hat{y}_0, \hat{y}_0, ..., \hat{y}_{T-1}\} \in \mathbb{R}^T$ and their mean values $\bar{y}$ and $\bar{\hat{y}}$, PCC can be defined as:

$$\text{PCC}(Y, \hat{Y}) = \frac{\sum_{t=0}^{T-1}(y_t - \bar{y})(\hat{y}_t - \bar{\hat{y}})}{\sqrt{\sum_{t=0}^{T-1}(y_t - \bar{y})^2} \cdot \sqrt{\sum_{t=0}^{T-1}(\hat{y}_t - \bar{\hat{y}})^2}} \tag{16}$$

## D. Transformers Are Fully-connected GNNs

GNNs employ the graph's connective structure to propagate and aggregate information among adjacent nodes. Let $h_i$ denote the node attributes of node $i$. In Graph Attention Networks (GATs) (Veličković et al., 2018), the relationship between the attributes of nodes $i$ and its neighbors $j \in \mathcal{N}_i$ can be computed as:

$$\begin{aligned}
\psi(h_i^l, h_j^l) &= \text{Attention}(W_Q^l h_i^l, \{W_K^l h_j^l, \forall j \in \mathcal{N}_i\}, \{W_V^l h_j^l, \forall j \in \mathcal{N}_i\}), \\
&= \frac{\exp(W_Q^l h_i^l \cdot W_K^l h_j^l)}{\sum_{j' \in \mathcal{N}_i} \exp(W_Q^l h_i^l \cdot W_K^l h_{j'}^l)} \cdot W_V^l h_j^l,
\end{aligned} \tag{17}$$

where $W_Q^l, W_K^l, W_V^l \in \mathbb{R}^{d \times d}$ are learnable weight matrices. The $\psi(h_i^l, h_j^l)$ allows GATs to determine the significance of each neighbor for a given node in the aggregation process. The updated attribute features for node $i$ is derived by combining the information from all of its adjacent nodes:

$$h_i^{l+1} = h_i^l + \sum_{j \in \mathcal{N}_i} \psi(h_i^l, h_j^l), \tag{18}$$

In Variate Transformer, the self-attention captures correlations between all input tokens in MTS input $X$ as follows:

$$\begin{aligned}
\psi(h_i^l, h_j^l) &= \text{Attention}(W_Q^l h_i^l, \{W_K^l h_j^l, \forall j \in X\}, \{W_V^l h_j^l, \forall j \in X\}), \\
&= \frac{\exp(W_Q^l h_i^l \cdot W_K^l h_j^l)}{\sum_{j' \in X} \exp(W_Q^l h_i^l \cdot W_K^l h_{j'}^l)} \cdot W_V^l h_j^l,
\end{aligned} \tag{19}$$

Here, $\psi(h_i^l, h_j^l)$ determines the message between the token pairs $(i, j)$, with each token's relative significance derived through attention mechanism. Subsequently, these weighted messages from all tokens within the $X$ are combined via summation. Then, the token representations for token $i$ are updated using residual connection (He et al., 2016), layer normalization and MLP:

$$h_i^{l+1} = \phi(h_i^l, m_i^l) = \text{MLP}(\text{LayerNorm}(h_i^l + \sum_{j \in X} \psi(h_i^l, h_j^l))). \tag{20}$$

Equation. 17 bears a strong resemblance to the self-attention mechanism within the Transformer. The primary distinction lies in the scope of the aggregation: whereas in GNN the index $j$ is constrained to the local neighborhood of node $i$, in Transformer's self-attention, the aggregation is performed over the entire set of tokens in the sequence. This effectively means the Transformer can be interpreted as a special instance of a GNN operating on a dynamically-weighted, fully-connected graph, where every token is considered a neighbor to all others (Joshi, 2025).

iTransformer presented insightful experiments (see Table 3 in iTransformer paper) where they replaced the FFN with a self-attention layer, essentially constructing a Transformer with two self-attention layers. *The experimental results indicated that simply stacking multiple self-attention layers did not facilitate the learning of correct inter-variate dependencies and temporal patterns.* Therefore, based on the aforementioned analysis, we resort to GNNs for modeling inter-variate dependencies within the deeper layers of the Transformer. Theoretically, GNNs and self-attention layers are closely linked in their ability to capture global relationships. **This insight forms the basis of our novel, theoretically grounded perspective: how to effectively integrate graph-learned dependencies with inter-variate relationships captured by Transformers.**

# E. Why We Use Multi-hop Graph Convolution Network with the Same Graph Structure?

*Table 5.* Comparative Performance of different GNNs in Modeling of Inter-Variable Dependencies in the Network Deep Layer.

| GNN | $F$ | ETTm1 | | ETTh1 | | Weather | | ECL | |
|-----|-----|-------|-------|-------|-------|---------|-------|-------|-------|
| | | MSE | MAE | MSE | MAE | MSE | MAE | MSE | MAE |
| GCN | 96 | **0.315** | **0.344** | **0.372** | **0.387** | 0.152 | 0.190 | **0.137** | **0.227** |
| | 192 | **0.366** | **0.372** | **0.424** | **0.418** | **0.203** | **0.239** | **0.155** | **0.243** |
| | 336 | **0.398** | **0.395** | 0.473 | 0.443 | **0.257** | **0.279** | **0.170** | **0.259** |
| | 720 | 0.472 | 0.435 | 0.473 | 0.464 | **0.338** | **0.334** | **0.198** | **0.283** |
| GAT | 96 | 0.321 | 0.348 | 0.375 | 0.388 | **0.150** | **0.190** | 0.141 | 0.231 |
| | 192 | 0.370 | 0.374 | 0.428 | 0.420 | 0.209 | 0.242 | 0.165 | 0.247 |
| | 336 | 0.405 | 0.399 | **0.469** | **0.441** | 0.262 | 0.281 | 0.175 | 0.262 |
| | 720 | **0.465** | **0.432** | **0.471** | **0.463** | 0.344 | 0.337 | 0.213 | 0.289 |

We reformulate Equation (18) as Equation (21). We note that the key distinction between GAT and GCN lies in their adjacency matrix weights: in GAT, the weights of the adjacency matrix are learned dynamically and vary across each layer $l$ (a.k.a., each hop in multi-hop GNN), whereas a standard GCN employs a fixed adjacency matrix for feature propagation.

$$\mathbf{H}^{l+1} = \mathbf{H}^l + \sigma(\mathbf{A}^l \mathbf{H}^l \mathbf{W}^l), \tag{21}$$

where $\sigma$ is the activation function, $H$ and $W$ are node features and learnable weights, respectively. In our `CGTFra`, graph structures are dynamically learned from global inputs via linear transformations and gating mechanisms, rather than being predetermined. Consequently, the typical distinctions between GCN and GAT in terms of their edge weights fixed or varying across different hops are attenuated. As presented in Table 5, we compare the performance difference within the `CGTFra` framework when using either identical or distinct graph weights for information aggregation at each hop in DGL (Essentially, based on input-constructed graph structures, we implement standard GCN and GAT). Notably, when each hop employs a dynamically relearned graph structure based on its current input, we apply a consistency constraint (CAL) to the graph structure of the final hop. The results indicate that using dynamically updated edge weights at each hop does not yield significant performance gains. We attribute this to the fact that **shallow self-attention layers capture inter-variate dependencies based on global tokens, learning association weights only once**. Although GNNs in DGL employ multi-hop strategies to aggregate information from broader nodes, the graph structure proposed is also dynamically learned from the global input tokens (i.e., the output of the self-attention layer). Therefore, utilizing the same graph structure across all hops is more conducive to subsequent consistent alignment of inter-variate dependencies. **Therefore, the DGL within the proposed `CGTFra` framework employs consistent adjacency matrix weights across all hops. Furthermore, for different layers ($L$ in Figure 4) of `CGTFra`, the graph structure in DGL is distinct (input-dependent), which aligns with the re-computation of attention scores in each self-attention layer.**

# F. Theoretical Guarantees of CAL from Information Bottleneck Principle

The Information Bottleneck (IB) principle (Tishby et al., 2000) aims to find a compressed representation, denoted as $Z$, that maximally preserves information about a target variable $Y$ while simultaneously compressing the input $X$. This objective is typically formulated as the following optimization problem:

$$\max I(Z; Y) - \beta \cdot I(Z; X), \tag{22}$$

where $I(\cdot; \cdot)$ represents mutual information and $\beta$ is a Lagrange multiplier. Within our `CGTFra` framework, we can interpret the self-attention map (MCM) as a high-bandwidth, yet potentially noisy, representation of the inter-variable relationships in the input $X$. While its mutual information with the input, $I(\text{MCM}; X)$, is high, much of this information may constitute **noise** irrelevant to the final prediction target $Y$. Conversely, the GNN's adjacency matrix, $A$, is intended to be the compressed and cleaner representation $Z$ that we seek to learn. The goal for $A$ is to **discard the noise** present in MCM and **retain only the structured information** pertinent to predicting $Y$. In this context, our alignment loss, $\mathcal{L}_{align} = \text{KL}(\text{MCM}||A)$, can be viewed as a proxy or an upper bound for the compression term, $I(Z; X)$, in the IB objective. By minimizing $\text{KL}(\text{MCM}||A)$, we encourage the learned adjacency matrix $A$ not to deviate excessively from the attention map MCM. This implicitly controls the mutual information $I(A; \text{MCM})$, and by extension, $I(A; X)$, aligning our method with the core IB principle of learning a compressed yet informative representation. We provide the theoretical derivation as follows.

Since the entropy term $H(Y)$ depends solely on the data distribution and remains constant with respect to the optimization process, we have:

$$I(Z;Y) = H(Y) - H(Y|Z), \tag{23}$$

$$\max I(Z;Y) \equiv \min H(Y|Z) \tag{24}$$

The conditional entropy $H(Y|Z)$ can be approximated by the upper bound of the negative log-likelihood. Let us assume a decoder (or prediction head) parameterized by $\theta$ is used to predict $Y$, denoted as the distribution $q_\theta(y|z)$. Then:

$$H(Y|Z) = \mathbb{E}_{p(y,z)}[-\log p(y|z)] \approx \mathbb{E}_{p(y,z)}[-\log q_\theta(y|z)] \tag{25}$$

We postulate that the conditional probability distribution $q_\theta(y|z)$ of the target variable $Y$ follows a Laplace Distribution. Its location parameter (mean) corresponds to the model's prediction $\hat{y} = f_\theta(z)$, and $b$ denotes the scale parameter. Consequently, minimizing the negative log-likelihood is equivalent to minimizing the Mean Absolute Error (MAE):

$$
\begin{aligned}
-\log q_\theta(y|z) &= -\log\left(\frac{1}{2b}\exp\left(-\frac{|y-\hat{y}|}{b}\right)\right) \\
&= \log(2b) + \frac{1}{b}|y-\hat{y}|
\end{aligned}
\tag{26}
$$

$$\min_\theta\left[-\log q_\theta(y|z)\right] \equiv \min_\theta \|y-\hat{y}\|_1 = \mathcal{L}_{MAE} \tag{27}$$

In the following, we start the derivation with the mutual information term $I(Z;X)$. By definition:

$$
\begin{aligned}
I(Z;X) &= \mathbb{E}_{p(x,z)}\left[\log\frac{p(z|x)}{p(z)}\right] \\
&= \mathbb{E}_{p(x)}\left[\int p(z|x)\log\frac{p(z|x)}{p(z)}dz\right]
\end{aligned}
\tag{28}
$$

Since the true marginal distribution $p(z) = \int p(z|x)p(x)dx$ is intractable, we introduce a variational approximation distribution $q_{\text{gnn}}(z)$ (parameterized by the GNN's adjacency matrix $A$) to approximate $p(z)$. Using an identity transformation, we multiply and divide by $q_{\text{gnn}}(z)$ simultaneously inside the logarithmic term:

$$
\begin{aligned}
I(Z;X) &= \mathbb{E}_{p(x)}\left[\int p(z|x)\log\frac{p(z|x)\cdot q_{\text{gnn}}(z)}{p(z)\cdot q_{\text{gnn}}(z)}dz\right] \\
&= \mathbb{E}_{p(x)}\left[\int p(z|x)\log\frac{p(z|x)}{q_{\text{gnn}}(z)}dz + \int p(z|x)\log\frac{q_{\text{gnn}}(z)}{p(z)}dz\right] \\
&= \mathbb{E}_{p(x)}\left[D_{\text{KL}}(p(z|x)||q_{\text{gnn}}(z))\right] - \int\left(\int p(x)p(z|x)\,dx\right)\log\frac{p(z)}{q_{\text{gnn}}(z)}\,dz \\
&= \mathbb{E}_{p(x)}\left[D_{\text{KL}}(p(z|x)||q_{\text{gnn}}(z))\right] - \int p(z)\log\frac{p(z)}{q_{\text{gnn}}(z)}\,dz \\
&= \mathbb{E}_{p(x)}\left[D_{\text{KL}}(p(z|x)||q_{\text{gnn}}(z))\right] - D_{\text{KL}}(p(z)||q_{\text{gnn}}(z))
\end{aligned}
\tag{29}
$$

The equation above can be decomposed into two terms:

$$I(Z;X) = \underbrace{\mathbb{E}_{p(x)}\left[D_{\text{KL}}(p(z|x)||q_{\text{gnn}}(z))\right]}_{\text{Term 1}} - \underbrace{D_{\text{KL}}(p(z)||q_{\text{gnn}}(z))}_{\text{Term 2}} \tag{30}$$

where:

- **Term 1** represents the expectation of the KL divergence between the encoder distribution $p(z|x)$ (i.e., MCM) and the variational distribution $q_{\text{gnn}}(z)$ (i.e., $A$).

- **Term 2** is the KL divergence between the true marginal distribution and the variational distribution. Based on the non-negativity of KL divergence (Gibbs' inequality), we know that $D_{\text{KL}}(p(z)||q_{\text{gnn}}(z)) \geq 0$.

Therefore, by omitting the non-negative Term 2, we derive a tractable upper bound for the mutual information $I(Z; X)$:

$$I(Z; X) \leq \mathbb{E}_{p(x)}[D_{KL}(p(z|x)||q_{\text{gnn}}(z))] \tag{31}$$

We instantiate the variational distributions using the model components:

$$\begin{aligned} p(z|x) &:= \text{MCM} \\ q_{\text{gnn}}(z) &:= A \end{aligned} \tag{32}$$

Consequently, minimizing this upper bound is equivalent to minimizing our proposed alignment loss:

$$\mathcal{L}_{align} = D_{KL}(\text{MCM}||A) \tag{33}$$

In summary, the total loss function in Equation (10) serves as a direct implementation of the Variational Information Bottleneck (VIB) objective:

$$\mathcal{L}_{total} = \underbrace{\mathcal{L}_{MAE}}_{\text{Maximizes } I(Z;Y)} + \lambda \cdot \underbrace{D_{KL}(\text{MCM}||A)}_{\text{Upper bound of } I(Z;X)} \tag{34}$$

This provides a theoretical justification that incorporating the KL divergence constraint is not merely a regularization technique; rather, it is designed to learn a robust dependency graph that simultaneously ensures accurate future prediction (**Information Preservation**) and filters out input noise (**Information Compression**).

## G. Existing Transformer-based Methods Modeling IVD

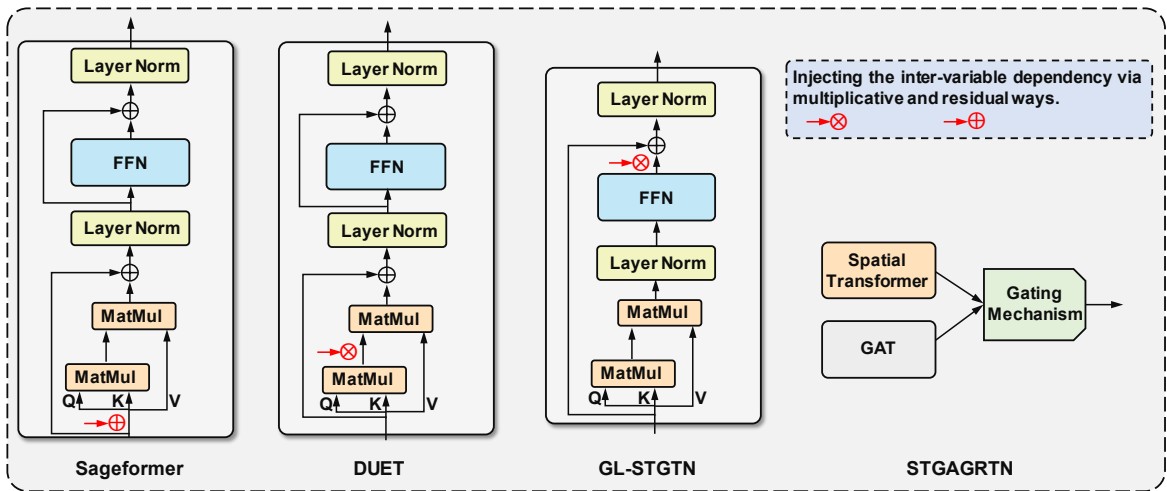

*Figure 15.* Typical Transformer-based approaches to modeling inter-variate dependencies.

We analyze four representative Transformer-based approaches for modeling inter-variable dependencies, namely Sageformer (Zhang et al., 2024), DUET (Qiu et al., 2025), GL-STGTN (Li et al., 2024), and STGAGRTN (Wu et al., 2023a). As illustrated in Figure 15, these methods embed inter-variable dependencies primarily by incorporating them as masks or biases within the Transformer, which we categorize as Figure 2(b).

- Sageformer (Zhang et al., 2024): SageFormer first employs a GNN (with totally self-learned graph structure) to capture inter-variate correlations from the input MTS. The resulting global, graph-enhanced embeddings are then fused with the original series to serve as the input for a **vanilla Transformer** (i.e., temporal Transformer), which subsequently models temporal dependencies.

- DUET (Qiu et al., 2025): DUET captures IVD in the frequency domain using metric learning. The resulting dependency is then integrated into the self-attention mechanism as a mask for the attention scores of **Variate Transformer**.

- GL-STGTN (Li et al., 2024): GL-STGTN learns the graph structure from both global and local perspectives, and then the learned inter-variable dependencies are then encoded into a spatial attention mechanism.

- STGAGRTN (Wu et al., 2023a): STGAGRTN utilizes a gating mechanism to fuse the inter-variable dependencies learned separately by a GAT and a proposed spatial Transformer.

However, such approaches do not adequately address the challenge of modeling inter-variable dependencies in deeper layers. Although GL-STGTN introduces inter-variable relations after the feed-forward network (FFN), the additional branch is prone to capturing spurious correlations. More importantly, unlike our work, **GL-STGTN does not explore and account for the consistency between shallow- and deep-layer modeling of inter-variable dependencies.**

Unlike GL-STGTN with graph learning, which uses the **raw input** $X$, or methods like Sageformer and MSGNet that rely **solely on self-learned node embeddings**, our graph constructor is uniquely informed by a combination of **outputs from the self-attention layer** and **learnable node embeddings**. Importantly, to the best of our knowledge, we are the first to comprehensively analyze the connections and differences between self-attention (within Variate Transformers) and GNNs for modeling IVD. Furthermore, as we emphasize earlier, we have compactly integrated the two linear layers of the original FFN into our DGL module. It is this compact architecture that allows our DGL to serve as a general-purpose IVD modeling method for the deeper layers of Variate Transformers—**a level of universality not achieved by existing dynamic graph learning techniques**. Crucially, the introduction of DGL and CAL significantly accelerates the convergence of both training and validation losses, achieving an 8.78% reduction in MSE of iTransformer (see Figure 5). **This substantial performance gain is achieved with only a minimal computational overhead—time complexity of $\mathcal{O}(N(D + nd + D * nd))$ (see Equation (7)), which is linear with respect to the number of variables $N$, where $nd$ is a small hyperparameter (e.g., 8, 10, or 32), and $D$ is the hidden dimension.**

Grounded in Information Bottleneck principle, by integrating DGL and CAL, we propel **strong, superior, and well-known baseline models to establish a new state-of-the-art benchmark without altering their underlying architectures or running hyperparameters, demonstrating strong potential to serve as a versatile forecasting framework applicable across diverse domains.**

## H. Dataset Details

As shown in Table 6, total 13 datasets utilized in our study encompass data from five domains: Temperature, Finance, Weather, Electricity, and Transportation, providing a comprehensive assessment of a model's effectiveness and generality. **Forecastability** is computed by one minus the entropy of Fourier decomposition, a lower value indicating worse predictability.

*Table 6.* Details of different datasets.

| Datasets | # Variables | # Samples* | Frequency | Forecastability | Information |
|---|---|---|---|---|---|
| ETTm1 | 7 | (34465, 11521, 11521) | 15min | 0.46 | Temperature |
| ETTm2 | 7 | (34465, 11521, 11521) | 15min | 0.55 | Temperature |
| ETTh1 | 7 | (8545, 2881, 2881) | 1hour | 0.38 | Temperature |
| ETTh2 | 7 | (8545, 2881, 2881) | 1hour | 0.45 | Temperature |
| Exchange | 8 | (5120, 665, 1422) | Daily | - | Finance |
| Weather | 21 | (36792, 5271, 10540) | 10min | 0.75 | Weather |
| Solar-Energy | 137 | (36601, 5161, 10417) | 10min | 0.33 | Electricity |
| Electricity | 321 | (18317, 2633, 5261) | 1hour | 0.77 | Electricity |
| Traffic | 862 | (12185, 1757, 3509) | 1hour | 0.68 | Transportation |
| PEMS03 | 358 | (15701, 5216, 434) | 5min | 0.65 | Transportation |
| PEMS04 | 307 | (10172, 3375, 281) | 5min | 0.45 | Transportation |
| PEMS07 | 883 | (16911, 5622, 468) | 5min | 0.58 | Transportation |
| PEMS08 | 170 | (10690, 3548, 265) | 5min | 0.52 | Transportation |

* The number of samples indicates the split ratio (Train, Val, Test).

---

**Algorithm 1** Optimization process of Consistent Graph Transformer Framework CGTFra)

**Input:** MTS dataset $\mathcal{D} = \{(\mathbf{X}, \mathbf{Y})\}$; encoder backbone with $L$ layers; FMR module with learnable mask $\mathcal{F}_{mask}$; Self-Attention layer $\mathcal{T}$ with **MCM**; DGL layer with hop $i$, $\Theta_{\mathcal{G}}$, $\mathbf{A}$; alignment weight $\lambda$; learning rate $\eta$.

**Output:** Well-trained parameters $\Theta$ of CGTFra.

1: **for** each mini-batch $(\mathbf{X}, \mathbf{Y}) \in \mathcal{D}$ **do**
2:    **Stage 1: Adaptive Frequency Enhancement**
3:    $\mathbf{F}_{freq} \leftarrow \text{DCT}(\mathbf{X})$                                               ▷ Transform to frequency domain
4:    $\hat{\mathbf{F}} \leftarrow \text{Resampling}(\mathcal{F}_{mask}(\mathbf{F}_{freq}))$                            ▷ Frequency masking & resampling
5:    $\mathbf{H}^{(0)} \leftarrow \text{iDCT}(\hat{\mathbf{F}})$                                                   ▷ Inverse transform
6:    **Stage 2: Consistent Representation Learning**
7:    Initialize alignment loss $\mathcal{L}_{align} \leftarrow 0$
8:    **for** $l = 1$ to $L$ **do**
9:       *# Multivariate Correlation (Noisy Prior)*
10:      $\mathbf{H}_{sa}^{(l)}, \mathbf{MCM}_{attn}^{(l)} \leftarrow \mathcal{T}(\mathbf{H}^{(l-1)})$                          ▷ Derive attention map $p(z|x)$
11:      *# Dynamic Graph Structure (Variational Posterior)*
12:      $\mathbf{A}^{(l)} \leftarrow \text{GraphConstructor}(\mathbf{H}_{sa}^{(l)}, \Theta_{\mathcal{G}})$                   ▷ Learn structure $q_{gnn}(z)$
13:      *# Multi-hop Information Aggregation*
14:      $\mathbf{H}_i^{(l)} \leftarrow \text{MultiHopGCN}(\mathbf{H}_{sa}^{(l)}, \mathbf{A}^{(l)}, i)$
15:      $\mathbf{H}^{(l)} \leftarrow \text{MLP}(\text{Concat}(\mathbf{H}_i^{(l)}))$
16:      *# Constraint: Information Bottleneck Alignment (CAL)*
17:      $\mathcal{L}_{align}^{(l)} \leftarrow \text{KL}(\mathbf{MCM}_{attn}^{(l)} \| \mathbf{A}^{(l)})$                            ▷ Minimize mutual info $I(Z; X)$
18:      $\mathcal{L}_{align} \leftarrow \mathcal{L}_{align} + \mathcal{L}_{align}^{(l)}$
19:    **end for**
20:    **Stage 3: Optimization Objective**
21:    $\hat{\mathbf{Y}} \leftarrow \text{Projection}(\mathbf{H}^{(L)})$                                       ▷ Final prediction
22:    $\mathcal{L}_{MAE} \leftarrow \frac{1}{T} \sum \|\hat{\mathbf{Y}} - \mathbf{Y}\|_1$                       ▷ Maximize predictive info $I(Z; Y)$
23:    $\mathcal{L}_{total} \leftarrow \mathcal{L}_{MAE} + \lambda \mathcal{L}_{align}$                        ▷ Joint optimization objective
24:    $\Theta \leftarrow \text{Adam}(\Theta, \nabla_\Theta \mathcal{L}^*, \eta)$                          ▷ Update all trainable parameters
25: **end for**

---

## I. Implementation Details

All experiments are implemented in PyTorch 2.0.1 with Python 3.8 on two NVIDIA GeForce RTX 3090 GPUs. We use Adam optimizer with $\mathcal{L} = \mathcal{L}_{MAE} + \lambda \mathcal{L}_{align}$ as the loss function for model optimization and evaluate the prediction performance with the Mean Squared Error: $\text{MSE} = \frac{1}{n} \sum_{i=1}^{n} (y_i - \hat{y}_i)^2$ and MAE, where $y_i$ and $\hat{y}_i$ represent the ground truth and predicted value at time $i$, respectively. The optimization process of CGTFra is detailed in Algorithm 1.

By default, we employ Kullback-Leibler (KL) divergence as the $\mathcal{L}_{align}$. And the number of stacked layers for CGTFra is selected from 1, 2, or 4, with 1 or 2 layers typically used for datasets with fewer variables, and 2 or 4 layers for those with more variables. DGL's default number of hops is 2, and the batch size is set between 16 and 128. Hyperparameter sensitivity analysis is provided in Appendix M.

## J. Additional Results

In this section, we present the complete comparison results for both long-term and short-term forecasting, as shown in Table 7 and Table 8, respectively. To further compare model performance under longer input horizons, we also provide results with an input length of 336 in Table 9. Across short-term forecasting, long-term forecasting, and extended input lengths, CGTFra consistently demonstrates superior predictive performance, underscoring its overall effectiveness.

Theoretically, a longer given historical input enables models to capture more information, leading to more accurate predictions. However, higher dimensionality can also introduce side effects such as model overfitting and training difficulty. To investigate the performance differences of various models across different historical input lengths, as illustrated in Figure 16, we evaluate the performance of five methods. We observed that all models achieved relatively comparable prediction accuracy when the given input length is 336. Further increasing the input length, however, potentially led to a decline in

*Table 7.* Long-term forecasting results with forecasting horizons $F \in \{96, 192, 336, 720\}$ and fixed look-back length $T=96$. **Bold**/underline: Best/second best one. "-" indicates that the original method was not evaluated in the corresponding scenario.

| Models | | CGTFra (*Ours*) | | DUET (KDD'25) | | TimePro (ICML'25) | | Soatten (AAAI'25) | | VCformer (IJCAI'24) | | FilterNet (NeurIPS'24) | | iTransformer (ICLR'24) | | MSGNet (AAAI'24) | | PatchTST (ICLR'23) | |
|---|---|---|---|---|---|---|---|---|---|---|---|---|---|---|---|---|---|---|---|
| Metrics | | MSE | MAE | MSE | MAE | MSE | MAE | MSE | MAE | MSE | MAE | MSE | MAE | MSE | MAE | MSE | MAE | MSE | MAE |
| ETTm1 | 96 | **0.315** | **0.344** | 0.324 | 0.354 | 0.326 | 0.364 | 0.329 | 0.365 | 0.319 | 0.359 | 0.318 | 0.358 | 0.334 | 0.368 | 0.319 | 0.366 | 0.329 | 0.367 |
| | 192 | 0.366 | **0.372** | 0.369 | 0.379 | 0.367 | 0.383 | 0.37 | 0.387 | **0.364** | 0.382 | **0.364** | 0.383 | 0.377 | 0.391 | 0.376 | 0.397 | 0.367 | 0.385 |
| | 336 | 0.398 | **0.395** | 0.404 | 0.402 | 0.402 | 0.409 | 0.401 | 0.407 | 0.399 | 0.405 | **0.396** | 0.406 | 0.426 | 0.420 | 0.417 | 0.422 | 0.399 | 0.410 |
| | 720 | 0.472 | **0.435** | 0.463 | 0.437 | 0.469 | 0.446 | 0.474 | 0.447 | 0.467 | 0.442 | 0.456 | 0.444 | 0.491 | 0.459 | 0.481 | 0.458 | **0.454** | 0.439 |
| ETTm2 | 96 | **0.171** | **0.249** | 0.174 | 0.255 | 0.178 | 0.260 | 0.180 | 0.264 | 0.180 | 0.266 | 0.174 | 0.257 | 0.180 | 0.264 | 0.177 | 0.262 | 0.175 | 0.259 |
| | 192 | **0.238** | **0.293** | 0.243 | 0.302 | 0.242 | 0.303 | 0.245 | 0.306 | 0.245 | 0.306 | 0.240 | 0.300 | 0.250 | 0.309 | 0.247 | 0.307 | 0.241 | 0.302 |
| | 336 | 0.300 | **0.333** | 0.304 | 0.341 | 0.303 | 0.342 | 0.312 | 0.349 | 0.307 | 0.345 | **0.297** | 0.339 | 0.311 | 0.348 | 0.312 | 0.346 | 0.305 | 0.343 |
| | 720 | 0.397 | **0.391** | 0.399 | 0.397 | 0.400 | 0.399 | 0.411 | 0.406 | 0.406 | 0.402 | **0.392** | 0.393 | 0.412 | 0.407 | 0.414 | 0.403 | 0.402 | 0.400 |
| ETTh1 | 96 | **0.372** | **0.387** | 0.377 | 0.393 | 0.375 | 0.398 | 0.383 | 0.400 | 0.376 | 0.397 | 0.375 | 0.394 | 0.386 | 0.405 | 0.390 | 0.411 | 0.414 | 0.419 |
| | 192 | **0.425** | **0.418** | 0.429 | 0.425 | 0.427 | 0.429 | 0.440 | 0.433 | 0.431 | 0.427 | 0.436 | 0.422 | 0.441 | 0.436 | 0.442 | 0.442 | 0.460 | 0.445 |
| | 336 | **0.470** | **0.443** | 0.471 | 0.446 | 0.472 | 0.450 | 0.475 | 0.449 | 0.473 | 0.449 | 0.476 | 0.443 | 0.487 | 0.458 | 0.480 | 0.468 | 0.501 | 0.466 |
| | 720 | **0.469** | **0.462** | 0.496 | 0.480 | 0.476 | 0.474 | 0.491 | 0.477 | 0.476 | 0.474 | 0.474 | 0.469 | 0.503 | 0.491 | 0.494 | 0.488 | 0.500 | 0.488 |
| ETTh2 | 96 | **0.288** | **0.336** | 0.296 | 0.345 | 0.293 | 0.345 | 0.295 | 0.348 | 0.292 | 0.344 | 0.292 | 0.343 | 0.297 | 0.349 | 0.328 | 0.371 | 0.302 | 0.348 |
| | 192 | **0.364** | **0.384** | 0.368 | 0.389 | 0.367 | 0.394 | 0.380 | 0.398 | 0.377 | 0.396 | 0.369 | 0.395 | 0.380 | 0.400 | 0.402 | 0.414 | 0.388 | 0.400 |
| | 336 | **0.410** | **0.422** | 0.411 | **0.422** | 0.419 | 0.431 | 0.420 | 0.431 | 0.417 | 0.430 | 0.420 | 0.432 | 0.428 | 0.432 | 0.435 | 0.443 | 0.426 | 0.433 |
| | 720 | 0.414 | **0.433** | **0.412** | 0.434 | 0.427 | 0.445 | 0.419 | 0.441 | 0.423 | 0.443 | 0.430 | 0.446 | 0.427 | 0.445 | 0.417 | 0.441 | 0.431 | 0.446 |
| Exchange | 96 | **0.083** | **0.202** | 0.086 | 0.205 | 0.085 | 0.204 | 0.085 | 0.204 | 0.085 | 0.205 | **0.083** | **0.202** | 0.086 | 0.206 | 0.102 | 0.23 | 0.088 | 0.205 |
| | 192 | **0.173** | **0.296** | 0.182 | 0.305 | 0.178 | 0.299 | 0.175 | 0.299 | 0.176 | 0.299 | 0.174 | **0.296** | 0.177 | 0.299 | 0.195 | 0.317 | 0.176 | 0.299 |
| | 336 | 0.324 | 0.412 | 0.310 | 0.403 | 0.328 | 0.414 | 0.330 | 0.417 | 0.328 | 0.415 | 0.326 | 0.413 | 0.331 | 0.417 | 0.359 | 0.436 | **0.301** | **0.397** |
| | 720 | **0.668** | **0.619** | 0.693 | 0.624 | 0.817 | 0.679 | 0.844 | 0.695 | 0.830 | 0.688 | 0.840 | 0.670 | 0.847 | 0.691 | 0.940 | 0.738 | 0.901 | 0.714 |
| Weather | 96 | **0.152** | **0.190** | 0.163 | 0.202 | 0.166 | 0.207 | 0.161 | 0.206 | 0.171 | 0.220 | 0.162 | 0.207 | 0.174 | 0.214 | 0.163 | 0.212 | 0.177 | 0.218 |
| | 192 | **0.203** | **0.239** | 0.218 | 0.252 | 0.216 | 0.254 | 0.208 | 0.250 | 0.230 | 0.266 | 0.210 | 0.250 | 0.221 | 0.254 | 0.212 | 0.254 | 0.225 | 0.259 |
| | 336 | **0.257** | **0.279** | 0.274 | 0.294 | 0.273 | 0.296 | 0.264 | 0.291 | 0.280 | 0.299 | 0.265 | 0.290 | 0.278 | 0.296 | 0.272 | 0.299 | 0.278 | 0.297 |
| | 720 | **0.338** | **0.334** | 0.349 | 0.343 | 0.351 | 0.346 | 0.347 | 0.346 | 0.352 | 0.344 | 0.342 | 0.340 | 0.358 | 0.347 | 0.350 | 0.348 | 0.354 | 0.348 |
| Electricity | 96 | **0.137** | **0.227** | 0.145 | 0.233 | 0.139 | 0.234 | **0.137** | 0.232 | 0.150 | 0.242 | 0.147 | 0.245 | 0.148 | 0.240 | 0.165 | 0.274 | 0.181 | 0.270 |
| | 192 | **0.155** | **0.243** | 0.163 | 0.248 | 0.156 | 0.249 | **0.155** | 0.247 | 0.167 | 0.255 | 0.160 | 0.250 | 0.162 | 0.253 | 0.184 | 0.292 | 0.188 | 0.274 |
| | 336 | **0.170** | **0.259** | 0.175 | 0.262 | 0.172 | 0.267 | 0.171 | 0.265 | 0.182 | 0.270 | 0.173 | 0.267 | 0.178 | 0.269 | 0.195 | 0.302 | 0.204 | 0.293 |
| | 720 | **0.198** | **0.283** | 0.204 | 0.291 | 0.209 | 0.299 | 0.200 | 0.290 | 0.221 | 0.302 | 0.210 | 0.309 | 0.225 | 0.317 | 0.231 | 0.332 | 0.246 | 0.324 |
| Solar | 96 | **0.191** | **0.205** | 0.200 | 0.207 | 0.196 | 0.237 | 0.198 | 0.239 | - | - | - | - | 0.203 | 0.237 | - | - | 0.234 | 0.286 |
| | 192 | **0.218** | **0.225** | 0.228 | 0.233 | 0.231 | 0.263 | 0.228 | 0.259 | - | - | - | - | 0.233 | 0.261 | - | - | 0.267 | 0.310 |
| | 336 | **0.238** | **0.240** | 0.262 | 0.244 | 0.250 | 0.281 | 0.244 | 0.272 | - | - | - | - | 0.248 | 0.273 | - | - | 0.290 | 0.315 |
| | 720 | 0.249 | **0.242** | 0.258 | 0.249 | 0.253 | 0.285 | **0.246** | 0.275 | - | - | - | - | 0.249 | 0.275 | - | - | 0.289 | 0.317 |
| Traffic | 96 | **0.387** | **0.239** | 0.407 | 0.252 | - | - | 0.401 | 0.270 | 0.454 | 0.310 | 0.430 | 0.294 | 0.395 | 0.268 | - | - | 0.544 | 0.359 |
| | 192 | **0.417** | **0.249** | 0.431 | 0.262 | - | - | 0.424 | 0.281 | 0.468 | 0.315 | 0.452 | 0.307 | **0.417** | 0.276 | - | - | 0.540 | 0.354 |
| | 336 | 0.434 | **0.261** | 0.456 | 0.269 | - | - | 0.445 | 0.288 | 0.486 | 0.325 | 0.470 | 0.316 | **0.433** | 0.283 | - | - | 0.551 | 0.358 |
| | 720 | 0.472 | **0.279** | 0.509 | 0.292 | - | - | 0.479 | 0.306 | 0.524 | 0.348 | 0.498 | 0.323 | **0.467** | 0.302 | - | - | 0.586 | 0.375 |
| 1st Count | | **26** | **35** | 1 | 1 | 0 | 0 | 3 | 0 | 1 | 0 | 5 | 3 | 3 | 0 | 0 | 0 | 2 | 1 |

performance. Consequently, as shown in Table 9, we also conducted a detailed comparison of how different models perform when predicting four distinct output lengths, with an input length of 336.

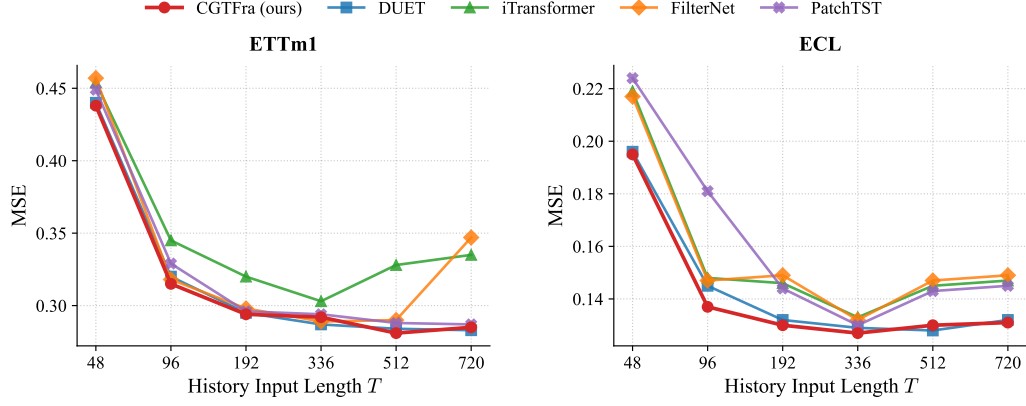

*Figure 16.* Performance comparison with different historical input lengths (Predict $F$=96).

*Table 8.* Short-term forecasting results with forecasting horizons $F \in \{12, 24, 48, 96\}$ and fixed look-back length $T$=96.

| Models | | CGTFra (Ours) | | iTransformer (ICLR'24) | | RLinear (ArXiv'23) | | PatchTST (ICLR'23) | | Crossformer (ICLR'23) | | TiDE (TMLR'23) | | TimesNet (ICLR'23) | | DLinear (AAAI'23) | |
|---|---|---|---|---|---|---|---|---|---|---|---|---|---|---|---|---|---|
| Metrics | | MSE | MAE | MSE | MAE | MSE | MAE | MSE | MAE | MSE | MAE | MSE | MAE | MSE | MAE | MSE | MAE |
| PEMS03 | 12 | **0.060** | **0.159** | 0.071 | 0.174 | 0.126 | 0.236 | 0.099 | 0.216 | 0.090 | 0.203 | 0.178 | 0.305 | 0.085 | 0.192 | 0.122 | 0.243 |
| | 24 | **0.079** | **0.184** | 0.093 | 0.201 | 0.246 | 0.334 | 0.142 | 0.259 | 0.121 | 0.240 | 0.257 | 0.371 | 0.118 | 0.223 | 0.201 | 0.317 |
| | 48 | **0.119** | **0.228** | 0.125 | 0.236 | 0.551 | 0.529 | 0.211 | 0.319 | 0.202 | 0.317 | 0.379 | 0.463 | 0.155 | 0.260 | 0.333 | 0.425 |
| | 96 | 0.173 | 0.278 | **0.164** | **0.275** | 1.057 | 0.787 | 0.269 | 0.370 | 0.262 | 0.367 | 0.490 | 0.539 | 0.228 | 0.317 | 0.457 | 0.515 |
| PEMS04 | 12 | **0.070** | **0.169** | 0.078 | 0.183 | 0.138 | 0.252 | 0.105 | 0.224 | 0.098 | 0.218 | 0.219 | 0.340 | 0.087 | 0.195 | 0.148 | 0.272 |
| | 24 | **0.084** | **0.187** | 0.095 | 0.205 | 0.258 | 0.348 | 0.153 | 0.275 | 0.131 | 0.256 | 0.292 | 0.398 | 0.103 | 0.215 | 0.224 | 0.340 |
| | 48 | **0.112** | **0.220** | 0.120 | 0.233 | 0.572 | 0.544 | 0.229 | 0.339 | 0.205 | 0.326 | 0.409 | 0.478 | 0.136 | 0.250 | 0.355 | 0.437 |
| | 96 | 0.153 | **0.260** | **0.150** | 0.262 | 1.137 | 0.820 | 0.291 | 0.389 | 0.402 | 0.457 | 0.492 | 0.532 | 0.190 | 0.303 | 0.452 | 0.504 |
| PEMS07 | 12 | **0.056** | **0.146** | 0.067 | 0.165 | 0.118 | 0.235 | 0.095 | 0.207 | 0.094 | 0.200 | 0.173 | 0.304 | 0.082 | 0.181 | 0.115 | 0.242 |
| | 24 | **0.075** | **0.167** | 0.088 | 0.190 | 0.242 | 0.341 | 0.150 | 0.262 | 0.139 | 0.247 | 0.271 | 0.383 | 0.101 | 0.204 | 0.210 | 0.329 |
| | 48 | **0.101** | **0.197** | 0.110 | 0.215 | 0.562 | 0.541 | 0.253 | 0.340 | 0.311 | 0.369 | 0.446 | 0.495 | 0.134 | 0.238 | 0.398 | 0.458 |
| | 96 | 0.144 | **0.242** | **0.139** | 0.245 | 1.096 | 0.795 | 0.346 | 0.404 | 0.396 | 0.442 | 0.628 | 0.577 | 0.181 | 0.279 | 0.594 | 0.553 |
| PEMS08 | 12 | **0.071** | **0.167** | 0.079 | 0.182 | 0.133 | 0.247 | 0.168 | 0.232 | 0.165 | 0.214 | 0.227 | 0.343 | 0.112 | 0.212 | 0.154 | 0.276 |
| | 24 | **0.096** | **0.193** | 0.115 | 0.219 | 0.249 | 0.343 | 0.224 | 0.281 | 0.215 | 0.260 | 0.318 | 0.409 | 0.141 | 0.238 | 0.248 | 0.353 |
| | 48 | **0.152** | 0.243 | 0.186 | **0.235** | 0.569 | 0.544 | 0.321 | 0.354 | 0.315 | 0.355 | 0.497 | 0.510 | 0.198 | 0.283 | 0.440 | 0.470 |
| | 96 | 0.263 | **0.299** | **0.221** | **0.267** | 1.166 | 0.814 | 0.408 | 0.417 | 0.377 | 0.397 | 0.721 | 0.592 | 0.320 | 0.351 | 0.674 | 0.565 |
| $1^{st}$ Count | | **13** | **13** | 4 | 3 | 0 | 0 | 0 | 0 | 0 | 0 | 0 | 0 | 0 | 0 | 0 | 0 |

*Table 9.* Multivariate forecasting results with forecasting horizons $F \in \{96, 192, 336, 720\}$ and fixed look-back window size $T = 336$.

| Models | | CGTFra (Ours) | | FilterNet (NeurIPS'24) | | iTransformer (ICLR'24) | | PatchTST (ICLR'23) | | TimesNet (ICLR'23) | |
|---|---|---|---|---|---|---|---|---|---|---|---|
| Metrics | | MSE | MAE | MSE | MAE | MSE | MAE | MSE | MAE | MSE | MAE |
| ETTm1 | 96 | 0.292 | **0.335** | **0.289** | 0.344 | 0.303 | 0.357 | 0.294 | 0.345 | 0.335 | 0.380 |
| | 192 | **0.326** | **0.361** | 0.331 | 0.369 | 0.345 | 0.383 | 0.334 | 0.371 | 0.358 | 0.388 |
| | 336 | 0.366 | **0.381** | **0.364** | 0.389 | 0.382 | 0.405 | 0.371 | 0.392 | 0.406 | 0.418 |
| | 720 | **0.422** | **0.417** | 0.425 | 0.423 | 0.443 | 0.439 | 0.421 | 0.419 | 0.449 | 0.443 |
| ETTh1 | 96 | **0.378** | **0.396** | 0.379 | 0.404 | 0.402 | 0.418 | 0.381 | 0.405 | 0.398 | 0.418 |
| | 192 | **0.415** | **0.420** | 0.417 | 0.428 | 0.450 | 0.449 | 0.442 | 0.446 | 0.447 | 0.449 |
| | 336 | 0.438 | **0.435** | **0.437** | 0.443 | 0.479 | 0.470 | 0.445 | 0.454 | 0.493 | 0.468 |
| | 720 | **0.442** | **0.428** | 0.458 | 0.472 | 0.584 | 0.548 | 0.490 | 0.493 | 0.518 | 0.504 |
| Exchange | 96 | **0.087** | **0.211** | **0.087** | 0.216 | 0.099 | 0.226 | 0.093 | 0.213 | 0.117 | 0.253 |
| | 192 | 0.169 | **0.299** | **0.163** | 0.301 | 0.216 | 0.337 | 0.194 | 0.315 | 0.298 | 0.410 |
| | 336 | 0.312 | 0.417 | **0.287** | **0.399** | 0.395 | 0.466 | 0.354 | 0.435 | 0.456 | 0.513 |
| | 720 | 0.673 | 0.621 | **0.413** | **0.492** | 0.962 | 0.745 | 0.903 | 0.712 | 1.608 | 0.961 |
| Weather | 96 | **0.147** | 0.187 | 0.150 | **0.183** | 0.164 | 0.216 | 0.151 | 0.197 | 0.172 | 0.220 |
| | 192 | **0.188** | 0.229 | 0.193 | **0.221** | 0.205 | 0.251 | 0.197 | 0.244 | 0.219 | 0.261 |
| | 336 | **0.241** | 0.272 | 0.246 | **0.258** | 0.256 | 0.290 | 0.251 | 0.285 | 0.280 | 0.306 |
| | 720 | **0.308** | 0.331 | **0.308** | **0.295** | 0.326 | 0.338 | 0.321 | 0.335 | 0.365 | 0.359 |
| Electricity | 96 | **0.127** | **0.219** | 0.132 | 0.224 | 0.133 | 0.229 | 0.130 | 0.222 | 0.168 | 0.272 |
| | 192 | **0.137** | **0.216** | 0.143 | 0.237 | 0.156 | 0.251 | 0.148 | 0.240 | 0.184 | 0.289 |
| | 336 | **0.153** | **0.253** | 0.155 | 0.253 | 0.172 | 0.267 | 0.167 | 0.261 | 0.198 | 0.300 |
| | 720 | **0.193** | **0.285** | 0.195 | 0.292 | 0.209 | 0.304 | 0.202 | 0.291 | 0.220 | 0.320 |
| $1^{st}$ Count | | **15** | **14** | 8 | 8 | 0 | 0 | 0 | 0 | 0 | 0 |

## J.1. Comparing with Other Graph Transformers

Furthermore, to validate the effectiveness of our Graph Transformer, which is predicated on modeling consistency between shallow and deep IVD, we conducted a comparison against two other sota Graph Transformer methods: Ada-MSHyper (Shang et al., 2024) and Sageformer (Zhang et al., 2024). The results are presented in Table 10. While Ada-MSHyper exhibits certain advantages on the ETT datasets, this is primarily reflected in its slightly better MSE scores. In contrast, `CGTFra` demonstrates a more pronounced advantage on datasets with a larger number of variables, such as ECL and Traffic. For instance, on the Traffic dataset, `CGTFra` achieves an average reduction of 7.89% in MAE compared to Ada-MSHyper.

*Table 10.* Performance comparison of `CGTFra` and two other graph Transformers. "-" indicates that the original method was not evaluated in the corresponding scenario, or we faced the NaN error.

| Datasets | | ETTm1 | | ETTm2 | | ETTh1 | | ETTh2 | | Exchange | | Weather | | ECL | | Traffic | |
|---|---|---|---|---|---|---|---|---|---|---|---|---|---|---|---|---|---|---|
| Metrics | | MSE | MAE | MSE | MAE | MSE | MAE | MSE | MAE | MSE | MAE | MSE | MAE | MSE | MAE | MSE | MAE |
| **CGTFra** | 96 | 0.315 | **0.344** | **0.171** | **0.249** | 0.372 | **0.387** | **0.288** | **0.336** | 0.083 | 0.202 | **0.152** | **0.190** | **0.137** | **0.227** | **0.387** | **0.239** |
| | 192 | 0.366 | **0.372** | 0.238 | **0.293** | **0.425** | **0.418** | **0.364** | **0.384** | **0.173** | **0.296** | **0.203** | **0.239** | **0.155** | **0.243** | **0.417** | **0.249** |
| | 336 | 0.398 | **0.395** | 0.300 | **0.333** | 0.470 | **0.441** | **0.410** | **0.422** | **0.324** | **0.412** | **0.257** | **0.279** | **0.170** | **0.259** | **0.434** | **0.261** |
| | 720 | 0.472 | **0.435** | 0.397 | **0.391** | **0.469** | **0.462** | **0.414** | **0.433** | **0.668** | **0.619** | **0.338** | **0.334** | **0.198** | **0.283** | 0.472 | **0.279** |
| | Avg | 0.388 | **0.386** | 0.277 | **0.316** | **0.434** | **0.427** | **0.369** | **0.394** | **0.312** | **0.382** | **0.238** | **0.260** | **0.165** | **0.253** | **0.427** | **0.257** |
| **Ada-MSHyper** | 96 | **0.309** | 0.357 | 0.173 | 0.261 | 0.376 | 0.395 | 0.291 | 0.338 | - | - | 0.161 | 0.202 | 0.144 | 0.241 | 0.405 | 0.263 |
| | 192 | **0.362** | 0.385 | **0.235** | 0.307 | 0.436 | **0.418** | 0.370 | 0.389 | - | - | 0.209 | 0.248 | 0.160 | 0.247 | 0.419 | 0.275 |
| | 336 | **0.394** | 0.409 | **0.295** | 0.340 | 0.468 | 0.447 | 0.426 | 0.434 | - | - | 0.263 | 0.289 | 0.176 | 0.273 | 0.439 | 0.278 |
| | 720 | **0.461** | 0.447 | **0.389** | 0.402 | **0.469** | 0.472 | 0.418 | 0.439 | - | - | 0.349 | 0.346 | 0.212 | 0.293 | **0.467** | 0.299 |
| | Avg | **0.382** | 0.400 | **0.273** | 0.328 | 0.437 | 0.433 | 0.376 | 0.400 | - | - | 0.246 | 0.271 | 0.173 | 0.264 | 0.433 | 0.279 |
| **Sageformer** | 96 | 0.333 | 0.366 | 0.175 | 0.259 | 0.377 | 0.394 | 0.291 | 0.339 | **0.082** | **0.201** | 0.165 | 0.207 | 0.148 | 0.246 | - | - |
| | 192 | 0.371 | 0.389 | 0.241 | 0.301 | 0.428 | 0.426 | 0.376 | 0.394 | 0.177 | 0.299 | 0.211 | 0.251 | 0.163 | 0.248 | - | - |
| | 336 | 0.406 | 0.409 | 0.302 | 0.341 | **0.466** | 0.448 | 0.417 | 0.428 | 0.333 | 0.418 | 0.269 | 0.292 | 0.181 | 0.265 | - | - |
| | 720 | 0.478 | 0.449 | 0.399 | 0.396 | 0.487 | 0.476 | 0.422 | 0.441 | 0.866 | 0.702 | 0.347 | 0.345 | 0.209 | 0.306 | - | - |
| | Avg | 0.397 | 0.403 | 0.279 | 0.324 | 0.440 | 0.436 | 0.377 | 0.401 | 0.365 | 0.405 | 0.248 | 0.274 | 0.175 | 0.266 | NaN | Error |

# K. Verification of Framework Generality

To validate the extensibility of the three core designs proposed in this work, we perform corresponding module replacements or introduce CAL for seven existing models, including DUET, iTransformer, VCformer, CASA, FilterNet, iFlashformer, iFlowformer, iInformer, and iReformer. **To ensure a fair comparison, all baseline experiments were conducted using their released code and hyperparameters, under identical hyperparameters, random seeds, and experimental hardware and software environment versions. Additionally, our released code includes the source files, scripts, and documentation necessary to reproduce these experiments**. As shown in Table 11, we observe that **introducing FMR alone yields only marginal gains**. In contrast, **incorporating DGL significantly yields greater performance improvements to a certain extent, highlighting the importance of explicitly modeling IVD in deeper layers**. Building on this, the introduction of CAL further improves forecasting performance across multiple heterogeneous datasets, with the effect **being more pronounced for VCformer**. For instance, on the ETTh1 dataset, MSE decreases from 0.398 to 0.382. Moreover, we have to acknowledge that achieving performance gains by modifying sota methods while using identical hyperparameters poses considerable challenges.

- DUET (Qiu et al., 2025): DUET captures IVD by employing metric learning in the frequency domain, subsequently feeding IVD as masks to the self-attention scores within a variable Transformer. As DUET does not involve linear upsampling, our proposed FMR cannot be directly validated. Concurrently, DUET also lacks deep-layer IVD modeling. Therefore, we embed DGL and CAL into DUET for comparative experiments. https://github.com/decisionintelligence/DUET

- iTransformer (Liu et al., 2024): iTransformer encodes timestamp information into the input signals via concatenation, and computes inter-variable correlations among tokens corresponding to individual variables, and then employs FFNs to capture deep temporal dynamics. Its architecture is consistent with the standard Transformer (i.e., temporal Transformer), except that inverted token embedding. Accordingly, we replace the FFN in iTransformer with DGL to emphasize the importance of deep layer IVD, and further incorporate CAL on top of DGL to enhance consistent IVD modeling across both deep and shallow layers. https://github.com/thuml/iTransformer

- VCformer (Yang et al., 2024): VCformer likewise encodes timestamp information into the input via concatenation, and computes the inter-series correlation on different lags between queries and keys, and employ another Koopman theory-based temporal learner (namely KTD) to replace the FFN. Therefore, VCformer also captures IVD only at shallow layers. To validate extensibility, we replace their input embedding layer with our FMR, and replace KTD with the proposed DGL. https://github.com/CSyyn/VCformer

- CASA (Lee et al., 2025): CASA replaces the self-attention layer in the Transformer with a CNN autoencoder-based score attention, and **is therefore not a Transformer architecture. Since CASA encodes inputs using a single linear layer, we only replace its input embedding method with the proposed FMR to evaluate the generality of FMR**. https://github.com/lmh9507/CASA

The complete results for the other Variate Transformers are reported in Table 12. As some Variate Transformers redesign more efficient self-attention layers that may lose explicit attention scores, CAL cannot be integrated into these Variate Transformers.

In Figure 10, we present the DTW and PCC of the ETTh1 dataset, they characterize the true similarities and dependencies among variables in multivariate time series. **MSE and MAE, focusing solely on point-wise numerical discrepancies, overlook overall time-series shape similarity and fail to measure inter-variable correlations.**

Therefore, we conduct the effectiveness verification of DGL and CAL on three existing baselines with DTW and PCC as comparative metrics (their definitions are provided in Appendix C). **DTW prioritizes trend pattern matching, while PCC quantifies the model's capacity to capture co-variation among variables.** As shown in Table 13, **incorporating DGL and CAL enables iTransformer to achieve lower MSE and MAE values, along with superior DTW and PCC, indicating that deep modeling of IVD enhances forecasting of future fluctuations (in terms of magnitude) and improves the accuracy of dependency modeling (in terms of similarity). These results collectively demonstrate the strong generalizability of DGL and CAL.**

*Table 11.* Verification of Framework Generality. Full results for four prediction length and fixed input $T$=96. "iTrans" is iTransformer. "-" indicates that the original method was not evaluated in the corresponding scenario or we faced the issue of out of memory.

| | | | ETTm1 | | ETTm2 | | ETTh1 | | ETTh2 | | Exchange | | Weather | | ECL | | Solar | | Traffic | |
|---|---|---|---|---|---|---|---|---|---|---|---|---|---|---|---|---|---|---|---|---|
| Datasets | | | MSE | MAE | MSE | MAE | MSE | MAE | MSE | MAE | MSE | MAE | MSE | MAE | MSE | MAE | MSE | MAE | MSE | MAE |
| DUET | original | 96 | 0.322 | 0.354 | 0.174 | 0.254 | 0.389 | 0.400 | 0.295 | 0.345 | 0.084 | 0.203 | 0.162 | 0.201 | 0.146 | 0.233 | 0.249 | 0.269 | 0.407 | 0.252 |
| | | 192 | 0.370 | 0.380 | 0.239 | 0.299 | 0.431 | 0.426 | 0.370 | 0.391 | 0.179 | 0.300 | 0.217 | 0.251 | 0.163 | 0.249 | 0.221 | 0.230 | 0.431 | 0.261 |
| | | 336 | 0.407 | 0.403 | 0.301 | 0.339 | 0.473 | 0.450 | 0.408 | 0.419 | 0.285 | 0.390 | 0.268 | 0.290 | 0.174 | 0.261 | 0.245 | 0.242 | 0.458 | 0.271 |
| | | 720 | 0.464 | 0.439 | 0.400 | 0.396 | 0.502 | 0.484 | 0.415 | 0.435 | 0.686 | 0.625 | 0.342 | 0.339 | 0.204 | 0.288 | 0.247 | 0.244 | 0.504 | 0.291 |
| DUET | + DGL | 96 | 0.322 | 0.353 | 0.171 | 0.251 | 0.383 | 0.398 | 0.290 | 0.341 | 0.083 | 0.202 | 0.166 | 0.208 | 0.139 | 0.228 | 0.230 | 0.249 | 0.410 | 0.253 |
| | | 192 | 0.367 | 0.376 | 0.237 | 0.295 | 0.427 | 0.423 | 0.366 | 0.388 | 0.177 | 0.299 | 0.212 | 0.250 | 0.155 | 0.243 | 0.233 | 0.246 | 0.434 | 0.262 |
| | | 336 | 0.406 | 0.403 | 0.302 | 0.339 | 0.476 | 0.448 | 0.407 | 0.421 | 0.287 | 0.391 | 0.268 | 0.291 | 0.169 | 0.257 | 0.258 | 0.271 | 0.453 | 0.269 |
| | | 720 | 0.460 | 0.433 | 0.399 | 0.396 | 0.488 | 0.476 | 0.409 | 0.431 | 0.636 | 0.600 | 0.342 | 0.339 | 0.202 | 0.291 | 0.251 | 0.267 | 0.496 | 0.288 |
| DUET | + CAL | 96 | 0.324 | 0.356 | 0.172 | 0.252 | 0.378 | 0.393 | 0.288 | 0.339 | 0.083 | 0.202 | 0.152 | 0.192 | 0.139 | 0.228 | 0.231 | 0.250 | 0.412 | 0.253 |
| | | 192 | 0.370 | 0.379 | 0.239 | 0.297 | 0.427 | 0.421 | 0.363 | 0.385 | 0.177 | 0.299 | 0.205 | 0.244 | 0.155 | 0.243 | 0.238 | 0.248 | 0.433 | 0.260 |
| | | 336 | 0.405 | 0.399 | 0.304 | 0.341 | 0.472 | 0.445 | 0.408 | 0.422 | 0.285 | 0.388 | 0.256 | 0.280 | 0.169 | 0.258 | 0.247 | 0.245 | 0.462 | 0.273 |
| | | 720 | 0.462 | 0.435 | 0.402 | 0.399 | 0.475 | 0.469 | 0.417 | 0.435 | 0.673 | 0.615 | 0.333 | 0.338 | 0.193 | 0.284 | 0.253 | 0.268 | 0.501 | 0.291 |
| iTrans | original | 96 | 0.342 | 0.377 | 0.186 | 0.272 | 0.387 | 0.405 | 0.301 | 0.350 | 0.086 | 0.206 | 0.181 | 0.221 | 0.148 | 0.239 | 0.201 | 0.234 | 0.392 | 0.268 |
| | | 192 | 0.383 | 0.396 | 0.254 | 0.314 | 0.441 | 0.436 | 0.381 | 0.399 | 0.181 | 0.303 | 0.226 | 0.259 | 0.167 | 0.258 | 0.239 | 0.263 | 0.413 | 0.277 |
| | | 336 | 0.418 | 0.418 | 0.317 | 0.353 | 0.491 | 0.462 | 0.423 | 0.432 | 0.338 | 0.422 | 0.283 | 0.300 | 0.181 | 0.275 | 0.248 | 0.272 | 0.425 | 0.283 |
| | | 720 | 0.487 | 0.456 | 0.416 | 0.408 | 0.509 | 0.494 | 0.430 | 0.446 | 0.869 | 0.704 | 0.359 | 0.351 | 0.209 | 0.299 | 0.250 | 0.275 | 0.459 | 0.300 |
| iTrans | + FMR | 96 | 0.340 | 0.373 | 0.183 | 0.265 | 0.382 | 0.398 | 0.299 | 0.350 | 0.084 | 0.204 | 0.180 | 0.222 | 0.141 | 0.235 | 0.199 | 0.237 | 0.393 | 0.268 |
| | | 192 | 0.377 | 0.389 | 0.249 | 0.309 | 0.434 | 0.429 | 0.379 | 0.399 | 0.176 | 0.299 | 0.222 | 0.258 | 0.157 | 0.250 | 0.233 | 0.259 | 0.413 | 0.276 |
| | | 336 | 0.412 | 0.411 | 0.314 | 0.350 | 0.483 | 0.454 | 0.419 | 0.430 | 0.339 | 0.423 | 0.279 | 0.300 | 0.171 | 0.264 | 0.242 | 0.269 | 0.428 | 0.282 |
| | | 720 | 0.481 | 0.450 | 0.418 | 0.409 | 0.492 | 0.480 | 0.426 | 0.445 | 0.834 | 0.690 | 0.356 | 0.350 | 0.233 | 0.316 | 0.244 | 0.273 | 0.458 | 0.299 |
| iTrans | + DGL | 96 | 0.331 | 0.368 | 0.183 | 0.268 | 0.384 | 0.403 | 0.305 | 0.355 | 0.086 | 0.207 | 0.167 | 0.211 | 0.137 | 0.233 | 0.200 | 0.238 | 0.410 | 0.281 |
| | | 192 | 0.376 | 0.392 | 0.253 | 0.313 | 0.435 | 0.432 | 0.392 | 0.406 | 0.179 | 0.303 | 0.214 | 0.254 | 0.154 | 0.248 | 0.239 | 0.264 | 0.421 | 0.281 |
| | | 336 | 0.409 | 0.412 | 0.317 | 0.352 | 0.481 | 0.453 | 0.426 | 0.436 | 0.336 | 0.420 | 0.275 | 0.299 | 0.167 | 0.263 | 0.248 | 0.274 | 0.440 | 0.286 |
| | | 720 | 0.483 | 0.453 | 0.417 | 0.408 | 0.494 | 0.481 | 0.435 | 0.451 | 0.870 | 0.704 | 0.353 | 0.349 | 0.219 | 0.307 | 0.249 | 0.276 | 0.466 | 0.304 |
| iTrans | + CAL | 96 | 0.330 | 0.367 | 0.183 | 0.264 | 0.382 | 0.402 | 0.305 | 0.354 | 0.086 | 0.207 | 0.166 | 0.211 | 0.135 | 0.232 | 0.199 | 0.234 | 0.399 | 0.272 |
| | | 192 | 0.372 | 0.388 | 0.251 | 0.311 | 0.434 | 0.431 | 0.384 | 0.401 | 0.175 | 0.299 | 0.213 | 0.254 | 0.154 | 0.248 | 0.234 | 0.262 | 0.433 | 0.281 |
| | | 336 | 0.407 | 0.410 | 0.316 | 0.352 | 0.481 | 0.455 | 0.424 | 0.434 | 0.338 | 0.422 | 0.267 | 0.294 | 0.167 | 0.262 | 0.251 | 0.275 | 0.451 | 0.287 |
| | | 720 | 0.479 | 0.449 | 0.415 | 0.407 | 0.479 | 0.473 | 0.426 | 0.444 | 0.816 | 0.684 | 0.350 | 0.347 | 0.194 | 0.288 | 0.249 | 0.276 | 0.460 | 0.303 |
| VCformer | original | 96 | 0.331 | 0.364 | 0.184 | 0.266 | 0.405 | 0.410 | 0.302 | 0.349 | 0.085 | 0.206 | 0.186 | 0.224 | 0.152 | 0.246 | - | - | - | - |
| | | 192 | 0.379 | 0.389 | 0.250 | 0.309 | 0.455 | 0.439 | 0.383 | 0.396 | 0.175 | 0.300 | 0.238 | 0.266 | 0.170 | 0.261 | - | - | - | - |
| | | 336 | 0.419 | 0.416 | 0.318 | 0.352 | 0.530 | 0.476 | 0.421 | 0.430 | 0.327 | 0.415 | 0.288 | 0.303 | 0.186 | 0.277 | - | - | - | - |
| | | 720 | 0.487 | 0.453 | 0.414 | 0.407 | 0.561 | 0.515 | 0.429 | 0.446 | 0.844 | 0.691 | 0.365 | 0.352 | 0.235 | 0.328 | - | - | - | - |
| VCformer | + FMR | 96 | 0.333 | 0.365 | 0.183 | 0.265 | 0.393 | 0.401 | 0.308 | 0.351 | 0.088 | 0.211 | 0.184 | 0.224 | 0.147 | 0.241 | - | - | - | - |
| | | 192 | 0.373 | 0.387 | 0.250 | 0.309 | 0.451 | 0.434 | 0.383 | 0.395 | 0.176 | 0.300 | 0.231 | 0.264 | 0.163 | 0.255 | - | - | - | - |
| | | 336 | 0.410 | 0.410 | 0.314 | 0.350 | 0.484 | 0.450 | 0.419 | 0.430 | 0.336 | 0.421 | 0.285 | 0.302 | 0.177 | 0.272 | - | - | - | - |
| | | 720 | 0.476 | 0.447 | 0.415 | 0.406 | 0.499 | 0.480 | 0.431 | 0.447 | 0.869 | 0.703 | 0.361 | 0.351 | 0.241 | 0.332 | - | - | - | - |
| VCformer | + DGL | 96 | 0.323 | 0.359 | 0.184 | 0.269 | 0.398 | 0.410 | 0.305 | 0.352 | 0.085 | 0.206 | 0.165 | 0.208 | 0.139 | 0.236 | - | - | - | - |
| | | 192 | 0.379 | 0.389 | 0.249 | 0.310 | 0.447 | 0.439 | 0.387 | 0.401 | 0.175 | 0.299 | 0.210 | 0.252 | 0.158 | 0.248 | - | - | - | - |
| | | 336 | 0.415 | 0.412 | 0.310 | 0.348 | 0.484 | 0.456 | 0.425 | 0.433 | 0.326 | 0.412 | 0.270 | 0.293 | 0.173 | 0.266 | - | - | - | - |
| | | 720 | 0.473 | 0.445 | 0.412 | 0.405 | 0.494 | 0.482 | 0.440 | 0.453 | 0.868 | 0.700 | 0.351 | 0.347 | 0.224 | 0.314 | - | - | - | - |
| VCformer | + CAL | 96 | 0.323 | 0.360 | 0.179 | 0.262 | 0.382 | 0.400 | 0.300 | 0.349 | 0.085 | 0.205 | 0.164 | 0.208 | 0.135 | 0.233 | - | - | - | - |
| | | 192 | 0.369 | 0.384 | 0.246 | 0.307 | 0.438 | 0.434 | 0.380 | 0.397 | 0.176 | 0.299 | 0.211 | 0.251 | 0.157 | 0.249 | - | - | - | - |
| | | 336 | 0.415 | 0.412 | 0.309 | 0.347 | 0.485 | 0.457 | 0.426 | 0.433 | 0.339 | 0.423 | 0.271 | 0.295 | 0.170 | 0.264 | - | - | - | - |
| | | 720 | 0.485 | 0.453 | 0.414 | 0.406 | 0.497 | 0.486 | 0.429 | 0.447 | 0.846 | 0.696 | 0.351 | 0.346 | 0.207 | 0.299 | - | - | - | - |
| CASA | original | 96 | 0.322 | 0.359 | 0.175 | 0.257 | 0.378 | 0.403 | 0.298 | 0.347 | - | - | 0.162 | 0.207 | 0.140 | 0.236 | 0.193 | 0.234 | 0.392 | 0.260 |
| | | 192 | 0.368 | 0.386 | 0.241 | 0.300 | 0.428 | 0.429 | 0.375 | 0.396 | - | - | 0.209 | 0.251 | 0.160 | 0.253 | 0.227 | 0.260 | 0.415 | 0.274 |
| | | 336 | 0.407 | 0.409 | 0.299 | 0.339 | 0.478 | 0.453 | 0.420 | 0.431 | - | - | 0.267 | 0.292 | 0.181 | 0.274 | 0.240 | 0.274 | 0.434 | 0.281 |
| | | 720 | 0.468 | 0.447 | 0.399 | 0.397 | 0.482 | 0.476 | 0.439 | 0.451 | - | - | 0.359 | 0.352 | 0.206 | 0.298 | 0.242 | 0.276 | 0.468 | 0.296 |
| CASA | + FMR | 96 | 0.321 | 0.359 | 0.174 | 0.256 | 0.378 | 0.401 | 0.294 | 0.346 | - | - | 0.163 | 0.207 | 0.136 | 0.232 | 0.192 | 0.233 | 0.405 | 0.262 |
| | | 192 | 0.369 | 0.386 | 0.240 | 0.299 | 0.426 | 0.428 | 0.372 | 0.395 | - | - | 0.207 | 0.248 | 0.159 | 0.253 | 0.222 | 0.258 | 0.432 | 0.274 |
| | | 336 | 0.418 | 0.416 | 0.298 | 0.337 | 0.480 | 0.454 | 0.418 | 0.430 | - | - | 0.264 | 0.291 | 0.179 | 0.273 | 0.238 | 0.272 | 0.447 | 0.280 |
| | | 720 | 0.460 | 0.444 | 0.398 | 0.396 | 0.484 | 0.477 | 0.429 | 0.446 | - | - | 0.347 | 0.344 | 0.204 | 0.295 | 0.240 | 0.274 | 0.492 | 0.301 |
| FilterNet | original | 96 | 0.317 | 0.357 | 0.175 | 0.257 | 0.381 | 0.399 | 0.296 | 0.346 | - | - | 0.164 | 0.210 | 0.147 | 0.242 | - | - | 0.431 | 0.295 |
| | | 192 | 0.364 | 0.384 | 0.239 | 0.300 | 0.440 | 0.428 | 0.369 | 0.396 | - | - | 0.214 | 0.256 | 0.162 | 0.254 | - | - | 0.448 | 0.298 |
| | | 336 | 0.396 | 0.407 | 0.295 | 0.337 | 0.487 | 0.451 | 0.420 | 0.432 | - | - | 0.273 | 0.299 | 0.177 | 0.272 | - | - | 0.465 | 0.303 |
| | | 720 | 0.457 | 0.444 | 0.398 | 0.395 | 0.494 | 0.471 | 0.432 | 0.447 | - | - | 0.359 | 0.353 | 0.228 | 0.318 | - | - | 0.497 | 0.320 |
| FilterNet | + FMR | 96 | 0.318 | 0.359 | 0.174 | 0.255 | 0.376 | 0.397 | 0.293 | 0.343 | - | - | 0.160 | 0.206 | 0.144 | 0.239 | - | - | 0.422 | 0.289 |
| | | 192 | 0.364 | 0.383 | 0.238 | 0.299 | 0.438 | 0.427 | 0.368 | 0.394 | - | - | 0.209 | 0.252 | 0.159 | 0.252 | - | - | 0.445 | 0.295 |
| | | 336 | 0.395 | 0.406 | 0.295 | 0.337 | 0.489 | 0.451 | 0.416 | 0.433 | - | - | 0.270 | 0.296 | 0.177 | 0.273 | - | - | 0.462 | 0.301 |
| | | 720 | 0.455 | 0.442 | 0.396 | 0.395 | 0.496 | 0.472 | 0.438 | 0.450 | - | - | 0.353 | 0.350 | 0.228 | 0.321 | - | - | 0.494 | 0.317 |

*Table 12.* Verification of Framework Generality on Variate Transformers (fixed input length $T$=96). As some variant Transformers redesign more efficient self-attention layers that may lose explicit attention scores, CAL cannot be integrated into these variate Transformers. To further evaluate the effectiveness of DGL and CAL, we employed additional evaluation metrics as seen in Table 13.

| Datasets | | | ETTh1 | | Weather | | ECL | | Solar | |
|---|---|---|---|---|---|---|---|---|---|---|
| Metrics | | | MSE | MAE | MSE | MAE | MSE | MAE | MSE | MAE |
| iTransformer | original | 96 | 0.387 | 0.405 | 0.181 | 0.221 | 0.148 | 0.239 | 0.201 | **0.234** |
| | | 192 | 0.441 | 0.436 | 0.226 | 0.259 | 0.167 | 0.258 | 0.239 | 0.263 |
| | | 336 | 0.491 | 0.462 | 0.283 | 0.300 | 0.181 | 0.275 | **0.248** | **0.272** |
| | | 720 | 0.509 | 0.494 | 0.359 | 0.351 | 0.209 | 0.299 | 0.250 | **0.275** |
| | + DGL | 96 | 0.384 | 0.403 | 0.167 | 0.212 | 0.137 | 0.233 | 0.200 | 0.238 |
| | | 192 | 0.435 | 0.432 | 0.214 | 0.255 | **0.154** | **0.248** | 0.236 | 0.264 |
| | | 336 | **0.481** | **0.453** | 0.275 | 0.299 | **0.167** | 0.263 | **0.248** | 0.274 |
| | | 720 | 0.494 | 0.481 | 0.353 | 0.349 | 0.219 | 0.307 | **0.249** | 0.276 |
| | + CAL | 96 | **0.382** | **0.402** | **0.166** | **0.211** | **0.135** | **0.232** | **0.199** | **0.234** |
| | | 192 | **0.434** | **0.431** | **0.213** | **0.254** | **0.154** | **0.248** | **0.234** | **0.262** |
| | | 336 | **0.481** | 0.455 | **0.267** | **0.294** | **0.167** | **0.262** | 0.251 | 0.275 |
| | | 720 | **0.479** | **0.473** | **0.350** | **0.347** | **0.194** | **0.288** | **0.249** | 0.276 |
| iFlashformer | original | 96 | 0.388 | 0.406 | 0.180 | 0.221 | 0.164 | 0.254 | 0.213 | 0.251 |
| | | 192 | **0.438** | 0.435 | 0.227 | 0.259 | **0.175** | **0.263** | **0.242** | **0.275** |
| | | 336 | 0.487 | 0.458 | 0.283 | 0.300 | **0.192** | **0.280** | **0.263** | **0.291** |
| | | 720 | 0.504 | 0.491 | 0.360 | 0.351 | **0.232** | **0.314** | **0.267** | **0.296** |
| | + DGL | 96 | **0.384** | **0.402** | **0.171** | **0.216** | **0.160** | **0.253** | **0.209** | **0.250** |
| | | 192 | 0.441 | **0.434** | **0.216** | **0.255** | **0.175** | 0.265 | 0.246 | **0.275** |
| | | 336 | **0.484** | **0.455** | **0.278** | **0.299** | 0.193 | 0.283 | 0.266 | 0.292 |
| | | 720 | **0.500** | **0.483** | **0.352** | **0.348** | **0.232** | 0.315 | 0.273 | 0.298 |
| iFlowformer | original | 96 | **0.385** | **0.402** | 0.187 | 0.226 | 0.169 | 0.255 | **0.215** | 0.255 |
| | | 192 | 0.446 | 0.437 | 0.230 | 0.262 | 0.180 | **0.265** | **0.246** | **0.277** |
| | | 336 | 0.503 | 0.470 | 0.285 | 0.301 | 0.198 | **0.283** | **0.266** | **0.292** |
| | | 720 | 0.559 | 0.522 | 0.363 | 0.352 | **0.238** | **0.317** | **0.272** | **0.297** |
| | + DGL | 96 | 0.387 | 0.403 | **0.176** | **0.220** | **0.163** | **0.254** | 0.218 | **0.254** |
| | | 192 | **0.443** | **0.435** | **0.220** | **0.257** | **0.174** | **0.265** | 0.251 | 0.279 |
| | | 336 | **0.484** | **0.454** | **0.273** | **0.296** | **0.197** | 0.285 | 0.277 | 0.297 |
| | | 720 | **0.500** | **0.481** | **0.351** | **0.345** | **0.238** | 0.319 | 0.285 | 0.304 |
| iInformer | original | 96 | 0.388 | 0.404 | 0.169 | **0.213** | 0.168 | 0.255 | **0.220** | 0.264 |
| | | 192 | 0.445 | 0.436 | 0.217 | 0.254 | 0.181 | **0.266** | **0.254** | 0.287 |
| | | 336 | 0.492 | 0.461 | 0.273 | 0.296 | 0.198 | **0.284** | **0.278** | **0.304** |
| | | 720 | 0.504 | 0.490 | 0.353 | 0.348 | 0.242 | **0.319** | **0.280** | **0.305** |
| | + DGL | 96 | 0.390 | 0.405 | **0.168** | 0.214 | 0.164 | 0.255 | 0.223 | 0.261 |
| | | 192 | 0.445 | 0.435 | 0.212 | 0.253 | 0.179 | 0.267 | 0.263 | 0.287 |
| | | 336 | 0.489 | 0.457 | 0.271 | 0.295 | 0.198 | 0.286 | 0.287 | 0.305 |
| | | 720 | 0.501 | 0.482 | 0.351 | 0.347 | 0.241 | 0.321 | 0.292 | 0.308 |
| | + CAL | 96 | **0.386** | **0.401** | **0.168** | **0.213** | **0.159** | **0.251** | 0.223 | **0.260** |
| | | 192 | **0.443** | **0.433** | **0.211** | **0.252** | **0.178** | 0.267 | 0.261 | **0.286** |
| | | 336 | **0.482** | **0.453** | **0.268** | **0.293** | **0.197** | 0.285 | 0.290 | 0.305 |
| | | 720 | **0.494** | **0.478** | **0.347** | **0.344** | **0.240** | 0.320 | 0.293 | 0.308 |
| iReformer | original | 96 | 0.386 | 0.402 | 0.185 | 0.226 | 0.169 | 0.257 | 0.222 | 0.263 |
| | | 192 | 0.447 | 0.437 | 0.230 | 0.262 | 0.180 | **0.266** | **0.255** | 0.285 |
| | | 336 | 0.502 | 0.469 | 0.283 | 0.301 | 0.198 | **0.284** | **0.277** | **0.302** |
| | | 720 | 0.548 | 0.516 | 0.359 | 0.349 | 0.241 | **0.319** | **0.280** | **0.303** |
| | + DGL | 96 | **0.383** | **0.401** | **0.176** | **0.220** | **0.161** | **0.254** | **0.221** | **0.259** |
| | | 192 | **0.442** | **0.434** | **0.222** | **0.259** | **0.176** | **0.266** | 0.258 | **0.283** |
| | | 336 | **0.480** | **0.452** | **0.274** | **0.296** | **0.195** | 0.285 | 0.285 | **0.302** |
| | | 720 | **0.492** | **0.478** | **0.351** | **0.346** | **0.239** | 0.320 | 0.289 | 0.305 |

*Table 13.* Additional evaluation metrics for evaluating the effectiveness of DGL and CAL.

| Datasets | | | ETTh1 | | | Weather | | | ECL | | | Solar | | |
|---|---|---|---|---|---|---|---|---|---|---|---|---|---|---|
| Metrics | | | MSE↓ | DTW↓ | PCC↑ | MSE↓ | DTW↓ | PCC↑ | MSE↓ | DTW↓ | PCC↑ | MSE↓ | DTW↓ | PCC↑ |
| DUET | original | 96 | 0.389 | 13.95 | 0.559 | 0.162 | 16.16 | 0.398 | 0.146 | 65.68 | 0.901 | 0.249 | 45.68 | 0.841 |
| | | 192 | 0.431 | 21.09 | 0.532 | 0.217 | 27.12 | 0.362 | 0.163 | 97.05 | 0.897 | **0.221** | **66.59** | **0.917** |
| | | 336 | 0.473 | 29.41 | 0.506 | 0.268 | 40.31 | 0.339 | 0.174 | 132.90 | 0.894 | **0.245** | **94.00** | 0.882 |
| | | 720 | 0.502 | 44.36 | 0.468 | 0.342 | 67.91 | 0.315 | 0.204 | 218.83 | 0.879 | **0.247** | **140.27** | **0.866** |
| | | Avg | 0.449 | 27.20 | 0.516 | 0.247 | 37.88 | 0.354 | 0.172 | 128.62 | 0.893 | **0.241** | 86.64 | **0.877** |
| | +DGL | 96 | 0.383 | 13.85 | 0.562 | 0.166 | 16.19 | 0.392 | **0.139** | 63.70 | **0.913** | **0.230** | **42.30** | **0.875** |
| | | 192 | **0.427** | 20.01 | 0.526 | 0.212 | 26.88 | 0.366 | **0.155** | 95.67 | 0.903 | 0.233 | 67.25 | 0.899 |
| | | 336 | 0.476 | 29.62 | 0.494 | 0.268 | 40.33 | 0.338 | **0.169** | 132.73 | **0.899** | 0.258 | 94.90 | 0.870 |
| | | 720 | 0.488 | 44.28 | 0.466 | 0.342 | 67.89 | 0.317 | 0.202 | 216.49 | 0.882 | 0.251 | 140.29 | 0.858 |
| | | Avg | 0.444 | 26.94 | 0.512 | 0.247 | 37.82 | 0.353 | 0.166 | 127.15 | 0.899 | 0.243 | 86.19 | 0.876 |
| | +CAL | 96 | **0.378** | **13.77** | **0.564** | **0.152** | **16.08** | **0.405** | **0.139** | **63.41** | 0.908 | 0.231 | 42.36 | 0.873 |
| | | 192 | **0.427** | **19.98** | **0.530** | **0.205** | **26.63** | **0.371** | **0.155** | **95.49** | **0.909** | 0.238 | 67.68 | 0.892 |
| | | 336 | **0.472** | **29.36** | **0.500** | **0.256** | **40.01** | **0.349** | **0.169** | **132.64** | **0.899** | 0.247 | 94.07 | **0.883** |
| | | 720 | **0.475** | **43.21** | **0.470** | **0.333** | **66.89** | **0.326** | **0.193** | **210.58** | **0.894** | 0.253 | 140.32 | 0.850 |
| | | Avg | **0.438** | **26.58** | **0.516** | **0.237** | **37.40** | **0.363** | **0.164** | **125.53** | **0.903** | 0.242 | **86.11** | 0.875 |
| iTransformer | original | 96 | 0.387 | 13.92 | 0.561 | 0.181 | 16.97 | 0.364 | 0.148 | 65.77 | 0.906 | 0.201 | 39.56 | 0.902 |
| | | 192 | 0.441 | 21.19 | 0.527 | 0.226 | 27.74 | 0.354 | 0.167 | 99.55 | 0.899 | 0.239 | 67.96 | **0.895** |
| | | 336 | 0.491 | 29.87 | 0.496 | 0.283 | 41.04 | 0.334 | 0.181 | 137.33 | 0.892 | **0.248** | **94.01** | 0.880 |
| | | 720 | 0.509 | 44.49 | 0.468 | 0.359 | 68.12 | 0.300 | 0.209 | 218.70 | 0.878 | 0.250 | 140.50 | 0.862 |
| | | Avg | 0.457 | 27.37 | 0.513 | 0.262 | 38.47 | 0.338 | 0.176 | 130.33 | 0.894 | 0.235 | 85.51 | 0.885 |
| | +DGL | 96 | 0.384 | 13.89 | 0.562 | 0.167 | **16.27** | 0.383 | 0.137 | 63.31 | 0.911 | 0.200 | 39.80 | 0.904 |
| | | 192 | 0.435 | 21.23 | 0.526 | 0.214 | **26.81** | 0.363 | **0.154** | 95.75 | 0.903 | 0.236 | 68.17 | **0.895** |
| | | 336 | **0.481** | 29.66 | 0.494 | 0.275 | **40.24** | 0.332 | **0.167** | 132.85 | 0.896 | **0.248** | 94.20 | **0.881** |
| | | 720 | 0.494 | 44.40 | 0.466 | 0.353 | **67.95** | 0.290 | 0.219 | 223.27 | 0.875 | **0.249** | 140.32 | 0.863 |
| | | Avg | 0.449 | 27.30 | 0.512 | 0.252 | **37.82** | 0.342 | 0.169 | 128.79 | 0.896 | **0.233** | 85.62 | **0.886** |
| | +CAL | 96 | **0.382** | **13.83** | **0.564** | **0.166** | 16.28 | **0.394** | **0.135** | **62.96** | **0.912** | **0.199** | **39.53** | **0.905** |
| | | 192 | **0.434** | **21.07** | **0.530** | **0.213** | 27.01 | **0.367** | **0.154** | **95.44** | **0.905** | 0.234 | **67.32** | **0.895** |
| | | 336 | **0.481** | **29.55** | **0.500** | **0.267** | 40.36 | **0.340** | **0.167** | **132.68** | **0.898** | 0.251 | 94.60 | 0.880 |
| | | 720 | **0.479** | **43.55** | **0.470** | **0.350** | 68.07 | **0.304** | **0.194** | **210.73** | **0.885** | **0.249** | **140.31** | **0.864** |
| | | Avg | **0.444** | **27.00** | **0.516** | **0.249** | 37.93 | **0.351** | **0.163** | **125.45** | **0.900** | **0.233** | **85.44** | **0.886** |
| VCformer | original | 96 | 0.405 | 14.096 | 0.5496 | 0.186 | 17.41 | 0.346 | 0.152 | 66.03 | 0.892 | - | - | - |
| | | 192 | 0.455 | 21.57 | 0.528 | 0.238 | 28.12 | 0.339 | 0.170 | 99.79 | 0.896 | - | - | - |
| | | 336 | 0.530 | 30.89 | 0.502 | 0.288 | 41.53 | 0.336 | 0.186 | 138.81 | 0.890 | - | - | - |
| | | 720 | 0.561 | 47.28 | 0.4528 | 0.365 | 68.49 | 0.287 | 0.235 | 231.67 | 0.871 | - | - | - |
| | | Avg | 0.488 | 28.46 | 0.508 | 0.269 | 38.89 | 0.327 | 0.186 | 134.01 | 0.887 | - | - | - |
| | +DGL | 96 | 0.398 | 14.08 | 0.561 | 0.165 | **16.14** | 0.376 | 0.139 | 64.51 | 0.909 | - | - | - |
| | | 192 | 0.447 | 21.18 | 0.526 | **0.210** | 26.89 | **0.365** | 0.158 | 96.92 | 0.900 | - | - | - |
| | | 336 | **0.484** | 29.77 | 0.500 | **0.270** | **40.32** | **0.329** | 0.173 | 135.46 | 0.897 | - | - | - |
| | | 720 | **0.494** | 43.92 | 0.470 | **0.351** | 68.23 | **0.299** | 0.224 | 225.94 | 0.873 | - | - | - |
| | | Avg | 0.456 | **27.24** | 0.514 | **0.249** | 37.90 | **0.342** | 0.174 | 130.70 | 0.895 | - | - | - |
| | +CAL | 96 | **0.382** | **13.79** | **0.567** | **0.164** | 16.19 | **0.381** | **0.135** | **63.09** | **0.910** | - | - | - |
| | | 192 | **0.438** | 21.17 | **0.535** | 0.211 | **26.71** | 0.353 | **0.157** | **96.26** | **0.907** | - | - | - |
| | | 336 | 0.485 | **29.68** | **0.504** | 0.271 | 40.33 | 0.328 | **0.170** | **134.00** | **0.899** | - | - | - |
| | | 720 | 0.497 | 44.33 | **0.475** | **0.351** | **68.01** | 0.294 | **0.207** | **218.66** | **0.876** | - | - | - |
| | | Avg | **0.451** | **27.24** | **0.520** | **0.249** | **37.81** | 0.339 | **0.167** | **128.00** | **0.898** | - | - | - |

## K.1. T-SNE Visualization

Given that the ETT datasets exhibit more stable inter-variable dependencies compared to Weather and Traffic, we investigated the latent representations learned by different models on the ETTh1 dataset (specifically, using 1,500 samples from the test set). Building on our preliminary analysis, Variables 1, 2, 3, and 5 were selected to visualize the learned dependencies via t-SNE. We employ t-SNE visualization to demonstrate two key insights regarding the learned representations: **1)**

**representations of the same variable should form compact clusters**; and **2) the spatial distances between different variables should reflect the inter-variable dependencies captured by the model, where stronger dependencies imply closer proximity in the embedding space**. Figures 3, 17, 18, and 19 illustrate the latent representations generated by iTransformer and DUET, respectively. Following the integration of DGL and CAL, we observe that intra-variable clusters become significantly more compact, while inter-variable distances align more closely with the ground-truth dependencies. These results demonstrate the generalizability and effectiveness of deep IVD modeling and the consistency constraints.

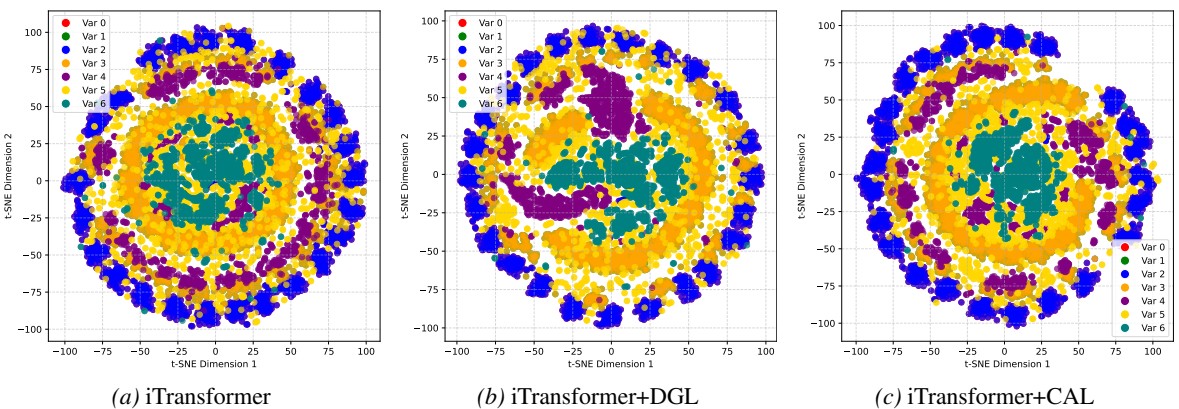

*(a)* iTransformer  *(b)* iTransformer+DGL  *(c)* iTransformer+CAL

*Figure 17.* T-SNE visualization for the total seven variables of Figure 3 on the 1500 samples of ETTh1 test set. Based on the DTW and PCC metrics (as detailed in Figure 10), the dependency strengths of these variables are ranked in descending order: 1&3 > 3&5 > 2&5.

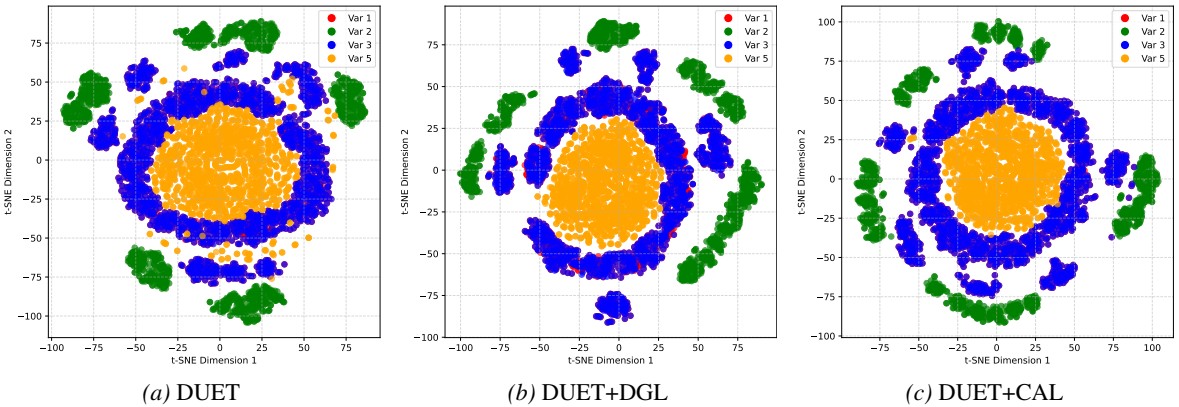

*(a)* DUET  *(b)* DUET+DGL  *(c)* DUET+CAL

*Figure 18.* T-SNE visualization for latent representations of variable 1, 2, 3, and 5 on the 1500 samples of ETTh1 test set. **A higher degree of dependency or similarity corresponds to a smaller spatial distance**. All variable's t-SNE is provided in Figure 19.

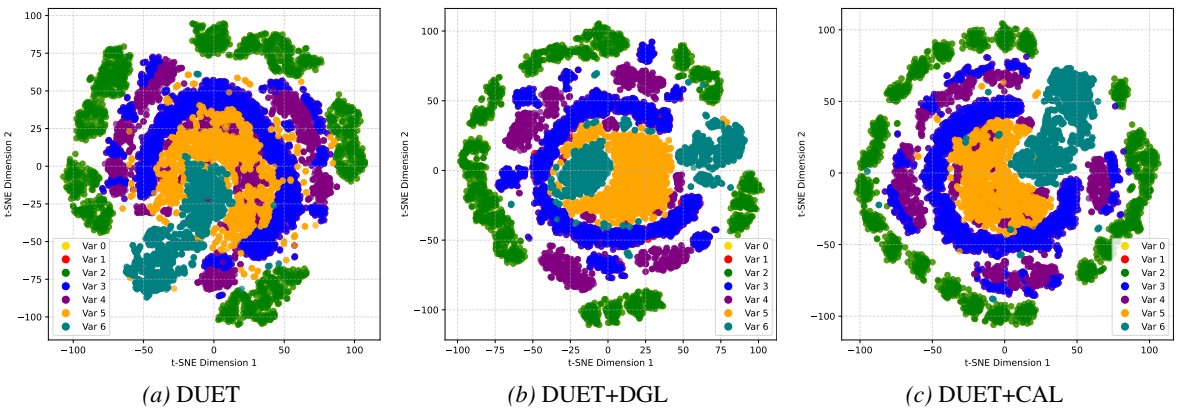

*(a)* DUET  *(b)* DUET+DGL  *(c)* DUET+CAL

*Figure 19.* T-SNE visualization for seven variables on the 1500 samples of ETTh1 test set.

### K.2. Visualization of the Loss Curves Indicating Optimization Difficulty

Figures 20, 21 and 22 illustrate the training and validation loss curves for three distinct frameworks, DUET, VCformer and iTransformer, across two forecasting scenarios on the Weather dataset. The curves reveal that methods relying solely on self-attention mechanisms in shallow layers to model inter-variable dependencies—such as DUET and iTransformer—encounter significant optimization challenges. Meanwhile, we observe that even in specific scenarios where the introduction of DGL initially leads to a slight performance degradation (e.g., DUET with input-96/predict-96), the model ultimately achieves superior performance when reinforced by the CAL constraint. This demonstrates that CAL provides substantial robustness to the model optimization.

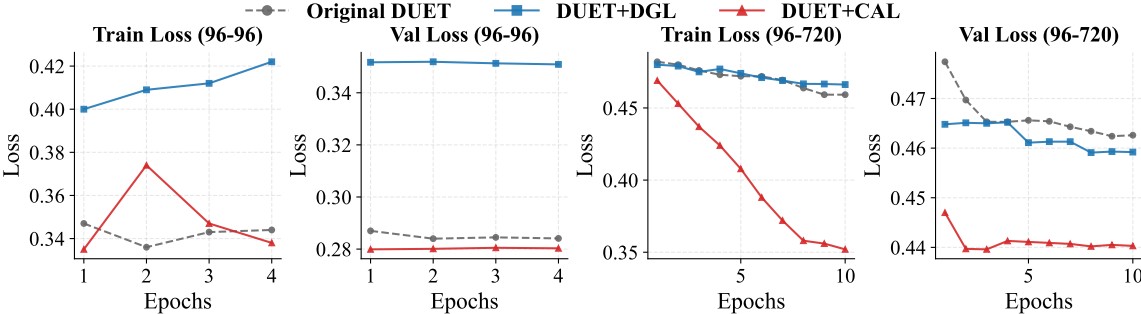

*Figure 20.* Visualization of training and validation loss curves for DUET on two prediction scenarios (Weather dataset). In certain scenarios (Input 96-Predict 96), early stopping is triggered as early as the fourth epoch.

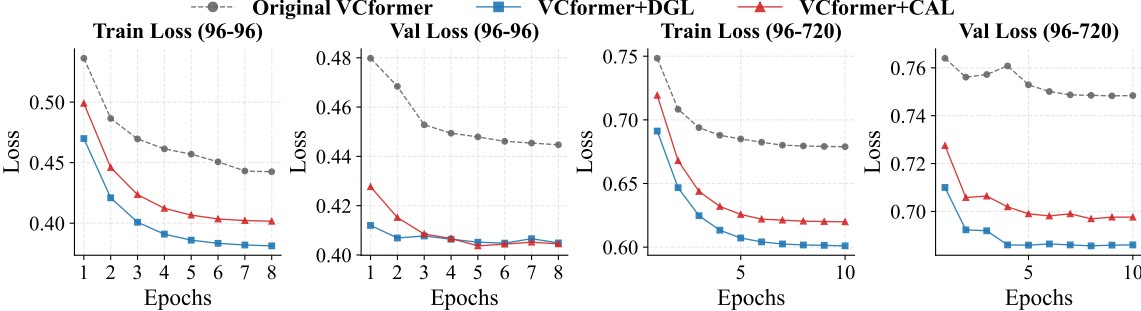

*Figure 21.* Visualization of training and validation loss curves for VCformer on two prediction scenarios (Weather dataset). Although VCformer with DGL outperforms CAL on the validation set on the 96-720 prediction, CAL achieves superior results on the test set, indicating that CAL confers additional robustness.

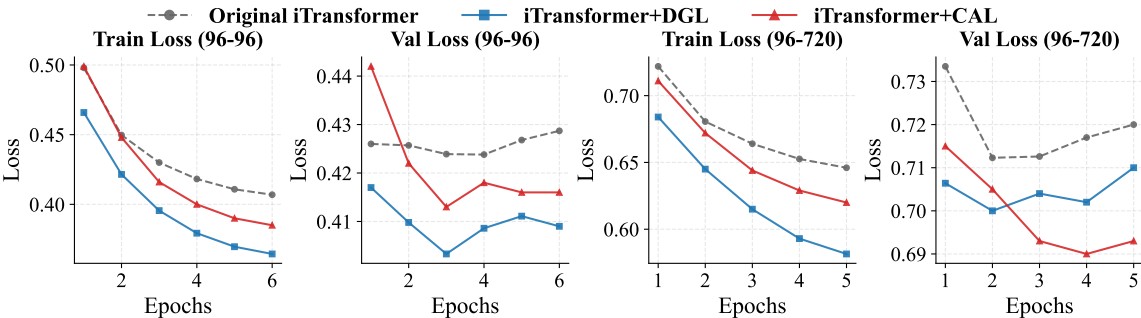

*Figure 22.* Visualization of training and validation loss curves for iTransformer on two prediction scenarios (Weather dataset).

As shown in Figure 23, the loss landscape visualizations demonstrate that DGL and CAL yields a remarkably wide and flat valley (a broader region of low loss around the optimum). Specifically, we observe that incorporating CAL stabilizes the optimization process of DUET and results in clearer gradient directions. And this extreme flatness for two other modified architectures indicates that modeling IVD in deep layers and aligning IVD between shallow and deep layers, effectively smooth the optimization landscape, guiding the model towards a solution with superior robustness and generalization capabilities.

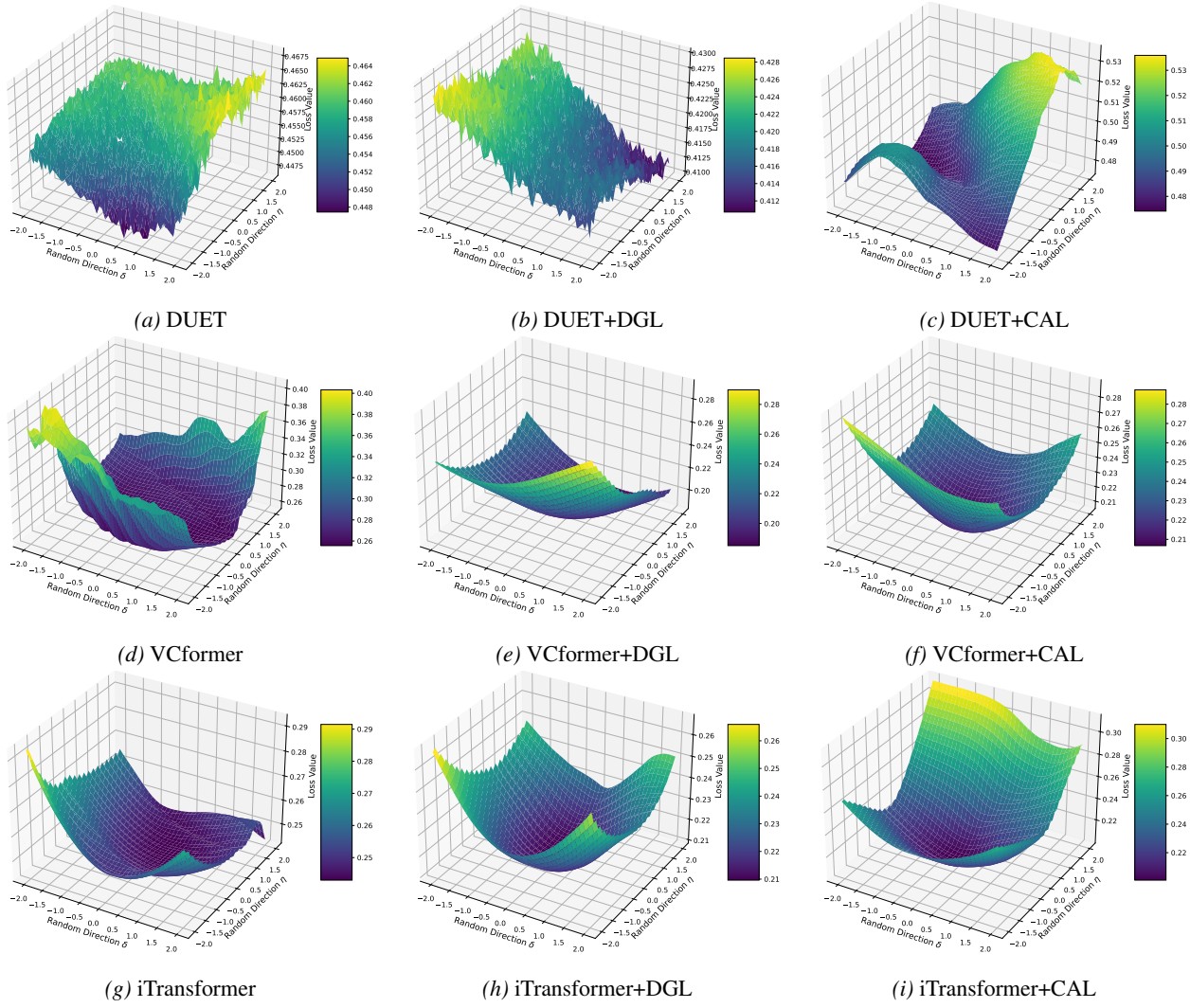

*(a)* DUET        *(b)* DUET+DGL        *(c)* DUET+CAL

*(d)* VCformer        *(e)* VCformer+DGL        *(f)* VCformer+CAL

*(g)* iTransformer        *(h)* iTransformer+DGL        *(i)* iTransformer+CAL

*Figure 23.* Loss landscape visualizations around the converged model parameters on the same batch of Weather test set. The surfaces are plotted along two random direction vectors, $\delta$ and $\eta$, in the parameter space (Li et al., 2018). Due to the high computational cost of generating the entire loss landscape, we utilized a single batch from the test set for this calculation. Consequently, the minimum loss value shown in the figures does not necessarily represent the global optimum. Nevertheless, the smoothness of the surface and the width of the valley serve as reliable indicators of the optimization difficulty and the robustness of the learned model.

## L. Further Analysis of Consistent Inter-Series Dependency Modeling

To further investigate the necessity and effectiveness of modeling inter-variable dependencies in the deeper layers of the network, we conducted additional experiments on model modifications. Specifically, we removed the DGL and CAL modules from our `CGTFra` framework, **retaining only the FMR**. Concurrently, we integrated the proposed dynamically constructed graph structure into the self-attention scores (a method similar to that used in DUET, see Figure 15) by using two fusion strategies: element-wise addition (acting as bias or guidance) and element-wise multiplication (acting as masking). The comparative results are presented in Table 14. The results indicate that merely guiding the self-attention mechanism with dynamic graph information is insufficient to achieve superior performance. **We attribute this to the fact that this approach fails to model inter-variable dependencies in the deeper network layers**, a limitation previously discussed in this paper. **This finding implicitly underscores the necessity and effectiveness of consistently modeling inter-variable dependencies across both the shallow and deep layers of the network architecture**.

*Table 14.* Performance comparison of `CGTFra` and two variants without deep IVD modeling.

| Datasets | | | ETTm1 | | ETTm2 | | ETTh1 | | ETTh2 | | Exchange | | Weather | | ECL | | Solar | | Traffic | |
|---|---|---|---|---|---|---|---|---|---|---|---|---|---|---|---|---|---|---|---|---|
| Metrics | | | MSE | MAE | MSE | MAE | MSE | MAE | MSE | MAE | MSE | MAE | MSE | MAE | MSE | MAE | MSE | MAE | MSE | MAE |
| CGTFra | (original) shallow + deep | 96 | 0.315 | 0.344 | 0.171 | 0.249 | 0.372 | 0.387 | 0.288 | 0.336 | 0.083 | 0.202 | 0.152 | 0.190 | 0.137 | 0.227 | 0.191 | 0.205 | 0.387 | 0.239 |
| | | 192 | 0.366 | 0.372 | 0.238 | 0.293 | 0.425 | 0.418 | 0.364 | 0.384 | 0.173 | 0.296 | 0.203 | 0.239 | 0.155 | 0.243 | 0.218 | 0.225 | 0.417 | 0.249 |
| | | 336 | 0.398 | 0.395 | 0.300 | 0.333 | 0.470 | 0.441 | 0.410 | 0.422 | 0.324 | 0.412 | 0.257 | 0.279 | 0.170 | 0.259 | 0.238 | 0.240 | 0.434 | 0.261 |
| | | 720 | 0.472 | 0.435 | 0.397 | 0.391 | 0.469 | 0.462 | 0.414 | 0.433 | 0.668 | 0.619 | 0.338 | 0.334 | 0.198 | 0.283 | 0.249 | 0.242 | 0.472 | 0.279 |
| | | Avg | 0.388 | 0.386 | 0.277 | 0.316 | 0.434 | 0.427 | 0.369 | 0.394 | 0.312 | 0.382 | 0.238 | 0.260 | 0.165 | 0.253 | 0.224 | 0.228 | 0.427 | 0.257 |
| CGTFra | shallow bias | 96 | 0.319 | 0.346 | 0.177 | 0.253 | 0.372 | 0.388 | 0.296 | 0.340 | 0.086 | 0.205 | 0.159 | 0.195 | 0.142 | 0.230 | 0.193 | 0.207 | 0.395 | 0.245 |
| | | 192 | 0.370 | 0.375 | 0.243 | 0.296 | 0.435 | 0.423 | 0.369 | 0.386 | 0.179 | 0.301 | 0.211 | 0.244 | 0.158 | 0.245 | 0.221 | 0.227 | 0.419 | 0.254 |
| | | 336 | 0.404 | 0.396 | 0.305 | 0.336 | 0.478 | 0.445 | 0.420 | 0.429 | 0.353 | 0.429 | 0.266 | 0.285 | 0.170 | 0.259 | 0.242 | 0.242 | 0.448 | 0.272 |
| | | 720 | 0.501 | 0.445 | 0.411 | 0.401 | 0.487 | 0.473 | 0.426 | 0.438 | 0.797 | 0.675 | 0.345 | 0.339 | 0.201 | 0.285 | 0.248 | 0.242 | 0.485 | 0.284 |
| | | Avg | 0.399 | 0.391 | 0.284 | 0.322 | 0.443 | 0.432 | 0.378 | 0.398 | 0.354 | 0.403 | 0.245 | 0.266 | 0.168 | 0.255 | 0.226 | 0.230 | 0.437 | 0.264 |
| CGTFra | shallow mask | 96 | 0.314 | 0.343 | 0.172 | 0.249 | 0.371 | 0.387 | 0.291 | 0.337 | 0.086 | 0.205 | 0.162 | 0.198 | 0.144 | 0.231 | 0.197 | 0.212 | 0.408 | 0.252 |
| | | 192 | 0.369 | 0.375 | 0.238 | 0.293 | 0.438 | 0.424 | 0.364 | 0.383 | 0.181 | 0.303 | 0.211 | 0.243 | 0.159 | 0.244 | 0.224 | 0.226 | 0.426 | 0.258 |
| | | 336 | 0.414 | 0.401 | 0.302 | 0.335 | 0.484 | 0.446 | 0.424 | 0.427 | 0.338 | 0.421 | 0.266 | 0.284 | 0.174 | 0.261 | 0.246 | 0.242 | 0.449 | 0.270 |
| | | 720 | 0.479 | 0.437 | 0.410 | 0.339 | 0.493 | 0.475 | 0.432 | 0.440 | 1.040 | 0.773 | 0.345 | 0.338 | 0.213 | 0.295 | 0.248 | 0.242 | 0.505 | 0.292 |
| | | Avg | 0.394 | 0.389 | 0.281 | 0.319 | 0.447 | 0.433 | 0.378 | 0.397 | 0.411 | 0.426 | 0.246 | 0.266 | 0.173 | 0.258 | 0.229 | 0.231 | 0.447 | 0.268 |

*Table 15.* Ablation studies on five diverse datasets.

| Part | FMR | DGL | CAL | F | ETTm1 MSE | MAE | ETTh1 MSE | MAE | Weather MSE | MAE | ECL MSE | MAE | Traffic MSE | MAE |
|---|---|---|---|---|---|---|---|---|---|---|---|---|---|---|
| CGTFra | ✓ | ✓ | ✓ | 96 | 0.315 | 0.344 | 0.372 | 0.387 | 0.152 | 0.190 | 0.137 | 0.227 | 0.387 | 0.239 |
| | | | | 192 | 0.366 | 0.372 | 0.425 | 0.418 | 0.203 | 0.239 | 0.155 | 0.243 | 0.417 | 0.249 |
| | | | | 336 | 0.398 | 0.395 | 0.470 | 0.441 | 0.257 | 0.279 | 0.170 | 0.259 | 0.434 | 0.261 |
| | | | | 720 | 0.472 | 0.435 | 0.469 | 0.462 | 0.338 | 0.334 | 0.198 | 0.283 | 0.472 | 0.279 |
| aba1 | ✓ | ✗ | ✗ | 96 | 0.324 | 0.354 | 0.372 | 0.387 | 0.158 | 0.194 | 0.142 | 0.229 | 0.393 | 0.244 |
| | | | | 192 | 0.374 | 0.377 | 0.425 | 0.418 | 0.211 | 0.242 | 0.158 | 0.244 | 0.416 | 0.253 |
| | | | | 336 | 0.407 | 0.401 | 0.485 | 0.448 | 0.266 | 0.285 | 0.174 | 0.263 | 0.437 | 0.264 |
| | | | | 720 | 0.481 | 0.440 | 0.487 | 0.472 | 0.345 | 0.338 | 0.204 | 0.286 | 0.478 | 0.281 |
| aba2 | ✓ | ✓ | ✗ | 96 | 0.310 | 0.350 | 0.373 | 0.386 | 0.157 | 0.195 | 0.138 | 0.228 | 0.393 | 0.242 |
| | | | | 192 | 0.373 | 0.375 | 0.431 | 0.421 | 0.207 | 0.243 | 0.156 | 0.243 | 0.419 | 0.254 |
| | | | | 336 | 0.402 | 0.399 | 0.468 | 0.440 | 0.265 | 0.286 | 0.171 | 0.260 | 0.438 | 0.263 |
| | | | | 720 | 0.470 | 0.434 | 0.475 | 0.462 | 0.340 | 0.338 | 0.207 | 0.291 | 0.469 | 0.278 |
| aba3 | ✗ | ✓ | ✓ | 96 | 0.321 | 0.353 | 0.371 | 0.387 | 0.158 | 0.197 | 0.140 | 0.230 | 0.414 | 0.252 |
| | | | | 192 | 0.371 | 0.375 | 0.429 | 0.421 | 0.209 | 0.244 | 0.155 | 0.244 | 0.425 | 0.249 |
| | | | | 336 | 0.401 | 0.397 | 0.476 | 0.445 | 0.262 | 0.284 | 0.172 | 0.261 | 0.442 | 0.268 |
| | | | | 720 | 0.474 | 0.436 | 0.472 | 0.465 | 0.344 | 0.339 | 0.225 | 0.306 | 0.493 | 0.280 |

# M. Effect of Hyperparameters

To investigate the influence of hyperparameters on `CGTFra`'s prediction performance, we conducted a series of experiments on `CGTFra`'s stacking layers ($L$), the number of heads in the self-attention mechanism, the number of hops in DGL, the type of loss function used in CAL, and the consistency loss weight ($\lambda$) within the loss function. The results are presented in Figure 24. We present the following analysis: (1) **stacking layers ($L$)**: Stacking multiple layers in `CGTFra` enables the model to adapt to datasets of varying complexity, with the learning of multiple feature levels enhancing its representational capacity. Experimental results in ECL indicate that stacking multiple `CGTFra` layers improves performance for shorter prediction horizons (96 and 192), while showing an inverse, negative effect for longer horizons (336 and 720). (2) **the number of heads in the self-attention mechanism**: Multi-head attention allows the model to capture differentiated features and enhances parallelism. Concurrently, in our `CGTFra`, the number of heads influences the granularity of the alignment loss function's calculation. `CGTFra` achieves favorable performance gains when utilizing 4 or 8 heads. (3) **the number of hops in DGL**: While multi-hop propagation can achieve a larger global receptive field, it may also lead to negative effects such as oversmoothing, attenuation of node relevance, and amplified noise. In `CGTFra`, the default number of hops used is 2. We observe that as the number of hops increases, predicting excessively long sequences, such as those of length 720, exhibits significant performance fluctuations. (4) **the type of loss function used in CAL**: To investigate the impact of different similarity measures on `CGTFra`'s performance and the effectiveness of the regularization term, we explore various loss functions as regularizers, including Kullback-Leibler (KL) divergence, Mean Absolute Error (MAE), Mean Squared Error (MSE), and Cosine Similarity. The results indicate that KL divergence, MAE, and MSE yield comparable performance, whereas Cosine Similarity leads to a significant performance degradation. This is likely attributable to Cosine Similarity's exclusive focus on vector direction, disregarding magnitude. Consequently, when evaluating the discrepancy between inter-variate dependencies captured by self-attention and GNNs, it merely promotes directional alignment without encouraging similar scales or absolute values for the tensors. Therefore, Cosine Similarity is unsuitable for quantifying inter-variate dependency differences between shallow and deep layers. (5) **the consistency loss weight ($\lambda$)**: In Equation 10, we use $\lambda$ to control the contribution of CAL. In Figure 24, different values of $\lambda$ most significantly impact performance for scenarios with a prediction length of 720.

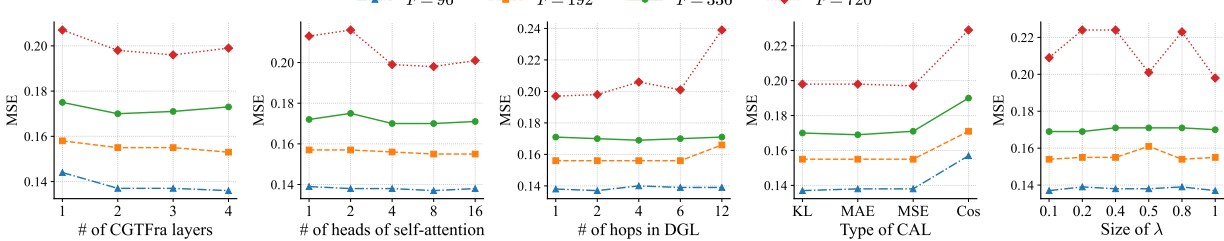

*Figure 24.* Sensitivity analysis of `CGTFra`'s hyperparameters on ECL dataset for forecasting four future lengths $\{96, 192, 336, 720\}$ with fixed input length 96.

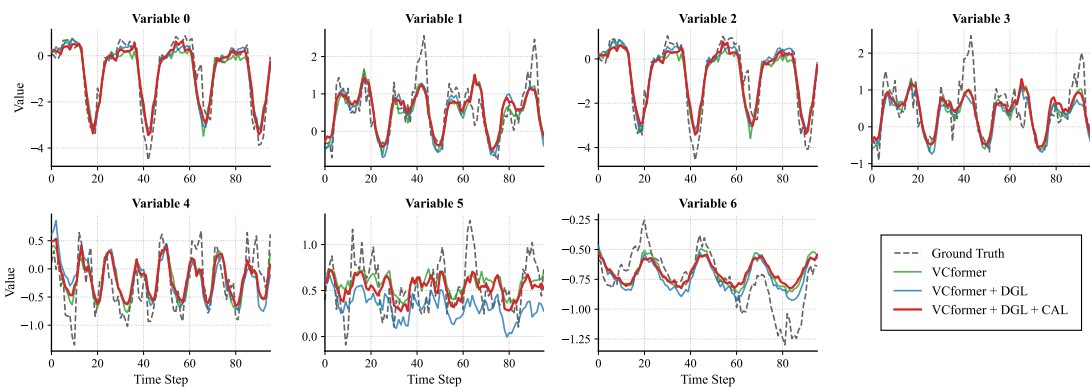

*Figure 25.* Prediction curves for VCformer and variates with our DGL and CAL on ETTh1 dataset.

# N. Analysis of Inter-Series Dependency Modeling

To further evaluate the effectiveness of the DGL and CAL, as shown in Figure 25, we visualized the VCformer's actual prediction curves for 7 variables of ETTh1 in Figure 10. We observe that VCformer, when embedded with DGL and CAL, achieves superior prediction accuracy in most cases, indicating the efficacy of modeling IVD simultaneously in both shallow and deep network layers. Furthermore, we note that for variable 5, the introduction of DGL alone leads to worse prediction. However, with the consistency constraint of CAL, thanks to bidirectionally validated inter-variate dependencies, significantly improved prediction capabilities are obtained, demonstrating that the introduction of CAL effectively promotes the model's optimization of deep-layer feature embeddings. Furthermore, comprehensive evaluation metrics are provided in Appendix K (Table 13) to validate the effectiveness of DGL and CAL.

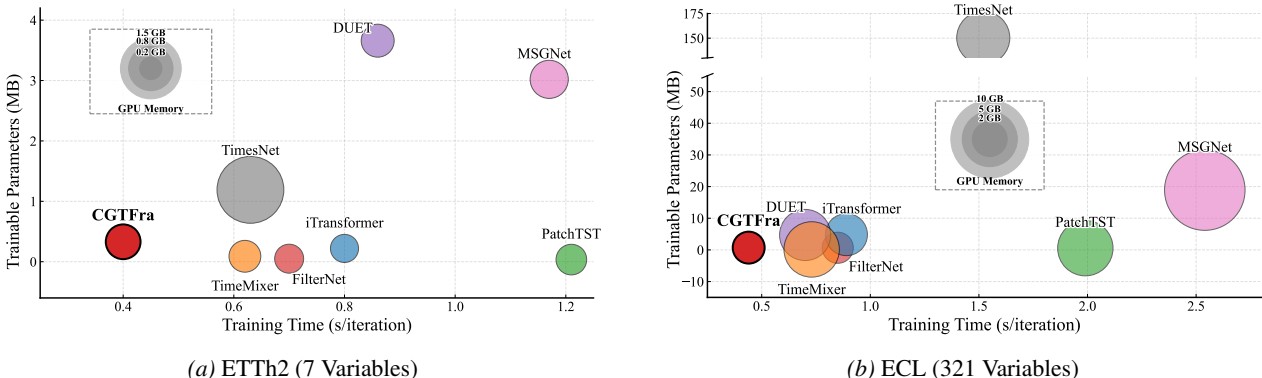

*(a)* ETTh2 (7 Variables)      *(b)* ECL (321 Variables)

*Figure 26.* Computation effectiveness analysis for eight methods on ETTh2 and ECL (Input 96-Predict 96). The size of the circle indicates the GPU memory footprint. For fair comparison, all batch sizes are set to 32.

# O. Efficiency Comparison

We fairly compare the training time, running GPU memory, and trainable parameter count against 7 sota methods in Figure 26. Benefiting from the computational efficiency of DGL in capturing variable dependencies and the performance gains of CAL without introducing additional learnable parameters, CGTFra achieves strong performance and computational efficiency with relatively less trainbale parameters. Compared to another sota method-DUET (Qiu et al., 2025), known for its high run-time efficiency, CGTFra reduces GPU memory usage by 61% and demonstrates a training speed improvement of approximately 42.86% on the complex ECL dataset, indicating the high effectiveness and efficiency of CGTFra.

# P. Limitations

Although our study significantly enhances the performance of existing studies by introducing deep inter-variate dependency modeling (DGL) within the Variate Transformer and further optimizing inter-variate associations across both deep and shallow layers through explicit dependency constraints (CAL), we still observe that Variable Transformers incorporating DGL and CAL, such as DUET, and iTransformer—exhibit limited improvements or even slight performance degradation on datasets like Solar and Traffic (Our current evaluation utilizes a **identical hyperparameter setting** to demonstrate generalization. And honestly, achieving improved performance simultaneously across all datasets with distinct distributions remains a significant challenge in this field). In our situation, a plausible explanation is that as the number of variables ($N$) increases, the probability of spurious correlations between any two variables rises dramatically. The self-attention mechanism, designed to find relationships within an $N{\times}N$ matrix, is compelled to assign attention weights across all variable pairs. In such a high-dimensional space, these weights are more likely to reflect coincidental noise within a sample rather than genuine, stable dependencies. Consequently, when CAL is applied, it forces the adjacency matrix $A$ learned by DGL to filter this noisy Correlation Map (MCM), greatly hinders effective compression. The GNN is thus coerced into encoding numerous useless or even erroneous connections in its graph structure, which undermines its ability to perform effective information propagation.

We will improve upon this in future work by proposing a more general method for modeling correlation constraint between deep and shallow layers.

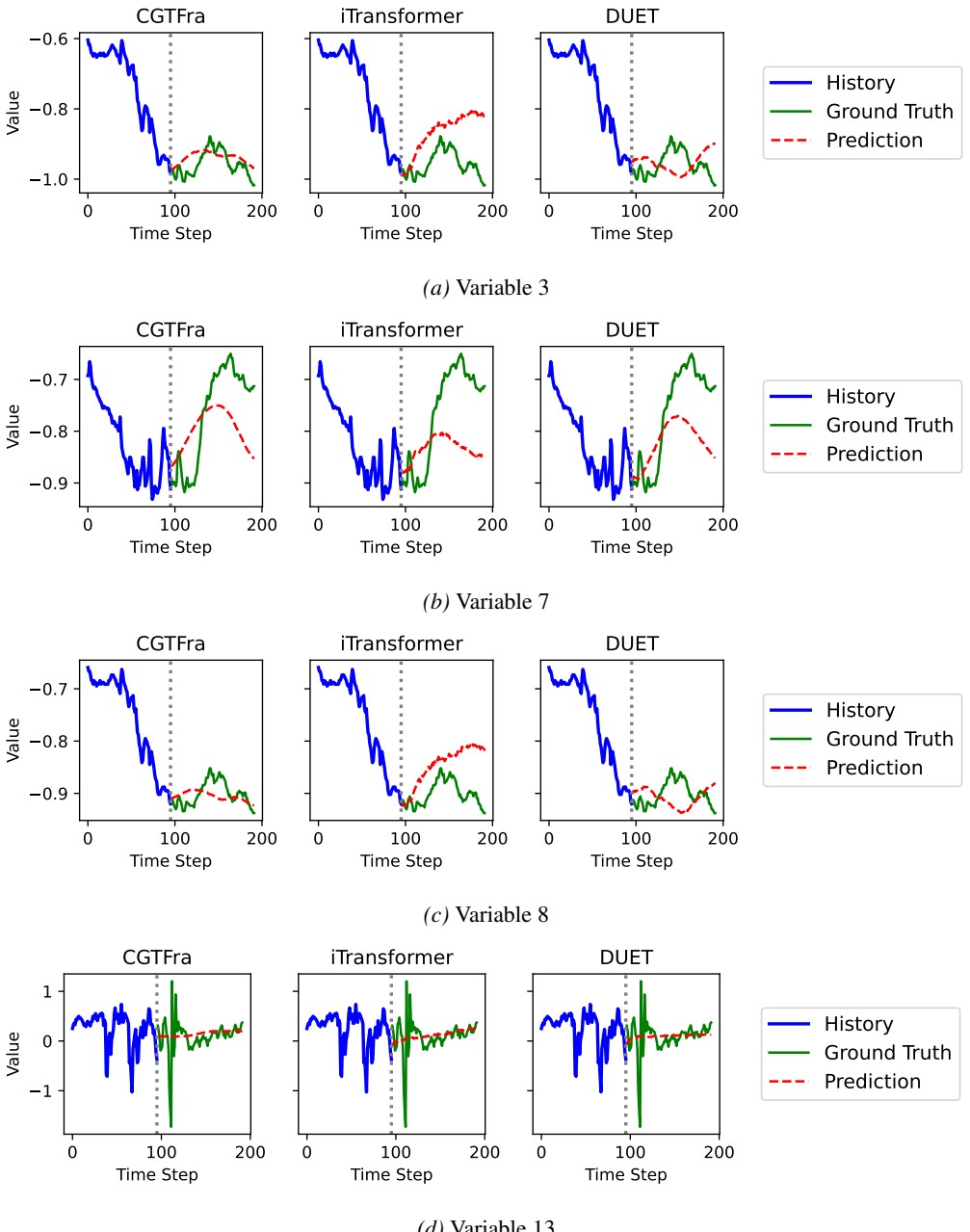

*Figure 27.* Actual prediction curves for three models capturing IVD on Variables 3, 7, 8, and 13 of Weather dataset (Input 96-Predict 96).

