# OpenReview forum: "Robust Inter-Series Dependency Modeling for Time Series Forecasting via Information-Theoretic Alignment"
_ICML.cc/2026/Conference — ICML 2026 regular_

### Official Review · Reviewer_dHNW · 2026-03-03

**Soundness:** 3
**Presentation:** 4
**Significance:** 3
**Originality:** 4
**Overall Recommendation:** 5
**Confidence:** 5

**Summary:**

This paper addresses, from a generalization perspective, the optimization difficulties and spurious dependency modeling caused by inconsistencies in how existing variate-based Transformer models capture inter-series dependencies. To this end, the authors propose CGTFra, a unified consistency-aware dependency modeling framework that consists of a frequency-domain enhancement module (FMR), a deep graph learning framework (DGL), and a consistency alignment loss (CAL). CGTFra achieves state-of-the-art performance across multiple datasets, particularly on complex datasets with a large number of variables, such as ECL, Solar, and Traffic.
Furthermore, the authors validate the effectiveness of the three core components of CGTFra not only within existing variate Transformer frameworks (including DUET, VCformer, and iTransformer), but also on non-Transformer architectures (as for FMR). The proposed method demonstrates strong performance across three key dimensions: generality, computational efficiency, and predictive accuracy.

**Compliance With Llm Reviewing Policy:**

Affirmed.

**Final Justification:**

This paper proposes a consistency-driven framework for modeling inter-variable dependencies, and validates the necessity of consistency from multiple perspectives, including theoretical analysis, generality, comprehensive investigation of inter-variable dependency modeling, and model optimization. Although the exposition in this paper may require further refinement, the authors’ rebuttal has adequately addressed my comments. As I have been continuously focusing on and working on inter-variate dependency modeling of multivariate time series, therefore, I recommend its acceptance.

**Key Questions For Authors:**

1. How do the authors conceptualize intra-series dependency modeling? Does the proposed method (see Figure 2(c)) implicitly model intra-series dependencies in deep layers?

2. For non-variate Transformer approaches, do the authors believe that inter-series dependencies should also be modeled in a consistent manner? What are the authors’ further thoughts on this issue?

3. Beyond adopting graph neural network–based modeling in deep layers, are there alternative approaches that could achieve similar objectives?

**Limitations:**

There are some inconsistencies in font usage across the main text and appendix (e.g., CGTFra, FMR), which should be corrected for presentation consistency.

**Strengths And Weaknesses:**

Strength
1. Starting from the inconsistency in inter-series dependency modeling in existing variate Transformers, the authors propose a general and unified consistency-aware dependency modeling framework. This perspective may provide valuable insights for future Transformer-based time-series analysis methods.

2. The necessity of consistency-aware dependency modeling is supported by extensive empirical results, and the authors further provide a theoretical upper bound for the proposed framework based on information bottleneck theory, aligning well with the ICML style of strong theoretical grounding combined with strong empirical performance.

3. The paper presents a relatively systematic and comprehensive analysis of the motivation and principles behind inter-series dependency modeling, offering meaningful insights. I believe this work deserves to be seen by the time-series research community.

4. The paper is logically structured, experimentally thorough, and accompanied by detailed reproducibility scripts and tutorials.

Weakness
1. If multiple self-attention layers are stacked within the Transformer to model inter-series dependencies, would consistency-aware dependency modeling still be necessary? This scenario is not discussed in the paper.

2. The paper appears to focus primarily on inter-series dependency modeling, while the modeling of intra-series dependencies is not explicitly analyzed. DUET is a representative method that jointly models intra-series and inter-series dependencies; a more detailed comparison and discussion between CGTFra and such methods would further highlight the effectiveness of the proposed approach.

3. The authors are encouraged to include t-SNE visualizations of the ground-truth representations as a reference for comparison.

4. The introduction contains multiple references to content presented in the appendix, which may reduce readability and cause frustration for readers.

5. Although the effectiveness of FMR is empirically validated, it is not a core contribution of the paper. Accordingly, the discussion of FMR in the introduction could be simplified.

6. The figure captions should be further refined and improved.

---

> ### Author Rebuttal · Authors · 2026-03-28
>
> Many thanks to you for providing an in-depth and constructive review, which helped us significantly improve the quality of our submission.
>
> **(1) Weakness 1**
>
> We appreciate your insightful perspective. Indeed, stacking multiple self-attention layers serves as an intuitive approach to modeling IVD. (Our analysis in Line 817) “iTransformer presented insightful experiments (see Table 3 in iTransformer paper) where they replaced the FFN with a self-attention layer, essentially constructing a Transformer with two self-attention layers. The experimental results indicated that simply stacking multiple self-attention layers did not facilitate the learning of correct inter-variate dependencies and temporal patterns.” To comprehensively demonstrate that merely stacking self-attention layers fails to establish robust dependencies, we evaluated CGTFra by stacking varying numbers of self-attention layers while removing both the DGL and CAL. The comparative results are in https://anonymous.4open.science/r/CGTFra/StackMHSA.pdf. These findings indicate that **stacking multiple self-attention layers leads to substantial performance degradation**—primarily because, as analyzed in our manuscript, self-attention tends to capture dense and noisy correlations.
>
> **(2) Weakness 2 & Question 1: Intra-series dependency modeling**
>
> Temporal dependencies are explicitly captured within both the FMR and DGL in our CGTFra. Specifically, **FMR (operating in a channel-independent manner) conducts global temporal modeling in the frequency domain**. DGL, on the other hand, utilizes a two-layer linear architecture: the first layer aggregates multi-hop neighborhood information, while the second layer further extracts temporal dependencies.
> We acknowledge that DUET is an excellent model designed to capture both inter- and intra-variable dependencies. Architecturally, DUET employs a dual-branch structure (modeling temporal and IVD concurrently) as the input to Variate Transformer. In contrast, CGTFra adopts a different modeling paradigm: it first enhances the input via FMR, and subsequently constructs a consistent IVD framework. Furthermore, regarding intra-variable modeling, as rigorously analyzed in Line 86 and Appendix G, **DUET applies the learned IVD as a mask for the self-attention layer. Consequently, it still suffers from the issue of inconsistent IVD modeling**. Our CGTFra is specifically proposed to resolve this limitation.
>
> **(3) Weakness 3: T-SNE visualization**
>
> We appreciate your constructive suggestion. To clarify, Figs. 3, 8, and 16-18 present the t-SNE visualizations of the **latent embeddings** obtained after the FFN or DGL modules and the residual connections. We completely agree with your insight that providing a t-SNE visualization of the ground-truth data serves as a crucial reference to evaluate the plausibility of the modeled IVD. Accordingly, we have supplemented the ground-truth t-SNE visualization for 1,500 test samples from the ETTh1 dataset: https://anonymous.4open.science/r/CGTFra/TSNEGroundTruth.pdf. The distribution aligns closely with the latent embeddings refined by DGL and CAL (see Figs. 3, 16, 17, and 18, notably Fig. 17).
>
> **(4) Q2: Extension to non-variate Transformer framework**
>
> We appreciate this profound and insightful question. We believe that modeling consistent IVD **remains essential even for non-variate Transformer frameworks**. We offer the following perspective: We conceptualize IVD modeling as a form of implicit structural constraint imposed on the network. **By capturing these correlations, the latent embeddings receive valuable directional guidance during optimization**. This mechanism potentially steers the model toward more stable gradient descent trajectories (a phenomenon we have empirically validated within the variate Transformer framework, as shown in Figures 19–22). We are truly grateful for the inspiration your comment has provided, and we will incorporate this discussion into the Future Work section of the revised manuscript.
>
> **(5) Q3: Expand thinking**
>
> In the CGTFra, our consistency paradigm is formulated based on the theoretical similarities and differences between GNNs and self-attention layers in modeling IVD. Regarding potential alternatives to GNNs for capturing these dependencies in deep layers, we hypothesize that **metric learning and kernelization methods** could serve as promising exploratory approaches. This is because such techniques can explicitly model IVD **by directly quantifying the pairwise similarities of latent embeddings**. We intend to conduct further empirical validation on this hypothesis in our future work.
>
> **(6) Weakness 4,5,6, & Limitation: Presentation**
> We will carefully refine our manuscript based on your constructive suggestions.
>
> We hope that our point-by-point responses have adequately addressed your comments and concerns. In light of these clarifications and improvements, we sincerely hope that you might consider re-evaluating our manuscript.

---

> > ### Author Rebuttal · Reviewer_dHNW · 2026-04-01
> >
> > Thanks for your clarifications, which has addressed my concerns, and I will increase my score.

---

> > > ### Author Response · Authors · 2026-04-03
> > >
> > > We appreciate your acknowledgement. We will meticulously revise the manuscript based on the feedback you previously provided.

---

### Official Review · Reviewer_oZGg · 2026-03-03

**Soundness:** 3
**Presentation:** 3
**Significance:** 3
**Originality:** 3
**Overall Recommendation:** 4
**Confidence:** 3

**Summary:**

CGTFra is a Graph Transformer framework that enhances multivariate time series forecasting by modeling inter-variate dependencies across both shallow and deep network layers,. It integrates frequency-domain masking to preserve periodicity and a consistency-constrained alignment to synchronize dependencies, achieving state-of-the-art performance.

**Compliance With Llm Reviewing Policy:**

Affirmed.

**Final Justification:**

The authors have done a good job responding to my initial questions. I maintain an overall recommendation of 4 and confidence at 3.

**Key Questions For Authors:**

•	What is the motivation for choosing the Discrete Cosine Transform (DCT) for time-to-frequency domain transformation? Were alternative transformations considered, and how would they compare?

•	It appears that temporal relationships are primarily modeled within the FMR component. Is this understanding correct? If so, is this mechanism sufficient to capture complex temporal interactions, and how might it be further strengthened?

•	Could Section 3.2 be expanded to provide a clearer description of tensor dimensions, intermediate representations, and mathematical operations?

**Limitations:**

Yes

**Strengths And Weaknesses:**

Strengths:

•	The paper is well written and clearly structured, with the exception of Section 3.2, which could benefit from further clarification.

•	The method is evaluated on a substantial number of datasets and compared against strong and relevant baselines.

•	Clear reasoning and motivation are provided for each major design choice.

•	The proposed method is modular and can be integrated as a plug-and-play component within other frameworks.

Weaknesses:

•	Section 3.2 requires further elaboration and more detailed explanation to improve clarity and reproducibility.

---

> ### Author Rebuttal · Authors · 2026-03-27
>
> We sincerely appreciate your valuable comments, which have helped us improve the quality of our manuscript.
>
> **(1) Section 3.2: Detailed explanation**
>
> Thank you for your valuable suggestion. We will further elaborate on the details of Section3.2. Unlike Sageformer and MSGNet (rely only on self-learned parameters), our node updating strategy utilizes the output  ($\mathbf{H}\_{sa}\in \mathbb{R}^{N\times D}$) of the self-attention layer as the **sample-level** input for node updates, along with **globally updated** node representations $\Theta$ to construct a dynamic graph structure **based on the theoretical analysis of similarities and distinctions between GNN and self-attention layer in modeling IVD** (Appendix D and E for details). Specifically, in the $l$-th layer of CGTFra, we concatenate $\mathbf{H}\_{sa}^{l}$ with the static node embeddings by $\textrm{Concat}(\mathbf{H}\_{sa}^{l},\Theta^l\_{1}) \in \mathbb{R}^{N\times (D+nd)}$ and $\textrm{Linear}(\textrm{Concat}(\mathbf{H}\_{sa}^{l},\Theta^l\_{1})) \in \mathbb{R}^{N\times 1}$, and apply activations to generate gating weights $\mathbf{W}\_{gating}\in \mathbb{R}^{N\times 1}$, capturing the critical information regarding dynamic variations in IVD present within $\mathbf{H}\_{sa}$. Then, we perform a Hadamard product between this gate and $\textrm{Linear}(\mathbf{H}\_{sa})\in \mathbb{R}^{N\times nd}$ to extract input-adaptive node features. These features are then combined with the globally updated node representations via a residual connection to form the final node embeddings $\Theta^l\_{1}\in \mathbb{R}^{N\times nd}$ ($\Theta^l\_{2}$ employs the same updating strategy but without parameter sharing). Ultimately, the dynamic graph matrix is computed via $\mathbf{A}^{l}=\textrm{Softmax}(\textrm{ReLU}(\Theta\_{1}^{l} \cdot (\Theta\_{2}^{l})^{\top})) \in \mathbb{R}^{N\times N}$, which is utilized to execute message aggregation.
>
> **(2) DCT analysis**
>
> Prevalent time series analysis methods typically employ FFT for frequency-domain transformations, such as DUET (KDD, 2025), TSLANet (ICML, 2024), and FilterNet (NeurIPS, 2024). The basis functions of FFT are complex exponentials ($e^{-i \frac{2\pi}{T} k t}$), which generally necessitates separate modeling for the real and imaginary components. In contrast, the DCT utilizes real-valued cosine functions ($\phi_k(t) = \cos\left(\frac{\pi}{T} \left(t + \frac{1}{2}\right) k \right)$) as its orthogonal basis. **Consequently, achieving equivalent functionality via DCT incurs a lower computational overhead**. To further demonstrate the effectiveness of DCT, we conducted comparative experiments by substituting DCT with FFT (including real FFT (rFFT) and Full complex FFT) and the Discrete Wavelet Transform (DWT), and then the same resampling mechanism is applied. The comparative results, in https://anonymous.4open.science/r/CGTFra/FMRSpectral.pdf, validate the efficacy of applying DCT. Code has been updated in our anonymous repository.
>
> **(3) Temporal Interactions**
>
> We model temporal dependencies within two distinct areas: FMR and DGL (with two linear layer for aggregating multi-hop neighborhood information and further extracting temporal features). Specifically, FMR captures temporal correlations by modeling the **global characteristics** of frequency-domain components, **which indeed serves as the primary temporal modeling mechanism in CGTFra**. As detailed in Appendix B, our analysis verifies that FMR preserves and enhances periodic features.
> In our generality evaluations (see Table 2), we find that integrating FMR yields performance gains across most datasets when applied to multiple baselines, including iTransformer (ICLR 2024), CASA (IJCAI 2025), FilterNet (NeurIPS 2024), and VCformer (IJCAI 2024). Simultaneously, we also noticed that FMR brings only marginal gains, or occasionally slight adverse effects, on the Traffic. We hypothesize this is because FMR may excessively suppress periodicity in datasets with highly regular patterns (Figure 11(a), Traffic with clear periodicity). Therefore, **following your suggestion, we have introduced an enhancement to our FMR to address this potential issue**. We first analyze a single batch from the training set to capture relatively stable periodicities. Based on that, **we apply a larger, fixed weight (e.g., 1.0, bypassing the mask) to the top-k dominant frequency components, while the remaining frequencies are still processed by a learnable mask**. Through this approach (named FMR+), we obtain gains for both iTransformer and CASA (compared to old FMR). Results are at: https://anonymous.4open.science/r/CGTFra/FMRStrengthen.pdf . FMR+ code has been updated in our anonymous repository.
>
> We hope that these responses have moved us closer to a common ground. We are always available to address any further comments promptly and sincerely look forward to your reconsideration of our manuscript.

---

> > ### Author Rebuttal · Reviewer_oZGg · 2026-04-01
> >
> > I want to thank the authors for their response. I will maintain my positive rating.

---

> > > ### Author Response · Authors · 2026-04-03
> > >
> > > Thank you for your acknowledgement. We will carefully revise the manuscript based on your previous comments.

---

### Official Review · Reviewer_P9RR · 2026-03-12

**Soundness:** 2
**Presentation:** 3
**Significance:** 2
**Originality:** 3
**Overall Recommendation:** 3
**Confidence:** 4

**Summary:**

This paper argues that existing Variate Transformer models for multivariate time series forecasting (MTSF) suffer from two structural limitations: (1) over-reliance on timestamp embeddings for input representation, and (2) a systematic neglect of inter-variate dependencies (IVD) in the deep feed-forward network (FFN) layers, despite modeling such dependencies in the shallow self-attention layers. The authors characterize the latter as an IVD inconsistency problem and claim it leads to spurious correlation capture and degraded out-of-distribution (OOD) generalization.

To address these, CGTFra (Consistent Graph Transformer Framework) is proposed, consisting of three components:

- **Frequency-domain Masking and Resampling (FMR)**: applies Discrete Cosine Transform (DCT) to the input, learns variable-specific frequency masks via softplus-activated parameters, performs linear resampling in the frequency domain, and reconstructs the signal via inverse DCT. The output $H^{extend} \in R^{N \times D}$, where each variable token $H^{extend}[n, :] \in R^{1 \times D}$, serves as a timestamp-free replacement for the original input embedding.

- **Dynamic Graph Learning (DGL)**: replaces the FFN sublayer in each Transformer block. It takes the self-attention output $H_{sa}^l$ and learnable static node embeddings $\Theta_1^l, \Theta_2^l \in R^{N \times nd}$, computes an adaptive gating weight $W_{gating}$ to produce dynamically updated node embeddings, and constructs a layer-wise adjacency matrix $A^l = Softmax(ReLU(\Theta_1^l \cdot (\Theta_2^l)^T))$. A multi-hop GCN then aggregates neighborhood IVD information: $Y_{out}^{DGL} = MLP(GCN(H_{sa}, A))$.

- **Consistency Alignment Loss (CAL)**: enforces representational consistency between the multivariate correlation map (MCM) derived from self-attention scores and the graph adjacency $A^l$ derived from DGL, via KL divergence summed across all $L$ layers and $N^2$ variable pairs: $L_{align} = \sum_{l=1}^{L} KL(P_l \| Q_l)$. The total loss is $L_{total} = L_{MAE} + \lambda L_{align}$, grounded in the Variational Information Bottleneck (VIB) principle.

The key claims are:

1. Timestamp embeddings are often harmful and FMR serves as a more effective universal replacement for input representation.
2. Deep-layer IVD modeling via DGL is an overlooked component in existing Variate Transformers, and the performance gains from DGL underscore the importance of this neglected aspect.
3. CAL is the first consistency constraint applied to IVD learned jointly by self-attention and deep graph learning, providing a theoretically grounded upper bound on mutual information $I(Z; X)$.

Experiments are conducted on 13 real-world datasets including ETT (4 subsets), Weather, Exchange, ECL, Solar-Energy, Traffic, PEMS03/04/07/08, with fixed input length $T = 96$ and forecasting horizons $F \in \{96, 192, 336, 720\}$ for long-term and $F \in \{12, 24, 48, 96\}$ for short-term tasks. Thirteen baselines are compared, spanning inter-series dependency models (DUET, Soatten, VCformer, iTransformer, MSGNet) and intra-series models (PatchTST, FilterNet, RLinear, TiDE, DLinear, TimePro).

**Compliance With Llm Reviewing Policy:**

Affirmed.

**Final Justification:**

Thank you for the detailed rebuttal and supplementary materials. I have carefully reviewed all provided materials and acknowledge the authors' genuine effort.

**W1: FFN and IVD Modeling**

I acknowledge that the FFN in iTransformer-style Variate Transformers operates as a point-wise MLP independently applied to each variate token, performing no explicit cross-variate computation. The reframing distinguishing "passive transmission via residual connection" from "active refinement of IVD structure" is more precise, and I appreciate the commitment to revise lines 80-81.

However, the causal consequence remains insufficiently supported. The CKA value of $1 - \text{CKA} = 0.9934$ is consistent with the residual formulation $H = H_{sa} + \text{FFN}(H_{sa})$ structurally preserving inter-variate information. The authors counter that $\Delta(1-\text{CKA}) = 0.0117$ may reflect meaningful structural refinement by DGL. I accept both readings are plausible and CKA alone cannot resolve this. However, the threshold of 0.0117 for "meaningful change" is not grounded in a pre-established criterion. If 0.0117 is sufficient to claim meaningful structural change by DGL, it is unclear why 0.9934 does not simultaneously constitute strong evidence for representational preservation. The two interpretations cannot coexist without a principled framework, and none is provided. The T-SNE visualizations show DGL learns different representations than FFN, but do not establish that FFN discards IVD.

**W2: Distribution Shift Claim**

The commitment to remove "OOD" and retain only "distribution shifts" is a meaningful revision. I accept chronological splitting as a standard temporal generalization evaluation and no longer require a dedicated OOD benchmark. I ask only that the revised manuscript avoid implying severe generalization degradation without direct experimental support.

**W4 and W5: IVD Capture and Forecasting Performance**

This is my central remaining concern. The paper's argumentative chain is:

$$\text{FFN passive} \rightarrow \text{inconsistent IVD modeling} \rightarrow \text{spurious correlations} \rightarrow \text{DGL+CAL fix} \rightarrow \text{consistent IVD capture} \rightarrow \text{better performance}$$

The core claim concerns the inconsistency between shallow-layer IVD modeling via self-attention and the absence of deep-layer IVD modeling in FFN. DGL replaces FFN with explicit deep-layer IVD modeling, and CAL enforces consistency between shallow and deep IVD representations. My concern lies in the final link: that consistent IVD capture is the primary driver of performance gains. Two patterns challenge this.

Regarding the Weather dataset: Section 4.6 states "without the CAL constraint, neither the self-attention layer nor the DGL success to capture critical dependencies." Yet Tables 7 and 15 show CGTFra without CAL achieves $\text{MSE} = 0.157$ and $\text{MAE} = 0.195$, outperforming all existing methods. The rebuttal does not explain what mechanism enables SOTA performance when both IVD mechanisms are described as failing. The most natural interpretation is that FMR's input enhancement and DGL's temporal feature extraction contribute substantially to performance independent of IVD capture quality, which weakens the IVD-to-performance causal story.

Regarding the Traffic dataset: the authors acknowledge near-zero IVD weights even with CAL applied. Their explanation that Traffic's temporal periodicity allows other components to compensate is plausible, and I accept that IVD modeling for 862 variables is a recognized open challenge. However, the unified results show iTransformer ($\text{MSE} = 0.422$) outperforming CGTFra ($\text{MSE} = 0.427$) on Traffic. Combined with near-zero IVD weights, this suggests DGL operates as a temporal feature extractor rather than an IVD modeler on this dataset, raising questions about the generality of the deep-layer IVD modeling motivation.

These patterns do not invalidate the paper's contributions. The plug-and-play improvements in Table 2, smoother loss landscapes in Figures 6 and 22, and the VIB-grounded framework are genuine contributions. What the patterns suggest is that the core causal chain — consistent IVD capture as the primary driver — is not consistently supported across these two key datasets.

**Overall Assessment**

The empirical contributions of CGTFra are substantial and the plug-and-play design is practically valuable. However, my concern about the alignment between the claimed problem, the proposed mechanism, and the experimental evidence remains unresolved. I maintain my Overall Recommendation of 3 and Confidence of 4.

**Key Questions For Authors:**

- **Q1.** (W1) What direct evidence supports the claim that FFN layers discard the IVD encoded in $H_{sa}$? Given that the standard residual connection $H = H_{sa} + FFN(H_{sa})$ structurally preserves $H_{sa}$'s inter-variate information regardless of the FFN's own computation, what mechanism causes IVD to be genuinely lost after the FFN sublayer? Probing experiments or representational similarity analysis would be needed to substantiate this foundational premise.

- **Q2.** (W2) The paper claims that IVD inconsistency "results in capturing spurious correlations, severely compromising generalization capabilities under distribution shifts (OOD)." However, the experimental design follows a standard in-distribution temporal split, which does not constitute an OOD evaluation. Can the authors provide a dedicated distribution-shift benchmark or out-of-domain evaluation, or cite prior work that establishes spurious correlation capture as a specific failure mode of Variate Transformers?

- **Q3.** (W3) FMR is presented as a replacement for timestamp embeddings, but timestamp embeddings encode temporal positional information such as periodicity and seasonality, while FMR operates as a frequency-domain feature enhancement module. What is the functional equivalence between the two? Specifically, do the learned frequency masks $\mathcal{M}$ capture representations that correspond to what timestamps encode, and is there any analysis showing that FMR recovers the positional structure that timestamp removal loses? Additionally, can the authors provide a controlled three-way ablation among (a) the original backbone with timestamps, (b) timestamps simply removed without FMR, and (c) FMR substituted, to properly attribute which factor drives the observed performance gains?

- **Q4.** (W4) The claim that "GNNs excel at capturing differentiated indirect and sparse dependencies" relative to global self-attention is supported solely by Figure 10, which analyzes a single dataset (ETTh1). Is there evidence of this advantage on other datasets with fundamentally different dependency structures, such as the densely connected PEMS sensor networks or the weakly coupled Exchange series? What justifies generalizing this claim across all 13 experimental datasets?

- **Q5.** (W5) Tables 15 and 7 report that CGTFra without CAL achieves MSE 0.157 and MAE 0.195 on the Weather test set, outperforming all existing methods. How do the authors reconcile this with the claim in Section 4.6 that without CAL, "neither the self-attention layer nor the DGL success to capture critical dependencies"? What specific mechanism explains the co-existence of critical dependency capture failure and state-of-the-art forecasting performance in the same model?

- **Q6.** (W6) Table 1 reports baseline results taken from official published reports, while Table 2 reproduces results under the authors' experimental environment. Can the authors provide a unified experimental comparison in which all directly competing baselines (at minimum DUET, iTransformer, and VCformer) are reproduced under the same environment as Table 2, so that performance differences reflect methodological rather than environmental factors?

**Limitations:**

The paper does not discuss several limitations that are relevant to understanding the scope of CGTFra's applicability.

(W3) First, regarding FMR: the paper does not explain how FMR functionally replaces the information provided by timestamp embeddings. Timestamp embeddings encode temporal positional structure such as periodicity and seasonality, and the only justification given for treating FMR as a replacement is that "the frequency domain introduces a powerful inductive bias of a global receptive field." This describes a property of FMR itself but establishes no functional equivalence with what timestamps provide. Whether the learned masks $\mathcal{M}$ capture anything resembling temporal positional information is never analyzed. Beyond this, the claim that timestamp embeddings are harmful is not substantiated by a direct ablation. Without a three-way comparison separating (a) timestamp removal only, (b) FMR addition only, and (c) both combined, the contributions of each factor remain entangled, and the design choice to replace timestamps with FMR is not fully justified.

(W1, W2) Second, regarding DGL: the multi-hop GCN introduces additional trainable parameters ($\Theta_1^l, \Theta_2^l \in R^{N \times nd}$ per layer) and additional computation relative to the FFN it replaces. No analysis of parameter count, computational complexity (FLOPs), or inference latency is provided. For deployment in real-world applications where inference cost matters, this omission limits the practical assessment of CGTFra.

(W5) Third, regarding CAL: the formulation sums KL divergence over all $N^2$ variable pairs per layer, which scales quadratically with the number of variates. On large-variable datasets such as Traffic or the PEMS benchmarks, this computational cost is substantial, yet no scalability analysis is presented. It is unclear whether CAL remains computationally feasible as $N$ grows.

(W1, W2, W4) Fourth, the paper's core structural claims -- that FFNs discard IVD, that self-attention captures spurious correlations, and that GNNs outperform attention for indirect dependencies -- are argued qualitatively but not quantitatively measured. Mechanistic interpretability tools such as attention entropy analysis, Centered Kernel Alignment, or probing classifiers would be required to substantiate these structural claims beyond the level of performance numbers.

(W5) Fifth, regarding the consistency constraint in CAL: whether forcing alignment between MCM and $A^l$ is always beneficial, or whether it can over-constrain the model when the two mechanisms should specialize differently, is not analyzed. The observation in Section 4.6 that without CAL, DGL captures the correlation between variables 3 and 7 while self-attention does not, suggests that the two mechanisms possess genuinely distinct inductive biases. It is possible that the forced KL alignment suppresses this complementarity in some settings, but the paper provides no analysis of this potential downside.

**Strengths And Weaknesses:**

**Strengths**

- **S1.** The plug-and-play design of DGL and CAL has practical merit. Table 2 demonstrates consistent performance improvements when integrating these components into five diverse existing architectures (DUET, iTransformer, VCformer, FilterNet, CASA) without architectural redesign. The gains are particularly pronounced on ECL and Traffic, datasets with large variable counts where IVD modeling poses a significant challenge for existing methods, as also noted in Section 4.3.

- **S2.** Grounding CAL in the VIB principle provides a theoretically principled motivation for aligning IVD representations across shallow and deep layers. Framing $L_{align}$ as an upper bound on $I(Z; X)$ and $L_{MAE}$ as maximizing $I(Z; Y)$ within a single unified objective is a conceptually coherent and novel contribution, with the derivation provided in Appendix F.

- **S3.** The experimental scope is broad, covering 13 datasets across diverse domains and variable counts for both long-term and short-term forecasting. The additional evaluation on variants of CGTFra across four architectures in the iTransformer family (iFlashformer, iFlowformer, iInformer, iReformer) in Table 12 further substantiates the generalizability of DGL.

- **S4.** The loss landscape and training loss curve visualizations in Figures 5 and 6 offer concrete evidence that CAL encourages smoother optimization geometry with wider valleys, consistent with the paper's claim that the model "possesses superior generalization performance."

---

**Weaknesses**

- **W1.** The foundational claim that "deeper FFNs completely disregard these dependencies, focusing solely on capturing the temporal dynamics within each individual variable" is stated without direct evidence. The self-attention output $H_{sa}$ already encodes mixed inter-variate representations. In the standard Transformer residual structure, $H = H_{sa} + FFN(H_{sa})$, the IVD information present in $H_{sa}$ is additively carried through the FFN sublayer regardless of the FFN's own computation. It is true that in Variate Transformers such as iTransformer, the FFN operates as a point-wise MLP applied independently to each variate token and therefore does not explicitly model cross-variate interactions. However, this does not imply that IVD information is lost after the FFN, because the residual connection structurally preserves it. The authors do not present any probing experiments, representational similarity analysis (e.g., Centered Kernel Alignment), or diagnostic visualizations to confirm that IVD information is genuinely discarded after the FFN sublayer. Without this, the core motivation for DGL rests on an unverified assumption.

- **W2.** The paper asserts that the IVD inconsistency "results in capturing spurious correlations, severely compromising generalization capabilities under distribution shifts (OOD), and posing challenges for model optimization." The training curve improvements in Figure 5 and the performance gains in Table 2 demonstrate empirical benefits of DGL and CAL, but these do not constitute evidence for spurious correlation reduction or OOD generalization improvement. The experimental design follows the standard in-distribution temporal split (train/val/test on the same dataset), which does not constitute an OOD evaluation. No dedicated distribution-shift benchmark or out-of-domain evaluation is provided, and no prior work is cited to establish this specific failure mode in Variate Transformers.

- **W3.** FMR is positioned as a replacement for timestamp embeddings, but the paper does not explain how or why FMR can substitute for the information that timestamp embeddings provide. Timestamp embeddings encode temporal positional information such as periodicity, seasonality, and time-of-day patterns. For FMR to serve as a functional replacement, the paper would need to demonstrate at minimum one of the following: that the learned frequency masks $\mathcal{M}$ capture representations equivalent to what timestamps encode, that the DCT-based decomposition implicitly recovers periodic positional structure, or that removing timestamps causes a specific and identifiable performance deficit that FMR then recovers. None of these analyses are present. The only justification offered is that "the frequency domain introduces a powerful inductive bias of a global receptive field," which describes FMR's own property but does not establish any functional equivalence with timestamp embeddings. This gap means the design choice to replace timestamps with FMR lacks a principled basis. Beyond this, no ablation isolates the effect of timestamp removal from the effect of frequency-domain feature enhancement. A controlled comparison is needed among (a) the original backbone with timestamps, (b) the backbone with timestamps simply removed, and (c) the backbone with FMR substituted. Without this, it remains unclear whether performance improvements are due to the removal of harmful timestamp information or to the additional representational capacity introduced by FMR. Furthermore, the stated motivation that "in many real-world application scenarios, future timestamps are unavailable" is factually imprecise: future calendar timestamps are known inputs, while future values are what must be predicted. Models such as the Temporal Fusion Transformer explicitly use calendar features (hour of day, day of week, etc.) as known future inputs. This mischaracterization further weakens the motivation for timestamp removal. The NeurIPS 2024 paper "Rethinking the Power of Timestamps for Robust Time Series Forecasting: A Global-Local Fusion Perspective" suggests that the problem may lie in suboptimal timestamp utilization rather than timestamp use per se.

- **W4.** The claim that "GNNs excel at capturing differentiated indirect and sparse dependencies" relative to global self-attention is derived solely from Figure 10, which is an analysis on ETTh1. This is insufficient to establish a general property applicable across 13 diverse datasets with fundamentally different dependency structures, including densely connected sensor networks (PEMS03/04/07/08), weakly coupled financial series (Exchange), and high-dimensional traffic sensor data (Traffic). The finding should be either qualified as dataset-specific or substantiated by multi-dataset analysis.

- **W5.** Section 4.6 states that "without the CAL constraint, neither the self-attention layer nor the DGL success to capture critical dependencies." However, the same section and Tables 15 and 7 report that CGTFra without CAL achieves MSE 0.157 and MAE 0.195 on the Weather test set, "both outperforming existing methods." If both representation learning mechanisms fail to capture critical dependencies without CAL, the mechanism by which CGTFra without CAL nonetheless achieves state-of-the-art forecasting performance is left unexplained. The authors' inline justification -- that this "likely represents an inherent modeling challenge for the network" -- does not resolve the contradiction. This inconsistency directly undermines the claimed necessity of CAL and must be addressed.

- **W6.** Table 1 reports baseline results taken from official published reports, while Table 2 reproduces results under the authors' experimental environment using "identical hyperparameters as publicly released by the authors." This dual-source protocol makes cross-table comparisons unreliable, as observed numerical differences may reflect environmental rather than methodological factors.  At a minimum, the most directly competing baselines (DUET, iTransformer, VCformer) should be reproduced under a unified experimental environment. Out-of-memory cases in Table 2 should be resolved by reducing batch size or hidden dimension rather than left as missing entries, for experimental completeness.

- **W7.** The symbol $N$ is used simultaneously as the total number of variates and as a variate index. For instance, in the problem setup $X_N^{1:T}$ uses $N$ as an index, while the same $N$ denotes the total count elsewhere. The paper partially addresses this by using lowercase $n$ in some places (e.g., $H^{extend}[n, :]$, $n = 1, 2, ..., N$), but the inconsistency persists in the problem formulation. For mathematical rigor, the variate index should be consistently denoted as lowercase $n$ throughout all equations.

- **W8.** Equation (7) writes $\Theta_*^l = W_{gating} \odot Linear(H_{sa}^l) + \Theta_*^l$, where $\Theta_*^l$ appears on both sides. It is ambiguous whether the right-hand side refers to the value before or after the update. If this is intended as a residual in-place update, it should be written with an assignment arrow or with superscripts that explicitly distinguish the pre- and post-update values.

- **W9.** In Equation (5) (the inverse DCT), the summation variable $t$ plays the role of the output time index in the reconstructed domain, but it shares the same symbol as the input time index $t$ in Equation (1) (the forward DCT). The two variables operate in different domains: the input has length $T$ while the output after resampling has length $D$. Using the same symbol without explicit redefinition introduces notational ambiguity.

- **W10.** While FMR (Equations 1-5), DGL (Equations 6-7), and CAL (Equations 8-9) are all formalized at the equation level, the computation of $H_{sa}$ is only described in prose. For internal consistency with the paper's own convention of providing explicit equations for each module, the self-attention sublayer computation should also be given a formal equation.

- **W11.** The use of learnable static node embeddings $\Theta_1^l, \Theta_2^l$ with an adaptive gating mechanism is not explained in terms of design intent. It is unclear whether this construction is a novel contribution of this paper or a standard approach shared with prior works such as Sageformer or MSGNet. A brief discussion comparing this design to existing node embedding strategies would strengthen the methodological presentation.

- **W12.** The representation of variable 1 in Figure 3 is barely visible, weakening the visual argument intended to motivate DGL. More critically, DTW (a temporal alignment distance) and PCC (a linear correlation measure) are heterogeneous metrics with fundamentally different mathematical properties. Treating them interchangeably under the claim that "higher dependency implies smaller spatial distance" conflates these properties without justification. A more careful characterization of what each metric captures, and why both support the same conclusion, is needed.

- **W13.** The Related Work section largely enumerates papers without discussing their limitations in relation to the problem CGTFra targets. For each group of prior work (timestamp embedding approaches, intra-series models, inter-series models), there should be a brief statement of the specific gap or failure mode that motivates the proposed design.

- **W14.** The claim that "the frequency domain introduces a powerful inductive bias of a global receptive field" is asserted but not analyzed. It would be more convincing to show, for example, that the learned frequency masks consistently retain low-frequency components, or that the resampled representations capture patterns that the original time-domain inputs miss, either through ablation or through visualization of the learned mask distributions.

- **W15.** The following should be corrected: "projection metrices" should be "projection matrices"; "existing researches" should be "existing research"; "principles that is plug-and-play" should be "that are"; "additional valuation metrics" should be "evaluation metrics"; "the effectiveness of CGTFra to modeling" should be "in modeling"; "on the Traffic" should be "on the Traffic dataset"; "sota" should be consistently capitalized as "SOTA"; "The self-attention layer capture a weight" should be "captures"; "averaged from four forecasting horizons" should be "averaged over."

---

> ### Author Rebuttal · Authors · 2026-03-26
>
> We thank you for the detailed feedback. Due to the **word count constraints**, we have prioritized responses to address the most critical concerns.
>
> **W1 & W12: the Motivation of DGL**
>
> We would like to clarify that the core motivation for introducing DGL is to **actively refine and capture high-order IVD structures**. We provide rationale and empirical evidence below:
> * 1. Passive transmission vs. Active refinement:
> While the residual connection ensures that IVD is carried forward, the FFN itself operates purely as a point-wise MLP. The IVD is merely passively transmitted **without any further structural refinement, thereby facing the inconsistent IVD modeling**. (Our analysis Line 96): “Self-attention layers typically capture global and dense dependencies; however, these dense connections often include spurious correlations that ...”, evidenced by Figs. 7 & 10). By explicitly modeling IVD with DGL, we transform this passive transmission into an active structural learning process, capturing indirect and high-order IVD and yielding a superior optimization landscape and enhanced robustness (see Figs. 5,6,19-22).
> * 2. Empirical evidence via T-SNE:
> Notably, **we have provided t-SNE plots of the hidden layers** (see **Figs. 3, 8, and 16-18**) to represent the IVD captured by latent embeddings (from FFN or DGL) after the residual connection. Given that IVD are complex & dynamic, we employ PCC and DTW (details in Appendix C) to measure **pairwise** distances, to better evaluate IVD (Response to W12: metric’s function). According to the PCC and DTW in Fig 9, and the IVD stability in ETTh1, Variable 1 and Variable 3 exhibit tight dependencies (their curves are more direct clarification). Therefore, in Figs 3 and 17, their t-SNE naturally show significant overlap (Response to W12: **why Variable 1 is barely visible**. We also provide T-SNE of ground truth in https://anonymous.4open.science/r/CGTFra/TSNEGroundTruth.pdf). By observing the T-SNE across **1,500 test samples** in both datasets, introducing DGL enables the model to capture more plausible interactions (e.g., see the yellow Variable 5 in Fig 17(a) vs. (b)).
>
> Following your suggestion, we conducted a Similarity Analysis using CKA (averaged on all test samples of ETTh1) to quantify how the IVD evolves across sublayers. Under identical residual constraints, the structural update intensity (defined as 1−CKA) of our DGL is 2.77 times that of the FFN (FFN CKA: 0.9934 vs. DGL CKA: 0.9817).
>
> **W3 and W14: FMR**
>
> FMR captures periodicity and global temporal structure in a data-driven manner, which partially overlaps with the role of timestamp embeddings. Actually, **we have provided extensive analyses in Appendix B to demonstrate that FMR preserves and reinforces periodicity**. Specifically, **Figure 12** shows that FMR captures a spectral distribution much closer to the ground truth compared to linear embedding using timestamp encoding. Besides, **Figure 13** reveals that the learned masks predominantly retain low-frequency components (representing the trend and periodicity) while the mask for Variable 3 suppresses more high-frequency components (representing noise and fine-grained details). Finally, **Table 4 explicitly provides the three comparisons you requested: (a) the original backbone; (b) timestamps removed; and (c) the backbone with FMR**. **We cordially invite you to examine these specific sections for verification**.
>
> **W4: Qualitative Analysis**
>
> In fact, our qualitative analysis **is not limited to the ETTh1**; **Fig 7 provides the visualization for the Weather**. These results consistently demonstrate that the self-attention layer captures dense correlations, whereas the GNN identifies sparser structural relationships (in Figs. 7(a) and 10(a)). **We opted not to visualize the Traffic and PEMS series because of their high dimensionality (e.g., 862 variables for Traffic). Visualizing an 862×862 matrix renders fine details indistinguishable.** We have supplemented the IVD visualization for the Exchange and Traffic (yielding consistent conclusions): https://anonymous.4open.science/r/CGTFra/AdditionalIVD.pdf
>
> **W6: Comparison Results**
>
> We would like to clarify that in the most scenarios, **the reproduced results in Table 2 are inferior to those reported in their original papers. Utilizing their reported metrics in Table 1 actually provides a more favorable benchmark for baselines, a common practice adopted in many recent studies**. Further, we provide the comparative results under an identical environment at: https://anonymous.4open.science/r/CGTFra/UnifiedEnvironment.pdf
>
> **W2,W5,W7-W11,W13, and W15: Presentation**
> For a detailed analysis of CAL, please refer to our Response 2 to Reviewer rJ1n.
> We will incorporate valuable comments for manuscript refinement.
>
> We hope that these responses have addressed your concerns and bridged the gap in our perspectives. Accordingly, **we sincerely hope that you might reconsider and re-evaluate our manuscript.**

---

> > ### Author Rebuttal · Reviewer_P9RR · 2026-04-01
> >
> > I thank the authors for their constructive rebuttal and supplementary materials (CKA analysis, ground truth T-SNE, Exchange/Traffic IVD visualizations, unified environment results). These demonstrate genuine effort, and I have carefully reviewed all materials including the linked PDFs.
> >
> > Before detailing my assessment, I want to clarify the principle underlying my evaluation: I consider it essential that a paper's identified problem and its proposed solution be well-aligned, with the claimed problem clearly substantiated and the solution demonstrably addressing that specific problem. My remaining concerns largely stem from gaps in this alignment.
> >
> > **W1 (Core motivation — FFN discards IVD):**
> > The CKA analysis is appreciated as a quantitative attempt. However, the FFN CKA value of 0.9934 indicates that 99.34% of input structure is preserved after FFN processing, which paradoxically supports my original argument that residual connections preserve IVD rather than the paper's claim that FFNs "completely disregard" dependencies (line 80–81). The "2.77×" framing is based on a 0.0117 absolute difference in 1−CKA, which I find insufficient to justify the strong motivation for the entire DGL module. The rebuttal's reframing as "passive transmission vs. active refinement" is more accurate but represents a significant departure from the paper's original claims. The probing experiments I originally requested remain unprovided.
> >
> > **W2 (OOD claim):**
> > This concern is entirely unaddressed. The rebuttal deferred to Reviewer rJ1n's response, which contains no OOD experiments or analysis. The claim that IVD inconsistency "severely compromises generalization capabilities under distribution shifts (OOD)" (line 94–95) is a strong assertion that lacks any supporting evidence in its current form.
> >
> > **W3 (FMR as timestamp replacement):**
> > I acknowledge Table 4 provides the three-way comparison. However, functional equivalence between FMR and timestamps remains unestablished, and the authors' concession of "partial overlap" is more accurate than the "replacement" framing. The motivation that "future timestamps are unavailable" remains factually inaccurate.
> >
> > **W4 (GNN advantage generalization):**
> > The Exchange visualization supports the claim. However, the Traffic visualization is deeply concerning — both self-attention and DGL show near-zero dependency weights ($\approx$0.000) even with CAL applied. This directly contradicts the universal claim that GNNs excel at capturing indirect dependencies, and is consistent with the authors' own acknowledged limitation in Appendix P. The coexistence of positive evidence (ETTh1, Weather, Exchange) and clearly negative evidence (Traffic) means the claim cannot be stated as a general property.
> >
> > **W5 (CAL necessity):**
> > The soft-regularization reframing is reasonable. However, the Traffic IVD visualization deepens the contradiction: CAL-applied CGTFra still shows near-zero IVD weights yet achieves competitive performance. Combined with the Weather case (SOTA without CAL), a consistent pattern emerges where forecasting performance appears largely decoupled from explicit IVD capture quality. This raises a fundamental question about whether deep-layer IVD modeling is truly the primary driver of CGTFra's improvements, or whether other factors (FMR input enhancement, multi-hop GCN's temporal feature extraction) contribute more significantly.
> >
> > **W6 (Unified comparison):**
> > The unified results are appreciated but cover only 4 of 9 baselines from Table 1. FilterNet — which outperforms CGTFra on ETTm1 (0.384 vs. 0.388) and ETTm2 (0.276 vs. 0.277) — is absent. In the unified comparison, iTransformer (0.422) outperforms CGTFra (0.427) on Traffic, which is consistent with the near-zero IVD weights observed in the supplementary Traffic visualization.
> >
> > **Overall:**
> > The empirical contributions of CGTFra (plug-and-play generality, broad benchmark performance) remain commendable. However, returning to the principle I stated above, I observe a persistent misalignment between the paper's claimed problems and the evidence supporting them: the core motivation (FFN discards IVD) is not substantiated by the CKA evidence; the OOD claim lacks any supporting experiments; and the link between IVD modeling quality and forecasting performance is weaker than claimed, as evidenced by both Weather (SOTA without CAL) and Traffic (near-zero IVD with CAL). Until this alignment between problem identification and solution validation is strengthened, I maintain my Overall Recommendation (3) and Confidence (4).

---

> > > ### Author Response · Authors · 2026-04-03
> > >
> > > Dear Reviewer P9RR,
> > >
> > > Thank you again for your detailed comments.
> > >
> > > **W1 CKA analysis**
> > >
> > > We agree that the phrase ‘completely disregard’ might introduce ambiguity. We will update the manuscript to clarify that “deep FFNs do not explicitly model IVD, nor do they actively refine these dependencies”. Our primary motivation has always been to address the inherent limitations: (shallow: dynamic IVD modeling VS deep: direct passive transmission via residual connection). Thus, the **computational mechanism** of FFN completely **disregards IVD modeling** (Line 21 in the Abstract) (This is what we originally intended to convey, and we apologize for any confusion caused.)
> > >
> > > We would also like to point out that while interpreting 0.9934 intuitively as ‘99.34% structure preservation’ is straightforward, CKA in high-dimensional spaces can be viewed as a normalized cosine similarity between Gram matrices, **rather than a linear percentage of retained information**.
> > >
> > > As analyzed in manuscript, the shallow self-attention layer captures dense and noisy IVD (Line 97). The noisy, dense topology is directly carried forward by residual connection without meaningful IVD updating (FFN CKA 0.9934).
> > >
> > > In high-dimensional spaces, absolute difference of 0.0117 in 1-CKA may reflect non-negligible changes and represent the active pruning and the refinement of a sparse, robust graph topology by DGL. **Since CKA is limited in reflecting the true effectiveness of DGL, even though we provided CKA analysis in the first-round rebuttal following your suggestion, this did not lead to a consensus between us. This is why we instead employ t-SNE visualizations (see Figures 3,8,16-18) to indicate that, with the introduction of DGL, more accurate IVD are learned compared to FFN.** And, the generality verifications in Table 2 are the most direct method to clarify the effectiveness of DGL compared to FFN.
> > >
> > > **W2 OOD claim**
> > >
> > > We are grateful for your reminder. To avoid any potential ambiguity regarding strict cross-domain OOD definitions, we will revise the manuscript to remove the term ‘OOD’, retain only ‘distribution shifts’.
> > >
> > > Our initial use of “the distribution shifts (OOD)” was inspired by ‘STONE: A Spatio-temporal OOD Learning Framework Kills Both Spatial and Temporal Shifts’. We provide distribution visualizations at: https://anonymous.4open.science/r/CGTFra/DistributeShift.pdf. Since our datasets strictly follows a **chronological split** (e.g., 6:2:2), assessing performance on the newly observed test set essentially evaluates the model’s pure generalization capability against non-stationarity (distribution shift [DUET]).
> > >
> > > **W3 FMR vs Timestamp**
> > >
> > > We agree that some of our phrasing regarding FMR might be imprecise. We will refine these statements in the revised manuscript. We would also like to clarify that in Line 117, we specifically used ‘potential replacement’ rather than a definitive ‘replacement’.
> > >
> > > **W4 & W5 (DGL & CAL)**
> > >
> > > To facilitate a clearer understanding, we provide a mind map: https://anonymous.4open.science/r/CGTFra/CGTFraCore.pdf
> > >
> > > **Regarding each core contribution**, Table 2 shows the incremental performance comparisons when **integrating FMR, DGL, and CAL into baselines** (note that the inclusion of CAL inherently incorporates DGL). Specifically, for DUET, +DGL and +CAL enhances performance across the vast majority of scenarios (except Solar & Traffic). For iTransformer, FMR yields more substantial improvements than DGL and CAL on ETTm2, ETTh2, & Solar. Conversely, for the remaining datasets (except Traffic), the simultaneous integration of DGL and **CAL** on top of the baseline consistently delivers more obvious gains—in ETTm1, ETTm2, ETTh1, Exchange, Weather, & ECL. Similar analyses apply to the other evaluated baselines.
> > >
> > > Other merits of CAL can be seen in above mind map.
> > >
> > > We believe the emergence of zero values in the learned correlation matrix for Traffic is structurally justifiable. **Capturing IVDs across 862 variables is an exceptionally challenging task** (see DUET, Soatten, VCformer in Table 1). **This exact bottleneck in high-dimensional IVD modeling motivates the research community continuous optimization efforts.**
> > >
> > > We do not shy away from the fact that CGTFra yields zeros on Traffic. To the best of our knowledge, we have found no existing studies that provide a comprehensive analysis of IVD for large-scale datasets, e.g., Traffic. Nevertheless, we provide a mechanistic explanation: Figure 11(a) shows that Traffic possesses dominant temporal periodicities. We postulate that despite the zero weight, CGTFra retains SOTA due to a strong compensatory mechanism in its temporal (intra-series) modeling. Specifically, the FMR and the two linear layers in DGL secure robust temporal features, ensuring accurate forecasting even w/o dense inter-series guidance (see aba1,aba2 & aba3 in Table 3).
> > >
> > > **W6 Unified comparison**: We provide the comparative results at: https://anonymous.4open.science/r/CGTFra/UnifiedEnvironment0404.pdf

---

### Official Review · Reviewer_rJ1n · 2026-03-13

**Soundness:** 2
**Presentation:** 2
**Significance:** 2
**Originality:** 2
**Overall Recommendation:** 4
**Confidence:** 4

**Summary:**

This paper proposes CGTFra, a Graph Transformer framework for multivariate time series forecasting (MTSF). It identifies a structural inconsistency in existing Variate Transformers (e.g., iTransformer): inter-variate dependencies (IVD) are modeled only in shallow self-attention layers while deep feed-forward layers ignore them entirely.

**Compliance With Llm Reviewing Policy:**

Affirmed.

**Final Justification:**

In light of the comprehensive ablation study and the overall consistency of improvements, I will raise my overall recommendation from 3 to 4.

**Key Questions For Authors:**

Please refer to weaknesses.

**Limitations:**

yes

**Strengths And Weaknesses:**

### Strengths

1. Clear problem identification with solid theoretical grounding. The paper pinpoints a genuine structural flaw in Variate Transformers and backs the CAL design with a rigorous VIB-based derivation, rather than treating it as ad-hoc regularization.

2. Strong generalizability via plug-and-play design. All three modules (FMR, DGL, CAL) integrate seamlessly into existing architectures without modifying hyperparameters, validated across 9 baselines and 13 datasets.


### Weaknesses

1.  Unstable gains on high-variable datasets. On Solar and Traffic, incorporating DGL and CAL into strong baselines yields marginal or even negative improvements, contradicting the claim of universal effectiveness.

2. CAL's alignment assumption may conflict with its own motivation. Forcing alignment between the dense self-attention map and the sparse GNN adjacency matrix risks suppressing GNN's ability to capture indirect dependencies — the very reason DGL was introduced.

3. Insufficient ablation design. CAL is never evaluated independently without DGL, making it impossible to disentangle their individual contributions. Ablations also cover only 5 of the 13 datasets.

---

> ### Author Rebuttal · Authors · 2026-03-25
>
> We appreciate your review and would like to take this opportunity to further discuss the technical details of the CGTFra framework.
>
> **1: Insufficient ablation: (1) CAL is never evaluated independently w/o DGL; (2) cover only 5 of the 13 datasets**
>
> We respectfully clarify the architectural logic flow of our proposed method. **The CAL is specifically formulated based on the premise of the DGL** module. Since CAL operates directly on the graph structures generated by DGL, it cannot be integrated into the framework as a standalone component if DGL is removed.
>
> Regarding the selection of datasets for our initial ablation study, we followed the standard evaluation conventions of well-known methods. For instance, both **DUET (evaluating on ETTh2, ETTm2, Weather, and Traffic)** and **iTransformer (selecting ECL, Traffic, Weather, and Solar)** conducted their ablation analyses on a subset of **four** representative datasets.
> To thoroughly address your concerns, we have conducted additional ablations on all 13 datasets. The results are at the following anonymous link: https://anonymous.4open.science/r/CGTFra/Ablation%20Study.pdf
>
> **2: CAL’s alignment assumption**
>
> We fully understand your concern. However, we respectfully clarify that **CAL acts as a soft regularization that encourages alignment while preserving complementary structures**. Please allow us to elaborate on the underlying logic step-by-step:
>
> * At the architectural level: DGL operates directly on the output of the self-attention layer. As we analyzed (see Figs. 7 and 10), DGL is capable of capturing indirect dependencies that the self-attention layer misses—a point you graciously acknowledged.
>
> * At the optimization level: CAL is introduced from the loss function perspective to guide parameter optimization (**not only for DGL’s parameters**). The total loss (Eq. 10, Line 248) comprises the forecasting error $\mathcal{L}\_{MAE}$ and the alignment loss $\mathcal{L}\_{align}$, and **we can control the intensity of the consistency alignment loss by tuning the hyperparameter $\lambda$** (According to the sensitivity analysis in Fig. 23, $\lambda$ becomes more sensitive when forecasting over extended horizons. Conversely, for shorter prediction lengths (e.g., 96, 192, and 336), the impact of $\lambda$ remains marginal). According to our rigorous derivations (Page 16, Line 862), CAL provides a theoretical upper bound for the mutual information $I(Z;X)$, and $A$ (DGL) acts as a structural bottleneck that filters out the dense noise in MCM, retaining only the structured information pertinent to predicting $Y$.
>
> * At the empirical level: The two specific case studies—Weather (Page 8, Fig. 7) and ETTh1 (Page 11, Fig. 10)—demonstrate that imposing the CAL consistency constraint enables DGL to capture more rational dependency correlations without sacrificing its ability to model indirect dependencies. Furthermore, CAL even facilitates an additional dependency correction for the self-attention layer itself (see Fig. 7(d)). In addition, to further indicate the effectiveness of CAL, we supplemented the IVD plots for the Exchange and Traffic (yielding consistent conclusions): https://anonymous.4open.science/r/CGTFra/AdditionalIVD.pdf
>
> **Beyond enabling more robust inter-variable dependency modeling, the incorporation of CAL yields an improved optimization landscape and significantly enhances the model’s generalization capabilities** (evidenced by Figs. 5, 6, and 19-22). In summary, we sincerely believe that the efficacy and necessity of CAL are well-substantiated and indispensable to the framework.
>
> **3: Unstable gains on Solar and Traffic**
>
> We fully understand your perspective. It is worth noting that the baselines evaluated—such as DUET (KDD, 2025), CASA (IJCAI, 2025), iTransformer (ICLR, 2024), and VCformer (IJCAI, 2024)—are very recent and represent the SOTA in the field. **For generality verification of DGL and CAL**, we acknowledge that **achieving further performance improvements** on these advanced baselines (on Solar and Traffic) is highly challenging. By applying our **general consistency modeling principle** with **strictly identical hyperparameters**, we **achieved performance gains in 109 out of 150 forecasting scenarios in Table 11.** (peak improvement of **10.2%** over VCformer on ECL). And for ECL (also a extremely large dataset), introducing DGL and CAL can obtain more consistent and significant performance gains. Furthermore, **our proposed CGTFra obtains significantly better performance on Solar and Traffic (compared to DUET, CGTFra reduces MSE by 5.1% (5.49%) on the Traffic (Solar)).**
>
> Due to strict rebuttal time constraints, fully optimizing our method across all datasets (for Traffic and Solar) is currently infeasible. We will dedicate our future work to exploring these comprehensive optimizations.
>
> We hope that these responses adequately address your comments.
> **And we sincerely hope that you could consider re-evaluating our manuscript.**

---

> > ### Author Rebuttal · Reviewer_rJ1n · 2026-04-06
> >
> > I thank the authors for the detailed rebuttal and the extended ablation study across all 13 datasets. In light of the comprehensive ablation study and the overall consistency of improvements, I will raise my overall recommendation from 3 to 4.

---

> > > ### Author Response · Authors · 2026-04-06
> > >
> > > We sincerely appreciate your acknowledgement. We will carefully revise our manuscript in accordance with your valuable suggestions.

---

### Decision · Program_Chairs · 2026-04-30

**Decision:**

Accept (regular)

**Comment:**

Three reviewers are positive; one reviewer (P9RR) remains negative (score 3). The negative reviewer acknowledges the paper’s strong empirical contributions (plug‑and‑play design, extensive benchmarks) but raises two unresolved concerns: (1) the claim that FFNs “disregard” IVD modeling is not convincingly supported by the CKA/t‑SNE evidence; (2) the causal link between IVD modeling and forecasting performance is weakened by counter examples on Weather and Traffic.

The paper’s engineering contributions are solid and practically useful. However, the motivational claims are over‑stated relative to the evidence. The authors should carefully revise the paper if it can be accepted.